# Deep Weight Factorization: Sparse Learning Through the Lens of Artificial Symmetries

**Chris Kolb, Tobias Weber, Bernd Bischl, & David Rügamer**
Department of Statistics, LMU Munich, Munich
Munich Center for Machine Learning (MCML), Munich
{chris.kolb,tobias.weber,bernd.bischl,david}@stat.uni-muenchen.de

## Abstract

Sparse regularization techniques are well-established in machine learning, yet their application in neural networks remains challenging due to the non-differentiability of penalties like the $L_1$ norm, which is incompatible with stochastic gradient descent. A promising alternative is shallow weight factorization, where weights are decomposed into two factors, allowing for smooth optimization of $L_1$-penalized neural networks by adding differentiable $L_2$ regularization to the factors. In this work, we introduce deep weight factorization, extending previous shallow approaches to more than two factors. We theoretically establish equivalence of our deep factorization with non-convex sparse regularization and analyze its impact on training dynamics and optimization. Due to the limitations posed by standard training practices, we propose a tailored initialization scheme and identify important learning rate requirements necessary for training factorized networks. We demonstrate the effectiveness of our deep weight factorization through experiments on various architectures and datasets, consistently outperforming its shallow counterpart and widely used pruning methods.

## 1 Introduction

Making models sparse is a contemporary challenge in deep learning, currently attracting a lot of attention. Among the more prominent methods to achieve sparsity are model pruning methods (Gale et al., 2019; Blalock et al., 2020) and regularization approaches sparsifying the model during training (Hoefler et al., 2021). While in statistics and machine learning, sparse regularization approaches are well-established (see, e.g., Tian & Zhang, 2022), the non-smoothness of sparsity penalties such as the $L_1$ norm impedes the optimization of neural networks when using classical stochastic gradient descent (SGD) optimization. A possible solution that allows SGD-based optimization while inducing $L_1$ regularization is *weight factorization*. Originally proposed in statistics for linear models (Hoff, 2017), the idea of factorizing the weights $\mathbf{w} = \boldsymbol{\omega}_1 \odot \boldsymbol{\omega}_2$ to obtain a differentiable $L_1$ penalty on the product $\boldsymbol{\omega}_1 \odot \boldsymbol{\omega}_2$ has recently been adopted also in deep learning (see, e.g., Ziyin & Wang, 2023). This simple trick allows the integration of convex $L_1$-based sparsity into neural network training while promising direct applicability of familiar SGD. As shown in Fig. 1, the obtained sparsity of differentiable $L_1$ is superior to vanilla $L_1$ regularization. This holds even after applying additional post-hoc pruning, demonstrating that the inferior sparsity performance of vanilla $L_1$ is not just due to a suboptimal threshold but also the incompatibility of SGD and non-smooth penalties.

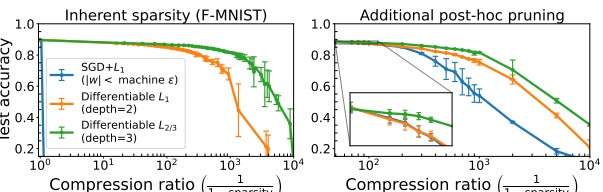

Figure 1: Sparsity-accuracy tradeoff using a vanilla $L_1$ penalization with SGD (blue) compared to (deep) weight factorization. Means and std. deviations over 3 random seeds are shown.

Given the success of (shallow) weight factorization, we study deep weight factorization in this work, i.e., factorizing $\mathbf{w} = \boldsymbol{\omega}_1 \odot \cdots \odot \boldsymbol{\omega}_D, D \geq 2$ (cf. Fig. 2). We investigate whether theoretical guarantees support the use of a *depth-D* factorization, whether it is beneficial for sparsity, what implications its usage has on training dynamics, and analyze other practices such as initialization.

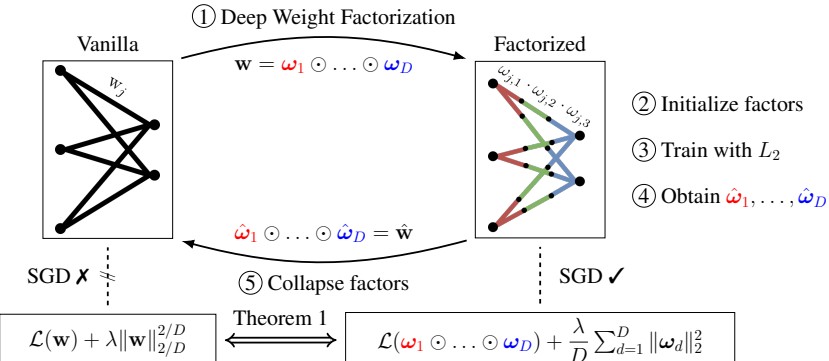

Figure 2: Overview of the proposed method (cf. Algorithm 2). Our approach proceeds by factorizing the neural network weights and running SGD on the factors $\boldsymbol{\omega}_d$ with weight decay. Post-training, the factors are collapsed again, with the resulting sparse solutions being minimizers of the non-smooth $L_{2/D}$-regularized objective.

**Our contributions** In this work, we address the aforementioned challenges and close an important gap in the current literature. We first theoretically show the equivalence of factorized neural networks with sparse regularized optimization problems for depth $D \geq 2$, allowing for differentiable non-convex sparsity regularization in any neural network. We then discuss optimization strategies for these factorized networks including their initialization and appropriate learning rate schedules. We also analyze the training dynamics of such networks, showing a particularly interesting connection between the evolution of weight norms, compression, accuracy, and generalization. Conducting experiments on a range of architectures and datasets, we further substantiate our theoretical findings and demonstrate that our proposed factorized networks usually outperform the recently proposed shallow factorization and yield competitive results to commonly used pruning methods.

## 2 BACKGROUND AND RELATED LITERATURE

### 2.1 NOTATION

Let $\{(\boldsymbol{x}_i, y_i)\}_{i=1}^n$ be the training data of independent samples $(\boldsymbol{x}_i, y_i) \in \mathcal{X} \times \mathbb{R}^c$, and $n, c \in \mathbb{N}$. Let $f(\mathbf{w}, \boldsymbol{x}) : \mathcal{X} \to \mathbb{R}^c$ denote a network realization for any $\mathbf{w} \in \mathbb{R}^p$. In general, we are interested in minimizing $\ell(\cdot, \cdot) : \mathbb{R}^c \times \mathbb{R}^c \to \mathbb{R}_0^+$ denoting a continuous per-sample loss. The $L_q$ *norm* of a vector $\mathbf{w} \in \mathbb{R}^p$ is defined as $\|\mathbf{w}\|_q = \left(\sum_{i=1}^p |\mathrm{w}_i|^q\right)^{1/q}$ for $q > 0$. Note that $L_q$ *regularizers* are defined differently as $\|\mathbf{w}\|_q^q$ and that for $q < 1$, only a non-convex quasi-norm is defined. For two vectors $\boldsymbol{\omega}_1, \boldsymbol{\omega}_2 \in \mathbb{R}^p$, we use $\odot$ to denote their element-wise multiplication. For an optimization problem $\min_{\mathbf{w}} \mathcal{L}(\mathbf{w})$, we denote $\hat{\mathbf{w}} := \arg\min_{\mathbf{w}} \mathcal{L}(\mathbf{w})$. Finally, the *compression ratio* (CR) is defined as the ratio of original to sparse model parameters.

### 2.2 DIFFERENTIABLE $L_1$ REGULARIZATION

Weight factorizations were previously mostly studied for regularized linear models or as toy models for deep learning theory. We briefly illustrate this using the idea of a differentiable lasso.

**Differentiable lasso** The original lasso objective is defined as

$$\min_{\mathbf{w} \in \mathbb{R}^p} \mathcal{L}_{\mathbf{w}, \lambda}(\mathbf{w}) := \sum_{i=1}^n \left(y_i - \boldsymbol{x}_i^\top \mathbf{w}\right)^2 + \lambda \|\mathbf{w}\|_1, \tag{1}$$

where $\lambda > 0$ promotes sparsity via the $L_1$ norm (Tibshirani, 1996). By factorizing $\mathbf{w}$ into $\boldsymbol{\omega}_1$ and $\boldsymbol{\omega}_2$ such that $\mathbf{w} = \boldsymbol{\omega}_1 \odot \boldsymbol{\omega}_2$, and replacing the non-differentiable $L_1$ penalty with an $L_2$ penalty on $\boldsymbol{\omega} = (\boldsymbol{\omega}_1, \boldsymbol{\omega}_2)$, we can obtain a differentiable formulation of the lasso (Hoff, 2017):

$$\min_{\boldsymbol{\omega}_1, \boldsymbol{\omega}_2 \in \mathbb{R}^p} \mathcal{L}_{\boldsymbol{\omega}, \lambda}(\boldsymbol{\omega}) := \sum_{i=1}^n \left(y_i - \boldsymbol{x}_i^\top (\boldsymbol{\omega}_1 \odot \boldsymbol{\omega}_2)\right)^2 + \frac{\lambda}{2}\left(\|\boldsymbol{\omega}_1\|_2^2 + \|\boldsymbol{\omega}_2\|_2^2\right), \tag{2}$$

The formulation Eq. (2) is equivalent to Eq. (1) in the sense that all minima of the non-convex objective in Eq. (2) are global and related to the unique lasso solution of Eq. (1) as $\hat{\boldsymbol{\omega}}_1 \odot \hat{\boldsymbol{\omega}}_2 = \hat{\mathbf{w}}$. Hoff (2017) proposes solving Eq. (2) via alternating ridge regression. However, this relies on the biconvexity of the problem and cannot be easily extended beyond linear models.

**Differentiable $L_1$ regularization in general neural networks**  Recently, Ziyin & Wang (2023) proposed applying a shallow factorization to arbitrary weights of a neural network. Coupled with weight decay, this allows obtaining a differentiable formulation of the sparsity-inducing $L_1$ penalty that can be optimized with simple SGD. Specifically, by factorizing the weights $\mathbf{w}$ of any neural network $f_{\mathbf{w}}(\mathbf{w}, \boldsymbol{x})$ as $\mathbf{w} = \boldsymbol{\omega}_1 \odot \boldsymbol{\omega}_2$, and applying $L_2$ regularization to the factors, the resulting optimization problem has the same minima as the $L_1$ regularized vanilla network. The key insight for the equivalence with $L_1$ regularization is that the factorization $\mathbf{w} = \boldsymbol{\omega}_1 \odot \boldsymbol{\omega}_2$ introduces a rescaling symmetry in the (unregularized) loss $\mathcal{L}_{\boldsymbol{\omega},0}$.

**Definition 1** (Rescaling Symmetry). Let the parameters of a loss function $\mathcal{L}_{\boldsymbol{\theta}}(\boldsymbol{\theta})$ be partitioned as $\boldsymbol{\theta} = (\boldsymbol{\omega}_1, \boldsymbol{\omega}_2, \boldsymbol{\theta}_0)$, with $\boldsymbol{\theta}_0$ denoting the remaining parameters. Then $\mathcal{L}_{\boldsymbol{\theta}}(\boldsymbol{\theta})$ possesses a rescaling symmetry w.r.t. arbitrary parameters $\boldsymbol{\omega}_1, \boldsymbol{\omega}_2$ belonging to $\boldsymbol{\theta} = (\boldsymbol{\omega}_1, \boldsymbol{\omega}_2, \boldsymbol{\theta}_0)$ if for any $c \neq 0$:

$$\mathcal{L}_{\boldsymbol{\theta}}(\boldsymbol{\omega}_1, \boldsymbol{\omega}_2, \boldsymbol{\theta}_0) = \mathcal{L}_{\boldsymbol{\theta}}(c \cdot \boldsymbol{\omega}_1, c^{-1} \cdot \boldsymbol{\omega}_2, \boldsymbol{\theta}_0) \ \forall \boldsymbol{\theta}.$$

While previous works mainly studied rescaling symmetries naturally arising in, e.g., homogeneous activation functions (Neyshabur et al., 2015; Parhi & Nowak, 2023), weight factorization constitutes an *artificial* symmetry that is independent of $\mathcal{L}$, and by extension also of $\ell(\cdot, \cdot)$ and $f_{\mathbf{w}}(\cdot, \boldsymbol{x})$. This applicability to any parametric problem designates artificial symmetries as a powerful tool for constrained learning (Ziyin, 2023; Chen et al., 2024). Intuitively, the additional $L_2$ regularization enforces preference for min-norm factorizations $(\boldsymbol{\omega}_1^*, \boldsymbol{\omega}_2^*)$ among all feasible factorizations of a given $\mathbf{w}$ (Fig. 3). At such a min-norm factorization of $\mathbf{w}$, the $L_2$ penalty in Eq. (2) reduces to $\|\boldsymbol{\omega}_1^* \odot \boldsymbol{\omega}_2^*\|_1 = \|\mathbf{w}\|_1$, effectively inducing $L_1$ regularization on the collapsed parameter. This approach allows for implementing $L_1$ regularization in general networks using GD without requiring specialized algorithms to handle non-differentiable regularization.

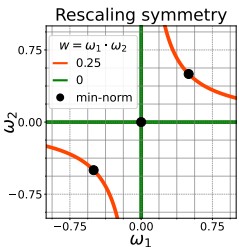

Figure 3: Scalar rescaling symmetry and min-norm factorizations.

We refer to Appendix A for additional related methods and discussion. Appendix B provides some intuition why weight factorization with $L_2$ regularization promotes sparse solutions based on Fig. 3.

## 3 THEORETICAL RESULTS

Based on a given network specification of $f(\mathbf{w}, \boldsymbol{x})$, we study its *depth-D* factorization with $D \geq 2$, which we call Deep Weight Factorization (DWF) and is defined as follows:

**Definition 2** (Deep Weight Factorization). A depth-$D$ factorization with $D \in \mathbb{N}_{\geq 2}$ of an arbitrary neural network $f_{\mathbf{w}}(\mathbf{w}, \boldsymbol{x})$, $\mathbf{w} \in \mathbb{R}^p$, is given by $f_{\mathbf{w}}(\boldsymbol{\omega}_1 \odot \ldots \odot \boldsymbol{\omega}_D, \boldsymbol{x})$ with $\boldsymbol{\omega} = (\boldsymbol{\omega}_1, \ldots, \boldsymbol{\omega}_D)$ and factors $\boldsymbol{\omega}_d = (\omega_{1,d}, \ldots, \omega_{p,d}) \in \mathbb{R}^p, d \in [D]$. The original and factorized parameters are related through $\mathbf{w} = \boldsymbol{\omega}_1 \odot \ldots \odot \boldsymbol{\omega}_D =: \boldsymbol{\varpi}$, where $\boldsymbol{\varpi}$ denotes the collapsed parameter. Further, a factorization depth is called *shallow* for $D = 2$ and otherwise *deep*.

In this work, we focus on unstructured sparsity. This means all weights and biases in $f_{\mathbf{w}}$ are factorized using DWF. In principle, however, the factorization can also be selectively applied to arbitrary subsets of the parameters $\mathbf{w}$. Importantly, while DWF does not alter the expressive capacity of the underlying network $f_{\mathbf{w}}$, it drastically alters the optimization dynamics and enables sparse learning in conjunction with $L_2$ regularization or weight decay. Therefore, our focus lies on examining the effects of $L_2$ regularization and the behavior of SGD optimization in factorized networks.

The regularized training loss with DWF and regularization strength $\lambda > 0$ is defined to be

$$\mathcal{L}_{\boldsymbol{\omega},\lambda}(\boldsymbol{\omega}) = \frac{1}{n} \sum_{i=1}^{n} \ell\left(y_i, f_{\mathbf{w}}\left(\boldsymbol{\omega}_1 \odot \ldots \odot \boldsymbol{\omega}_D, \boldsymbol{x}_i\right)\right) + \frac{\lambda}{D} \sum_{d=1}^{D} \|\boldsymbol{\omega}_d\|_2^2. \tag{3}$$

For a given $\mathbf{w}$, applying DWF to the training objective introduces an infinite set of feasible factorizations $\{(\boldsymbol{\omega}_1, \ldots, \boldsymbol{\omega}_D) : \boldsymbol{\varpi} = \mathbf{w}\}$ that leave the network output $f_{\mathbf{w}}(\mathbf{w}, \boldsymbol{x})$ and loss invariant. Those

factorizations, however, differ in their respective norms. While the norm of individual factors can grow arbitrarily large, there exist factorizations that minimize the Euclidean norm, or equivalently, the factor $L_2$ penalty. $L_2$ regularization thus biases the optimization toward min-norm factorizations. This regularization ensures that the parameter representation strives to be evenly distributed across factors. The following result formalizes the necessary optimality conditions for the factorized objective, identifying solution candidates as those that achieve minimal norm configuration.

**Lemma 1** (Necessary condition for solution and minimum $L_2$ penalty). *Let $\boldsymbol{\omega} = (\boldsymbol{\omega}_1, \ldots, \boldsymbol{\omega}_D) \in \mathbb{R}^{Dp}$ be a local minimizer of $\mathcal{L}_{\boldsymbol{\omega}, \lambda}(\boldsymbol{\omega})$. Then i) $|\omega_{j,1}| = \ldots = |\omega_{j,D}|$ for all $j \in [p]$, and ii) the factor $L_2$ penalty reduces to $D^{-1} \sum_{d=1}^{D} \|\boldsymbol{\omega}_d\|_2^2 = \|\boldsymbol{\varpi}\|_{2/D}^{2/D}$.*

Using the result of Lemma 1, we introduce the concept of **factor misalignment** to quantify the distance from balanced factorizations required for solutions. Specifically, the factor misalignment is defined as $M(\boldsymbol{\omega}) = D^{-1} \sum_{d=1}^{D} \|\boldsymbol{\omega}_d\|_2^2 - \|\boldsymbol{\varpi}\|_{2/D}^{2/D}$ and captures the difference between the factor $L_2$ penalty and that of a balanced minimum-norm factorization of the same collapsed $\boldsymbol{\varpi}$. The misalignment satisfies $M(\boldsymbol{\omega}) \geq 0$, with equality if and only if the factorization is balanced. This allows us to restrict the search for potential solutions to balanced factorizations $M(\boldsymbol{\omega}) = 0$, as required by Lemma 1. Lemma 4 in Appendix C.4 describes the remarkable implications of reaching zero misalignment for SGD dynamics, collapsing the dynamics to a constrained symmetry-induced subspace in which the parameters remain for all future iterations (cf. Fig. 19 for dynamics of $M(\boldsymbol{\omega})$).

The results from Lemmas 1 and 4 highlight the significance of factor misalignment for both the landscape of loss functions under DWF and the trajectories of SGD optimization. For balanced factorizations, the usual smooth $L_2$ penalty remarkably takes the equivalent form of a sparsity-inducing regularizer, with SGD dynamics being restricted to simpler symmetry-induced subspaces. Notably, both $L_2$ regularization and SGD noise naturally drive the dynamics towards balance (Chen et al., 2024). These observations motivate the following key result:

**Theorem 1** (Equivalence of optimization problems). *The optimization problems*

$$\min_{\mathbf{w} \in \mathbb{R}^p} \mathcal{L}_{\mathbf{w}, \lambda}(\mathbf{w}) := \frac{1}{n} \sum_{i=1}^{n} \ell\left(y_i, f_{\mathbf{w}}\left(\mathbf{w}, \boldsymbol{x}_i\right)\right) + \lambda \|\mathbf{w}\|_{2/D}^{2/D} \tag{4}$$

$$\min_{\boldsymbol{\omega} \in \mathbb{R}^{Dp}} \mathcal{L}_{\boldsymbol{\omega}, \lambda}(\boldsymbol{\omega}) := \frac{1}{n} \sum_{i=1}^{n} \ell\left(y_i, f_{\mathbf{w}}\left(\boldsymbol{\omega}_1 \odot \ldots \odot \boldsymbol{\omega}_D, \boldsymbol{x}_i\right)\right) + \frac{\lambda}{D} \sum_{d=1}^{D} \|\boldsymbol{\omega}_d\|_2^2 \tag{5}$$

*have the same global and local minima with the respective minimizers related as $\hat{\mathbf{w}} = \hat{\boldsymbol{\omega}}_1 \odot \ldots \odot \hat{\boldsymbol{\omega}}_D$.*

Practically speaking, instead of attempting to optimize the non-smooth problem in Eq. (4), we can alternatively optimize the smooth problem in Eq. (5) as every local or global solution of the DWF model will yield a corresponding local or global solution in the original model space. Hence, this allows inducing sparsity in typical deep learning applications with SGD-optimization by a simple $L_2$ regularization using Eq. (5). In contrast, the non-differentiability in (4) will cause the optimization to oscillate and not provide the desired sparsity (see Section 5.1 and Fig. 1). The $L_{2/D}$ penalty in Eq. (4) becomes non-convex and increasingly closer to the $L_0$ penalty for $D > 2$, permitting a more aggressive penalization of small weights than $L_1$ regularization (Frank & Friedman, 1993).

While the theoretical equivalence derived in Theorem 1 establishes correspondence of all minimizers and suggests a simple way to induce sparsity in arbitrary neural networks, the optimization of a DWF model is not straightforward and little is known about the learning dynamics of such a model. We will hence study these two aspects in the following section.

## 4 OPTIMIZATION AND DYNAMICS OF DEEP FACTORIZED NETWORKS

Two crucial aspects of successfully training DWF models are their initialization and the learning rate when optimizing with SGD.

### 4.1 INITIALIZATION

Applying DWF to a neural network factorizes each parameter into a product of $D$ factors. However, initializations for factorized neural networks are not straightforward since the product distribution of random variables often drastically differs from the factor distribution, leading to pathological behavior, especially for deep factorizations. To retain the properties of standard initializations, defined

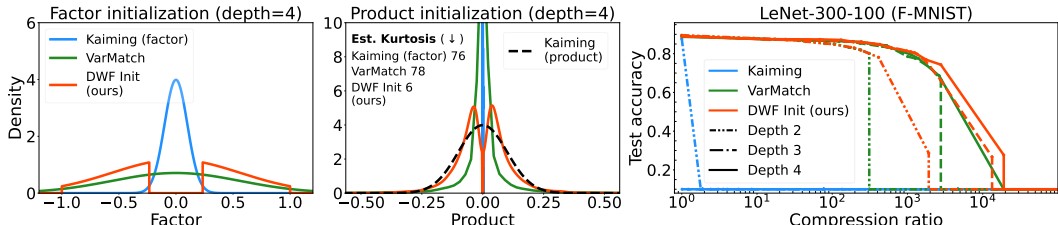

Figure 4: DWF initialization strategies. **Left**: factor densities with variance matching and truncation. **Middle**: product densities for $D = 4$ illustrating kurtosis explosion without truncation. **Right**: sparsity-accuracy curves for different initializations and $D$, showing the failure of standard initialization.

in Appendix C, adjustments to the factor initializations need to be implemented.

Our exposition focuses on the simplest case $w_j^{(l)} \sim \mathcal{N}(0, 1/n_{\text{in}}^{(l)})$, where $n_{\text{in}}^{(l)}$ is the number of input units to the $l$-th layer (LeCun et al., 2002), but similar arguments can be made for other approaches. While Ziyin & Wang (2023) use standard initialization for shallow factorizations with good results, this consistently fails for $D > 2$ in our experiments and only works in few cases for $D = 2$ (cf. Figs. 4 and 12b). The following result shows that initializing a factorized neural network using a standard scheme leads to deteriorating initialization quality and vanishing activation variance:

**Lemma 2** (Standard initializations in factorized networks). *Consider a factorized neural network with $L$ layers and factorization depth $D \geq 2$, where $\mathbf{w}^{(l)} = \boldsymbol{\omega}_1^{(l)} \odot \ldots \odot \boldsymbol{\omega}_D^{(l)}$ and the scalar factors $\omega_{j,d}^{(l)}$ are initialized using a standard scheme. Then **i)** the collapsed weights $\varpi_j^{(l)} = \prod_{d=1}^{D} \omega_{j,d}^{(l)} \xrightarrow{p} 0$ as $D$ grows, and **ii)** for any $D \geq 2$, the variance of the activations vanishes in both $n_{in}$ and $L$.*

**Rectifying the failure of standard initializations in DWF**  Given a standard initialization $w \sim \mathcal{P}(w)$ with variance $\sigma_w^2$, the first step is to correct the variance of the product $\varpi$ by initializing the factors $\omega_d$ so that the variance of their product matches that of $\mathcal{P}(w)$. This variance matching of $\varpi$ and $w$ is achieved by setting $\text{Var}(\omega_d) = \text{Var}(w)^{1/D}$ and named *VarMatch* initialization here. However, only considering the variance overlooks the importance of higher-order moments for initialization in deep learning. For example, given a factor initialization $\omega_d \sim \mathcal{N}(0, \sigma^2)$, we have $\mathbb{E}\big[(\varpi)^2\big] = \sigma^{2D}$, $\mathbb{E}\big[(\varpi)^4\big] = 3^D \sigma^{4D}$, implying the kurtosis of $\varpi$ grows exponentially as $\kappa_\varpi = 3^D$ regardless of variance matching (cf. Fig. 4). In DWF with plain variance matching, we observe a performance decline and the undesirable emergence of inactive weights (cf. Fig. 5a).[1]

Since variance matching alone does not yield satisfactory results for $D > 2$, we additionally propose a tailored interval truncation of the factor initialization outside of a certain absolute value range and name this approach `DWF` initialization (see also Algorithm 1). This redistributes the accumulating probability mass away from 0 and prevents catastrophic initialization of dead weights. The truncation thresholds control the smallest and largest possible absolute values $\varpi_{\min}$ and $\varpi_{\max}$ of $\varpi$, defining the support of the product distribution. Setting the upper truncation threshold to $(2\sigma_{\mathbf{w}})^{1/D}$ to address large outliers and the lower threshold to $\varepsilon^{1/D}$, for some $\varepsilon > 0$, successfully removes pathological product initializations in our experiments.

Together, the crucial ingredients for `DWF` initialization are corrections for both the vanishing variance of the product distribution and its concentration around zero.

**Remark 1.** The factorized bias parameters should not be initialized to all zeros, as this corresponds to a saddle point from which gradient descent cannot escape by symmetry (see Lemma 4).

## 4.2  LEARNING RATE

Another challenge in optimizing a depth-factorized model is the choice of learning rate (LR). As shown in Fig. 5b, if the LR is chosen too small, the model cannot learn a sparse representation despite achieving the same generalization as a $99.4\%$ sparse model trained with large LR. This closely follows previous analyses of large LRs in neural network training dynamics: Nacson et al.

---

[1]Inactive or dead weights are collapsed weights $\varpi$ consisting of factors $\omega_d$ with vanishingly small initialization values, resulting in $\varpi$ not changing during training.

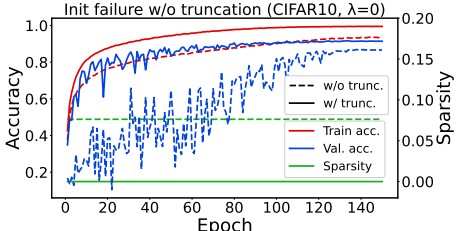 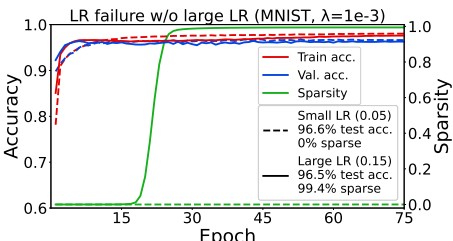

(a) Acc. degradation and dead weights w/o truncation despite $\lambda = 0$ for ResNet-18 and $D = 10$.

(b) Sparsity only emerges with sufficiently large LR for LeNet-300-100 and $D = 4$.

Figure 5: Failure modes when optimizing factorized neural networks.

(2022) show that large LRs help transition to a sparsity-inducing regime in diagonal linear networks. In more realistic scenarios, Andriushchenko et al. (2023) observe that a piece-wise constant (step decay) LR schedule with large initial LR induces phased learning dynamics including a short initial learning phase, followed by a period of sparse feature learning and loss stabilization, and sudden generalization upon reduction of the LR. Particular to symmetries and SGD, Chen et al. (2024) demonstrate how large LRs help generalization by causing SGD to be attracted to symmetry-induced structures through stochastic collapse. We conjecture that in DWF, the introduction of $D$-fold artificial symmetries (cf. Definition 1) accelerates this phenomenon and thus additionally aids sparse learning. The first row of Fig. 6 shows the training dynamics of deep factorized ResNets and demonstrates the requirement of large and small LR phases for DWF training. These training dynamics are further discussed in the following section. Additionally, Appendix E includes an ablation study on different LRs and factorization depths $D$, suggesting optimal sparsity-accuracy tradeoffs for initial LRs slightly below a critical threshold where training becomes unstable (Fig. 11).

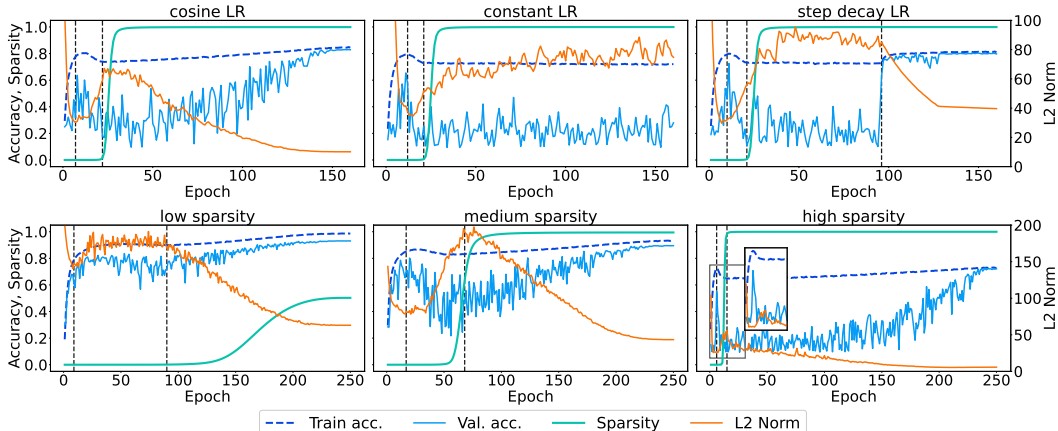

Figure 6: Factorized ResNet-18 on CIFAR10 with $D = 4$. Dashed lines indicate phase transitions. **Top**: Different LR schedules with same initial LR and $\lambda$. **Left**: cosine LR learns sparse and generalizing solutions. **Mid**: a const. large LR causes sparsification but no generalization. **Right**: step-decay LR displays sharply distinct sparsification and generalization phases in large and small LR phases. **Bottom**: For cosine LR, the three distinct learning phases occur at all sparsity levels, with sharper contracted dynamics for high sparsity.

### 4.3 LEARNING DYNAMICS AND DELAYED GENERALIZATION

The learning dynamics of DWF with cosine annealing exhibit three distinct phases, characterized by changes in accuracy, sparsity, and $L_2$ norm of the collapsed weights (Fig. 6, second row): In an **initial phase**, SGD learns easy-to-fit patterns without overfitting while the $L_2$ norm decreases. The **reorganization phase** is characterized by temporary drops in accuracy and an increase in weight norm, hinting at a period of representational restructuring to accommodate sparsity constraints. Sparsity emerges during or at the end of this phase. The final **mixed sparsification and generalization phase** shows improvements in training and validation accuracy as sparsification continues at a decreasing rate. The mixed nature of the final phase, contrasting sharply separated sparsification

and generalization with step decay, is owed to the gradual reduction in cosine annealing. Notably, with increasing regularization $\lambda$, the dynamics contract, and the phases occur in closer succession. This phased behavior shows that the more contracted the reorganization phase is, the higher compression and the more severe delayed generalization will be. This is reminiscent of the "grokking" phenomenon (Power et al., 2022) shown to be tightly linked to $L_2$ regularization (Liu et al., 2023).

## 4.4 IMPACT OF REGULARIZATION AND EVOLUTION OF LAYER-WISE METRICS

To investigate dynamics in more detail, we analyze the effect of $D$ and $\lambda$ on the sparsity and training trajectories (Fig. 7). Similar results for different architectures/datasets are included in Appendix F.

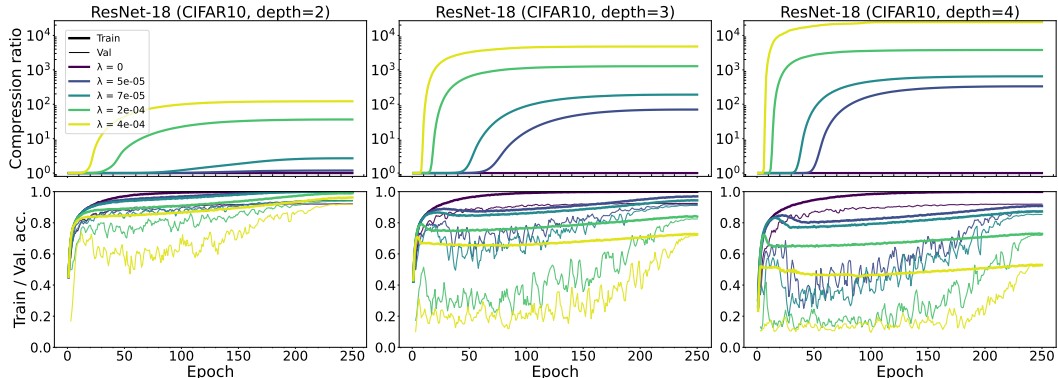

Figure 7: Impact of regularization $\lambda$ on compression (**top**), training, and validation accuracy (**bottom**) for factorized ResNet-18 and $D \in \{2, 3, 4\}$. For large $\lambda$ severely delayed generalization and extreme compression emerges simultaneously. Colors indicate the same $\lambda$ in both rows.

As expected, increasing $\lambda$ leads to higher compression ratios across all depths. Moreover, greater $D$ enables higher compression ratios for the same $\lambda$. During the initial phase, the regularized training curves coincide with the unregularized trajectory until their departure at the onset of the reorganization phase. This departure occurs earlier the stronger the regularization. For greater factorization depths, the same $\lambda$ values induce higher sparsity at the cost of reduced generalization performance, indicating a stronger regularizing effect[2]. The relationship between sparsity, $\lambda$, and the collapsed weight norm is further discussed in Appendix E.3. Appendix F.3 presents the layer-wise evolution of sparsity and weight norms, providing more detailed insights into the effects of DWF across the network topology for different architectures (e.g., Fig. 17a). Two key observations can be made: the first and last layers exhibit less sparsity, owed to their increased importance for the prediction. For the intermediate layers, there is a general trend toward higher compression for later layers. Secondly, the non-monotonic dynamics of the collapsed weight norm seem to be almost entirely driven by the first few and the last layers, while the intermediate layers behave homogeneously. Finally, the evolution of factor misalignment and its relation to the onset of sparsity is discussed in Appendix F.4.

## 5 PERFORMANCE EVALUATION

In this section, we evaluate the performance of DWF. In Appendices F and G, we provide further results and details on the experimental setup, including hyperparameters and training protocols.

## 5.1 FAILURE OF VANILLA $L_1$ OPTIMIZATION WITH SGD

The failure of SGD with vanilla $L_1$ regularization to achieve inherent sparsity has been previously observed by Ziyin & Wang (2023); Kolb et al. (2023). It is natural to ask whether this limitation is merely a benign optimization artifact or if it degrades the prunability of the regularized models. We, therefore, train a LeNet-300-100 on Fashion-MNIST with vanilla $L_1$ regularization as well as with DWF and $D = 2, 3$, inducing differentiable $L_1$ and non-convex $L_{2/3}$ regularization.

---

[2]Note that this does not imply worse performance in general, but a different optimal $\lambda$ for different $D$.

**Results** The left plot in Fig. 1 (page 1) shows the tradeoff between performance and inherent sparsity (before pruning) for 100 logarithmically spaced $\lambda$ values, confirming prior findings on the limitations of vanilla $L_1$ optimization. In contrast, differentiable $L_1$ regularization using DWF achieves a compression ratio of about 350 at $80\%$ test accuracy. In addition, our DWF network with $D = 3$ is up to four times sparser than $D = 2$ at the same accuracy, underscoring the advantages of deeper factorizations. In the right plot of Fig. 1, we subsequently apply post-hoc magnitude pruning to each of the models at increasing compression ratios (without fine-tuning) until reaching random chance performance and use the best-performing pruned model at each fixed compression ratio to obtain the pruning curves. Results indicate that differentiable sparse training with factorized networks provides better tradeoffs than vanilla $L_1$, even after accounting for the issues producing sparsity when using SGD with vanilla $L_1$. At $80\%$ test accuracy, vanilla $L_1$ plus pruning requires twice as many parameters as its DWF counterpart, and three times as many as DWF ($D = 3$). This suggests SGD with $L_1$ struggles to find similarly well-prunable structures, while DWF yields much sparser models.

## 5.2 RUN TIMES

The perceptive reader might be concerned about the computational overhead induced by training deep factorized networks. Our experiments show this concern to be unwarranted, as the effect of the factorization depth is rather unimportant compared to batch size for both time per sample and memory cost. Appendix I.2 illustrates this for WRN-16-8 (Zagoruyko & Komodakis, 2016) and VGG-19 (Simonyan & Zisserman, 2014). For both models, the impact of factorization depth on computation time and memory usage becomes negligible as batch size increases. These findings suggest that practitioners can leverage deeper factorized networks without incurring substantial additional computational costs, particularly at typical batch sizes used in modern deep learning.

## 5.3 COMPRESSION BENCHMARK

We now evaluate DWF for factorization depths $D \in \{2, 3, 4\}$ against various pruning methods concerning test accuracy vs. compression, as well as the layer-wise allocation of the remaining weights. Architectures and datasets: Our experiments cover commonly used computer vision benchmarks: LeNet-300-100 and LeNet-5 (LeCun et al., 1998) on MNIST, Fashion-MNIST, and Kuzushiji-MNIST, VGG-16 and VGG-19 (Simonyan & Zisserman, 2014) on CIFAR10 and CIFAR100, and ResNet-18 (He et al., 2016) on CIFAR10 and Tiny ImageNet.
Methods: We compare our method against Global magnitude pruning (**GMP**) after training (Han et al., 2015), a simple pruning method that removes the smallest weights across all layers and is surprisingly competitive, especially at low sparsities (Gale et al., 2019; Frankle et al., 2020); Single-shot Network Pruning (**SNIP**) (Lee et al., 2019), a pruning-at-initialization technique, showing competitive performance against other recent pruning methods (Wang et al., 2020); **SynFlow** (Tanaka et al., 2020), considered a state-of-the-art method for high sparsity regimes; **Random pruning**, serving as a naive baseline that removes weights uniformly at random; a shallow factorized network ($D = 2$) which is our variant of the **spred** algorithm (Ziyin & Wang, 2023) with our tailored initialization.
Tuning: For comparison methods, we use the established training configurations in Lee et al. (2019); Wang et al. (2020); Frankle et al. (2020) when available, and otherwise ensure comparability by using the same configuration for all methods. All models are trained with SGD and cosine learning rate annealing (Loshchilov & Hutter, 2022). For our method, **no** post-hoc pruning or fine-tuning is required and all layers are regularized equally. Further details are given in Appendix G.

**Results** Fig. 8 shows results for the fully-connected and convolutional LeNet-300-100 and LeNet-5 architectures on MNIST, F-MNIST, and K-MNIST. Across all datasets and models, our proposed DWF consistently outperforms existing pruning techniques, particularly at higher compression ratios. For LeNet-300-100 on MNIST, DWF with $D = 3, 4$ remains within $5\%$ of the dense performance even above a compression ratio of $500$, significantly surpassing other methods. On F-MNIST and K-MNIST, a similar performance gain is observed. LeNet-5 exhibits similar trends, with DWF maintaining high accuracy at compression ratios where other techniques, especially random pruning and SNIP, have collapsed. Notably, DWF sustains performance up to a compression ratio of $100$ on K-MNIST, while competitors rapidly decline. SynFlow and GMP generally outperform random pruning and SNIP, but still fall short of DWF. When comparing shallow and deep factorizations, we see that $D > 2$ retains performance and delays model collapse much longer than $D = 2$. The

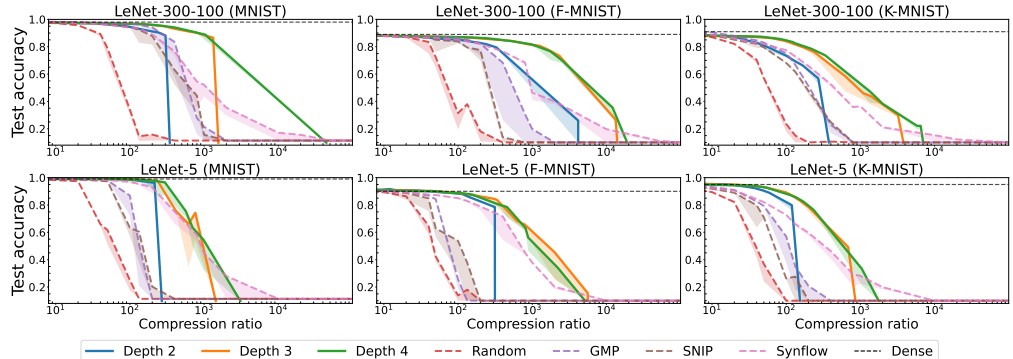

Figure 8: Accuracy vs. sparsity tradeoffs for LeNet architectures on MNIST and replacements of varying difficulty. Lines depict median test accuracies and shaded areas the minimum over three random initializations.

results demonstrate that DWF offers substantial gains in compression capability. Further, the clear separation between DWF and other methods in the high-sparsity regime indicates that our approach captures aspects of the model's representational power that are missed by the other techniques. These findings underscore the potential of DWF, particularly under severe parameter constraints.

For larger architectures and more complex datasets (Fig. 9), DWF continues to demonstrate superior performance, albeit less pronounced. We observe that the $D = 2$ factorization excels in the medium sparsity regime below a compression ratio of 100, while $D > 2$ shows enhanced resilience to performance degradation and delayed model collapse at more extreme sparsity levels.

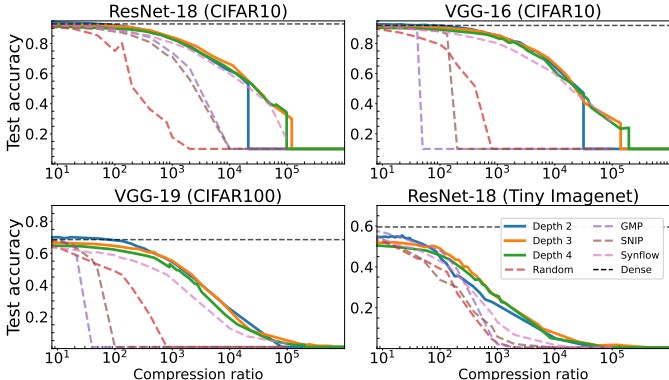

Figure 9: Accuracy vs. sparsity for larger ResNet and VGG architectures on CIFAR and Tiny ImageNet.

Table 1 showcases the sparsest models achieved by each method while maintaining performance within $5\%$ or $10\%$ of the dense model accuracy. These tolerance levels are suitable for testing the medium to high sparsity regimes we focus on. Other levels can be read from Fig. 9. Of the 10 presented scenarios, DWF with $D = 3, 4$ achieves the highest compression in 9 cases, demonstrating the robustness of DWF in preserving model performance under extreme sparsity requirements. The best-ranked DWF model achieves 2 to 5 times the compression ratio of the best pruning method, and surpasses shallow factorization in all but one setting, albeit with smaller improvements ranging from $8\%$ to $298\%$. For example, DWF with $D = 3$ reaches a compression ratio of $1014$ ($1456$) for VGG-19 on CIFAR100 (ResNet-18 on CIFAR10) at $10\%$ tolerance, improving by $8\%$ ($25\%$) over $D = 2$ and by $381\%$ ($102\%$) over the best pruning method SynFlow.

**Allocation of layer-wise sparsity** Finally, we investigate the reasons for model collapse in SNIP and GMP in the high sparsity regime by plotting the layer-wise remaining ratio (1/CR) for ResNet-18 and VGG-16 on CIFAR10 in the medium and extreme compression regimes, as shown in Fig. 10. At high compression, for both ResNet-18 and VGG-16, we observe that GMP and SNIP catastrophically prune entire layers. In contrast, SynFlow and DWF automatically learn adaptive layer-wise sparsity budgets which helps in avoiding such issues. While SynFlow does prune some layers entirely in ResNet-18, these correspond to skip connections that can be removed without interrupting

Table 1: Compression ratios (sparsities) of sparsest model within $\varepsilon_{acc}$ percentage points of the dense model test accuracy. Random pruning is left out for clarity.

| CR (↑) | $\varepsilon_{acc}$ | LeNet-300-100 | LeNet-5 | ResNet-18 | VGG-19 | ResNet-18 |
|---|---|---|---|---|---|---|
| | | F-MNIST | K-MNIST | CIFAR10 | CIFAR100 | Tiny ImageNet |
| Depth 2 | 5% | 141 (99.29%) | 52 (98.08%) | 466 (99.79%) | **484** (99.79%) | 60 (98.34%) |
| | 10% | 362 (99.72%) | 78 (98.71%) | 1169 (99.91%) | 939 (99.89%) | 99 (98.99%) |
| Depth 3 | 5% | **506** (99.80%) | **75** (98.67%) | **573** (99.83%) | 440 (99.77%) | **67** (98.51%) |
| | 10% | 1422 (99.93%) | 134 (99.25%) | **1456** (99.93%) | **1014** (99.90%) | **161** (99.38%) |
| Depth 4 | 5% | 486 (99.79%) | **75** (98.67%) | 445 (99.78%) | 215 (99.53%) | 13 (92.39%) |
| | 10% | **1442** (99.93%) | **139** (99.28%) | 1161 (99.91%) | 675 (99.85%) | 113 (99.12%) |
| GMP | 5% | 156 (99.36%) | 22 (95.37%) | 211 (99.53%) | 37 (97.27%) | 60 (98.33%) |
| | 10% | 235 (99.58%) | 32 (96.84%) | 484 (99.79%) | 68 (98.52%) | 133 (99.25%) |
| SNIP | 5% | 76 (98.69%) | 17 (94.10%) | 140 (99.29%) | 28 (96.47%) | 18 (94.34%) |
| | 10% | 146 (99.32%) | 24 (95.91%) | 339 (99.70%) | 42 (97.59%) | 41 (97.56%) |
| Synflow | 5% | 141 (99.29%) | 21 (95.23%) | 210 (99.52%) | 46 (97.81%) | 24 (95.84%) |
| | 10% | 302 (99.67%) | 37 (97.30%) | 721 (99.86%) | 218 (99.54%) | 71 (98.60%) |

the synaptic flow. Comparing SynFlow and DWF, we observe that DWF produces higher sparsity in the first and last layers across all settings. This is a particularly desirable property, as it leads to greater computational savings for a given overall sparsity level. Moreover, as opposed to removing the skip connections, DWF allocates less sparsity to these structures, suggesting a qualitatively distinct underlying structure optimization mechanism.

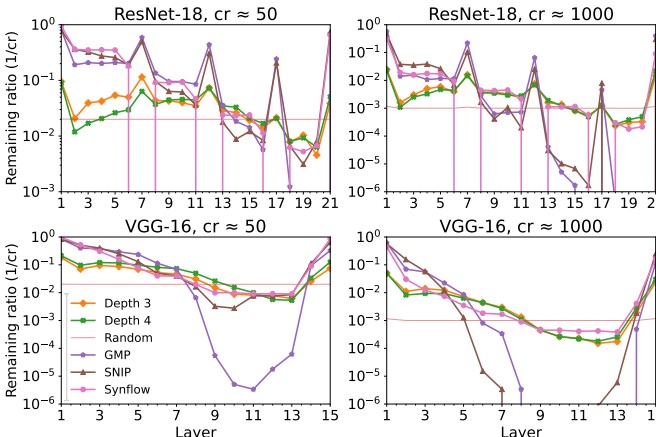

Figure 10: Allocation of layer-wise sparsity for different methods. SNIP and GMP show catastrophic pruning of whole layers (collapse) for high sparsities, whereas DWF, like SynFlow, finds adaptive sparsity allocations.

## 6 CONCLUSION

This paper introduces deep weight factorization (DWF), an extension of a previously proposed differentiable $L_1$ regularization to induce sparsity in general neural networks. By factorizing weights not only into two, but $D \geq 3$ parts, our method provably induces differentiable $L_{2/D}$ regularization that can be incorporated in any neural network. We identify practical training obstacles and propose tailored optimization strategies such as a depth-specific initialization and a sparsity-promoting learning rate scheme. We also characterize three distinct phases that describe the learning dynamics and (delayed) generalization behavior of DWF. Experiments demonstrate that DWF is usually superior to shallow factorization and outperforms dominant pruning techniques.

**Limitations and future work** In this work, we primarily focused on $D \in \{2, 3, 4\}$. While not incurring significant computational overhead (cf. Fig. 26), we found that increasing the factorization depth beyond four did not yield further sparsity improvements and introduced optimization challenges (cf. Fig. 14). Additionally, a limitation of our work is that we exclusively focused on DWF approaches resulting in unstructured sparsity regularization. Hence, an interesting potential direction for future research is to extend our factorization to structured sparsity problems.

ACKNOWLEDGMENTS

DR's research is funded by the Deutsche Forschungsgemeinschaft (DFG, German Research Foundation) – 548823575.

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

# Appendices

## A  FURTHER RELATED LITERATURE

**Convex and Non-convex sparse regularization**  Convex and non-convex regularization for sparsity-inducing effects has a long history as it is well-understood with strong theoretical underpinnings. Notable works in this direction include Tibshirani (1996); Fan & Li (2001); Meinshausen & Bühlmann (2006); Zhang & Huang (2008) for convex norm-based sparse regularization and Friedman et al. (2010); Fan & Li (2001); Zhang (2010); Xu et al. (2010) studying its non-convex extensions. While mathematically rigorous, they see limited application in deep learning due to their non-differentiability exactly where sparsity is achieved, requiring the implementation of non-smooth optimization routines that are often inflexible and do not scale well. While some works exist on proximal-type optimization routines in deep learning (Yang et al., 2020; Deleu & Bengio, 2021), their popularity remains far behind pruning and other approaches, with these methods being rarely applied, used as comparisons, or given much attention in surveys (Hoefler et al., 2021; Gale et al., 2019). Nevertheless, the use of $L_1$ or structured $L_{2,1}$ regularization is ubiquitous in sparse deep learning (Han et al., 2015; He et al., 2017; Liu et al., 2017; Li et al., 2022). These sparsity regularization pruning methods (Cheng et al., 2024) use sparse regularization to shrink weights before applying a subsequent pruning step for actual sparsity. However, the conceptual framework underlying these heuristic-based methods remains poorly understood.

**Weight factorization without and with explicit regularization**  Weight factorizations, also studied under the names diagonal linear networks (Woodworth et al., 2020), redundant parameterization (Ziyin & Wang, 2023), or Hadamard product parameterization (Hoff, 2017; Tibshirani, 2021; Kolb et al., 2023) in various contexts, can be traced back to Grandvalet (1998) and was rediscovered both in statistics (Hoff, 2017) and machine learning Neyshabur et al. (2015). Later, Tibshirani (2021); Kolb et al. (2023) and Ziyin & Wang (2023) used this approach to induce sparsity via $L_2$ regularization. Further works using weight factorization from the field of optimization include Poon & Peyré (2021; 2023). Ouyang et al. (2025) show that the Kurdyka-Łojasiewicz exponent at a second-order stationary point of a factorized and $L_2$ regularized objective can be inferred from its corresponding $L_1$ penalized counterpart. These works are closest to ours in spirit, namely, factorizing the weights of existing problems to achieve sparsity in the original weight space under $L_2$ regularization. None of these works, however, studied deeper factorizations of neural network parameters. A closely related approach to incorporate the induced non-convex $L_q$ regularization ($q < 1$) into DNN training was recently proposed by Outmezguine & Levi (2024), who base their method on the $\eta$-trick (Bach et al., 2012) instead of the $L_2$ regularized weight factorization we study. This method re-parametrizes the regularization function instead of factorizing the network parameters, utilizing a different variational formulation of the $L_q$ quasi-norm. Despite having the same effective $L_q$ regularization, DWF incorporates additional symmetry-induced sparsity-promoting effects through weight factorization and the stochastic collapse phenomenon (Ziyin, 2023; Chen et al., 2024). Another branch of literature studies the *representation cost* of DNN architectures $f_{\mathbf{w}}$ for a specific function $f$, defined as $R(f) = \min_{\mathbf{w}:f_{\mathbf{w}}=f} \|\mathbf{w}\|_2^2$, showing that the $L_2$ cost of representing a linear function using a diagonal linear network yields the $L_{2/D}$ quasi-norm (Dai et al., 2021; Jacot, 2023).

Apart from the previous works using explicit regularization, the implicit regularization effect of weight factorization was studied by various researchers, both in statistics and deep learning, including Neyshabur et al. (2015); Gunasekar et al. (2018); Gissin et al. (2019); Vaskevicius et al. (2019); Woodworth et al. (2020); Pesme et al. (2021); Zhao et al. (2022); Li et al. (2021). However, such approaches are usually impractical for real applications, requiring vanishing initializations or specific loss functions. For example, the implicit bias of deep weight factorizations does not extend to non-convex regularizers under the squared loss (Nacson et al., 2022). Powerpropagation (Schwarz et al., 2021) is a different sparsity-inducing weight transformation with implicit regularization effects, proposed as a modular tool designed for use in practical applications.

Various papers have also studied the factorization of weights without explicit regularization in the context of implicit acceleration caused by the factors acting as adaptive learning rates. Notable examples include Arora et al. (2019); Wang et al. (2022); Li & Lin (2024).

**Pruning in neural networks**  The landscape of pruning and sparse training methods is confusing due to the plethora of complex sparsification pipelines, incorporating numerous techniques at the same time, and an enormous amount of hyperparameters like pruning schedules or learning rates at

different stages. A fair comparison is further complicated by the lack of established, streamlined evaluation processes and opposing objectives in some methods. Pruning, arguably the most popular and widespread method, can be used in innumerable ways to sparsify neural networks, which makes comparisons among these particularly difficult (Wang et al., 2023). Existing pruning techniques use pruning at initialization (Lee et al., 2019; Wang et al., 2020; Tanaka et al., 2020), pruning after training by, e.g., magnitude pruning (Han et al., 2015; Gale et al., 2019), pruning during training, iterative pruning (Frankle et al., 2020), or pruning and re-growth (Evci et al., 2020). Recent surveys can be found in Blalock et al. (2020); Hoefler et al. (2021); Cheng et al. (2024). Prominent examples include the lottery ticket hypothesis (Frankle & Carbin, 2019), proposing that many networks contain equally performant, but much smaller subnetworks that can be found at initialization. A Bayesian pruning version is suggested in Dhahri et al. (2024). Another approach related to pruning is soft thresholding reparameterization (Kusupati et al., 2020), a sparse training method incorporating a soft-thresholding step into its network. Despite its success, however, it was shown to be outperformed by the differentiable $L_1$ approach, i.e., DWF with $D = 2$ (Ziyin & Wang, 2023). Recently, Zhang et al. (2024) established theoretical bounds on network prunability using convex geometry, showing that the fundamental one-shot pruning limit without sacrificing performance is determined by weight magnitudes and the sharpness of the loss landscape, providing a unified framework to help explain the effectiveness of magnitude-based pruning. They empirically show that $L_1$ regularized post-hoc magnitude pruning approximately matches the derived pruning limit before performance degrades significantly.

**Sparsity-inducing regularization in neural networks**   Apart from pruning, the application of norm-based regularizers is also common in deep learning (Scardapane et al., 2017; Wen et al., 2016; Han et al., 2015; Bui et al., 2021). This includes $L_0$-type reguarlization methods (Louizos et al., 2018; Zhou et al., 2021; Savarese et al., 2020). However, some of these, such as Louizos et al. (2018), were found to not work well due to the stochastic sampling in their training procedure. Other approaches include adaptive regularization (Glandorf et al., 2023) or dynamic masking (Junjie et al., 2020).

**Sparsity based on structural constraints**   While we focus on unstructured sparsity in this paper, various approaches for structured pruning were proposed, including Wen et al. (2016); Li et al. (2022); Bui et al. (2021); Liu et al. (2017). For a recent survey, see He & Xiao (2023). A link also made in our paper is the connection between sparsity and symmetries. Using structures of symmetries can guide the sparsification of neural networks. Papers studying this link include the works of Kunin et al. (2020); Simsek et al. (2021); Le & Jegelka (2022), but also some various recent work such Ziyin (2023); Ziyin et al. (2023); Chen et al. (2024).

**Matrix factorization related induced regularization**   In Srebro et al. (2004); Mazumder et al. (2010); Shang et al. (2020); Hastie et al. (2015) different matrix factorization regularization schemes are proposed to, e.g., learn incomplete matrices (Mazumder et al., 2010; Hastie et al., 2015) or for better generalization (Srebro et al., 2004). There are also neural architectures implementing flavors of matrix factorization to achieve better performance or acceleration (Guo et al., 2020; Jing et al., 2020; Bhardwaj et al., 2022). Although not directly related to our factorization, we will briefly explain their idea to contrast it with our approach. For example, Guo et al. (2020) note a beneficial effect of applying $L_2$ regularization on the (matrix) factors in the form of $\|\boldsymbol{W}_1\|_2^2 + \|\boldsymbol{W}_2\|_2^2$ as opposed to the $L_2$ regularizer $\|\boldsymbol{W}_1\boldsymbol{W}_2\|_2^2$ proposed in Arora et al. (2019). This observation can be explained by the low-rank bias induced on the product matrix, whereas the second approach is simple $L_2$ regularization on the product.

# B   INTUITION FOR SPARSITY VIA $L_2$ REGULARIZED WEIGHT FACTORIZATION

Deep Weight Factorization introduces overparameterization by decomposing each original weight w multiplicatively into $D \geq 2$ factors $\omega_1, \ldots, \omega_D$. Without additional $L_2$ regularization, this induces artificial rescaling symmetries (Definition 1), resulting in infinitely many possible factorizations for each weight, all producing the same collapsed network and thus leaving the loss function unchanged. However, when $L_2$ regularization is applied, it influences the choice among these factorizations by preferring those with minimal Euclidean norm. With $L_2$ regularization, only minimum-norm (bal-

anced) factorizations can be optimal, as otherwise, we could always decrease the $L_2$ penalty by picking a more balanced factorization while leaving the unregularized loss unchanged (cf. Lemma 1).

To provide some geometric intuition using a more concrete example, consider the simplest case of factorizing a scalar weight $w \in \mathbb{R}$ into two factors, $w = \omega_1 \cdot \omega_2$, as illustrated in Fig. 3 for $w \in \{0, 0.25\}$. The set of all possible factorizations $\{(\omega_1, \omega_2) \in \mathbb{R}^2 : \omega_1 \omega_2 = w\}$ is given by the points on the coordinate axes for $w = 0$ and forms a rectangular hyperbola in the $(\omega_1, \omega_2)$ plane for non-zero $w$ (cf. Fig. 3). Among these, the factorizations with minimal $L_2$ norm (i.e., minimal distance to the origin) are located at the vertices of the hyperbola. These minimum-norm factorizations are balanced, meaning the factors are equal in magnitude, as the vertices of a rectangular hyperbola always lie either on the diagonal $\omega_2 = \omega_1$ or $\omega_2 = -\omega_1$. Specifically, the two vertices of the resulting hyperbola are given by $(\sqrt{|w|}, \sqrt{|w|})$ and $(-\sqrt{|w|}, -\sqrt{|w|})$ for positive $w$, and $(\sqrt{|w|}, -\sqrt{|w|})$ and $(-\sqrt{|w|}, \sqrt{|w|})$ for negative $w$. Combined with the case $w = 0$, the minimum-norm factorizations $(\omega_1^*, \omega_2^*)$ for any $w$ are obtained as

$$(\omega_1^*, \omega_2^*) = \begin{cases} \left(\sqrt{|w|}, \sqrt{|w|}\right) \text{ or } \left(-\sqrt{|w|}, -\sqrt{|w|}\right) & , w > 0 \\ (0, 0) & , w = 0 \\ \left(\sqrt{|w|}, -\sqrt{|w|}\right) \text{ or } \left(-\sqrt{|w|}, \sqrt{|w|}\right) & , w < 0. \end{cases} \tag{6}$$

At these points, the $L_2$ penalty evaluates to $2|w|$, effectively turning into an $L_1$ penalty on the collapsed weight $w$ scaled by a factor of 2.

For deeper factorizations involving more than two factors the same line of reasoning applies, but visualizing the set of possible factorizations as in Fig. 3 for $D = 2$ becomes challenging. The minimum $L_2$ penalty at balanced factorizations reduces to a non-convex sparsity-inducing $L_{2/D}$ penalty on the collapsed weight. This serves as a lower bound of the $L_2$ penalty for every fixed value of $w$. Once the factors reach this balanced state, which is an "absorbing state" under (S)GD, the optimization process locks in this configuration for all future iterations by symmetry (cf. Lemma 4). Thus, the combination of DWF and $L_2$ regularization induces sparsity in the collapsed weights by promoting balanced factorizations, at which the $L_2$ penalty reduces to a lower-degree quasi-norm penalty on $w$.

## C    FURTHER RESULTS AND MISSING PROOFS

### C.1    PROOF OF LEMMA 1

*Proof.* Let $\boldsymbol{\omega} = (\boldsymbol{\omega}_1, \ldots, \boldsymbol{\omega}_D) \in \mathbb{R}^{Dp}$ be a local minimizer of $\mathcal{L}_{\boldsymbol{\omega},\lambda}(\boldsymbol{\omega})$. As the factorization is applied independently to each parameter, it suffices to treat the scalar case: We will prove that $|\omega_{j,1}| = \ldots = |\omega_{j,D}|$ for all $j \in [p]$.

The rescaling symmetries of DWF ensures that $\mathcal{L}_{\boldsymbol{\omega},0}$ (the factorized loss without regularization) is constant over all possible factorizations of a collapsed parameter $\boldsymbol{\varpi}$. However, the $L_2$ regularization term $\lambda D^{-1} \sum_{d=1}^{D} \|\boldsymbol{\omega}_d\|_2^2$ enforces a preference for min-norm factorization. For each scalar weight indexed by $j \in [p]$, consider its factors $\omega_{j,1}, \ldots, \omega_{j,D}$. Applying the AM-GM inequality to the $L_2$ penalty of the DWF loss yields

$$D^{-1} \sum_{d=1}^{D} \omega_{j,d}^2 \geq \left( \prod_{d=1}^{D} (\omega_{j,d})^2 \right)^{1/D} = |\omega_{j,1} \cdots \omega_{j,D}|^{2/D} = |\varpi_j|^{2/D} \quad \forall j \in [p] \qquad (7)$$

This shows the balancedness requirement for the minimizers of $\mathcal{L}_{\boldsymbol{\omega},\lambda}(\boldsymbol{\omega})$, as the AM-GM inequality holds tight if and only if all terms are equal, i.e., $|\omega_{j,1}| = \ldots = |\omega_{j,D}|$.

Summing over the factorizations of all weights yields the non-convex $L_{2/D}$ regularizer $\|\boldsymbol{\varpi}\|_{2/D}^{2/D}$ as the minimum $L_2$ penalty for a given collapsed weight $\boldsymbol{\varpi} \in \mathbb{R}^p$. $\qquad \square$

### C.2    PROOF OF LEMMA 2

**Definition 3** (Standard Weight Initialization). A standard weight initialization scheme for a neural network layer with $n_{in}$ input units and $n_{out}$ output units is a probability distribution with mean 0 and variance $\sigma^2$, where $\sigma^2 = \frac{cg^2}{n_{mode}}$. Here, $g$ is a gain factor depending on the activation function, $c$ is a constant, and $n_{mode}$ is either $n_{in}$, $n_{out}$ or their sum. Common examples include the Kaiming ($\sigma^2 = \frac{2}{n_{in}}$) (He et al., 2015), Glorot ($\sigma^2 = \frac{2}{n_{in}+n_{out}}$) (Glorot & Bengio, 2010), or LeCun initialization ($\sigma^2 = \frac{1}{n_{in}}$) (LeCun et al., 2002).

*Proof.* Recall that using a standard initialization (cf. Definition 3), each factor is initialized as $\omega_{j,d}^{(l)} \sim \mathcal{N}(0, \sigma_l^2)$, where $\sigma_l^2 = 1/n_{in}^{(l)} < 1$ in the case of LeCun initialization (LeCun et al., 2002). For clarity, we assume the width $n_{in}^{(l)}$ to be constant across layers $l \in [L]$.

To prove the first statement, we note that $\mathbb{E}[\varpi_j^{(l)}] = 0$ and $\text{Var}\left(\varpi_j^{(l)}\right) = \prod_{d=1}^{D} \text{Var}\left(\omega_{j,d}^{(l)}\right) = \sigma^{2D}$. Applying Chebyshev's inequality, we get for any $\varepsilon > 0$

$$\mathbb{P}\big(\big|\varpi_j^{(l)} - \mathbb{E}\big[\varpi_j^{(l)}\big]\big| \geq \varepsilon\big) = \mathbb{P}(|\varpi_j^{(l)}| \geq \varepsilon) \leq \frac{\text{Var}\big(\varpi_j^{(l)}\big)}{\varepsilon^2} = \frac{\sigma^{2D}}{\varepsilon^2}. \qquad (8)$$

Finally, we have $0 \leq \lim_{D\to\infty} \mathbb{P}(|\varpi_j^{(l)}| \geq \varepsilon) \leq \lim_{D\to\infty} \frac{\sigma^{2D}}{\varepsilon^2} = 0$, and thus, by the squeeze theorem: $\lim_{D\to\infty} \mathbb{P}(|\varpi_j^{(l)}| \geq \varepsilon) = 0$. This shows that $\varpi_j^{(l)} \xrightarrow{P} 0$ as $D \to \infty$.

For the second point, we denote the pre-activation of neuron $k$ in layer $l$ as

$$y_k^{(l)} = \sum_{i=1}^{n_{in}} \text{w}_{k_i}^{(l)} \phi\left(y_i^{(l-1)}\right) = \sum_{i=1}^{n_{in}} \left( \prod_{d=1}^{D} \omega_{k_i,d}^{(l)} \right) \phi\left(y_i^{(l-1)}\right), \qquad (9)$$

where $\omega_{k_i,d}^{(l)}$ is the $d$-th scalar factor of the weight $\text{w}_{k_i}^{(l)}$ associated with input $i$ of neuron $k$ in layer $l$. The activation $\phi\left(y_i^{(l-1)}\right)$ is the activation function $\phi$ applied to the pre-activations from layer $l - 1$. To simplify calculations, we assume that the activation function is approximately linear

around the origin, implying $\text{Var}\left(\phi\left(y_i^{(l-1)}\right)\right) \approx \text{Var}\left(y_i^{(l-1)}\right)$ and allowing us to ignore the gain factor, as valid for, e.g., tanh activation. Using that the factors $\omega_{k_i,d}^{(l)}$ and activations $\phi\left(y_i^{(l-1)}\right)$ are independent and identically distributed, respectively, the variance of $y_k^{(l)}$ is given by:

$$\text{Var}\left(y_k^{(l)}\right) = \sum_{i=1}^{n_{\text{in}}} \text{Var}\left(\prod_{d=1}^{D} \omega_{k_i,d}^{(l)} \cdot \phi\left(y_i^{(l-1)}\right)\right) = \sum_{i=1}^{n_{\text{in}}} \text{Var}\left(\phi\left(y_i^{(l-1)}\right)\right) \cdot \prod_{d=1}^{D} \text{Var}\left(\omega_{k_i,d}^{(l)}\right). \quad (10)$$

Since the factors are initialized with $\text{Var}\left(\omega_{k_i,d}^{(l)}\right) = \sigma_l^2 = \frac{1}{n_{\text{in}}}$, the variance of $y_k^{(l)}$ is:

$$\text{Var}\left(y_k^{(l)}\right) = \sum_{i=1}^{n_{\text{in}}} \text{Var}\left(y_i^{(l-1)}\right)\left(\frac{1}{n_{\text{in}}}\right)^D = n_{\text{in}} \cdot \frac{\text{Var}\left(y^{(l-1)}\right)}{n_{\text{in}}^D} = \frac{\text{Var}\left(y^{(l-1)}\right)}{n_{\text{in}}^{D-1}} \quad (11)$$

In non-factorized layers, the $n_{\text{in}}$ in Eq. (11) cancel out, resulting in equal activation variances across layers. In contrast, standard initializations in factorized layers do not account for the exponent $D$ appearing in the variance of the collapsed weight $\varpi_{k_i}^{(l)}$, and thus result in a variance reduction in each subsequent layer as a function of input units and factorization depth $D$. Applying the above relationship recursively, we see that the variance at layer $L$ is

$$\text{Var}(y_k^{(L)}) = \text{Var}(y^{(1)}) \cdot \left(\frac{1}{n_{\text{in}}}\right)^{(D-1)(L-1)} \quad (12)$$

To avoid reducing or amplifying the magnitudes of input signals exponentially, a proper initialization requires $\text{Var}(y_k^{(L)})$ to equal some constant, typically set to unity (He et al., 2015). In factorized networks with $D \geq 2$, however, standard initialization causes strong dependence on $n_{\text{in}}$, $D$, and $L$.

$\square$

## C.3 Proof of Theorem 1

Before proving the theorem, we introduce some required notation. We define the inverse factorization function as $\mathcal{K} : \mathbb{R}^{Dp} \to \mathbb{R}^p$, $\boldsymbol{\omega} \mapsto \boldsymbol{\omega}_1 \odot \ldots \odot \boldsymbol{\omega}_D = \boldsymbol{\varpi}$, and remark that it is a smooth surjection. Using $\mathcal{K}$, we can relate both objectives using the factor misalignment $M(\boldsymbol{\omega}) = D^{-1}\sum_{d=1}^{D}\|\boldsymbol{\omega}_d\|_2^2 - \|\boldsymbol{\varpi}\|_{2/D}^{2/D}$. Using Lemma 1, the DWF objective $\mathcal{L}_{\boldsymbol{\omega},\lambda}(\boldsymbol{\omega})$ can be expressed as $\mathcal{L}_{\boldsymbol{\omega},\lambda}(\boldsymbol{\omega}) = \mathcal{L}_{\mathbf{w},\lambda}(\mathcal{K}(\boldsymbol{\omega})) + \lambda M(\boldsymbol{\omega})$, where the misalignment $M(\boldsymbol{\omega}) \geq 0$ attains zero if and only if $\boldsymbol{\omega}$ represents a balanced factorization. Further, let $B(\mathbf{w}, \varepsilon)$ denote an open ball with radius $\varepsilon$ around $\mathbf{w} \in \mathbb{R}^p$ and recall that $\mathcal{K}$ is continuous at $\boldsymbol{\omega}$ if $\forall \varepsilon > 0 \, \exists \delta > 0 : \mathcal{K}(B(\boldsymbol{\omega}, \delta)) \subseteq B(\mathcal{K}(\boldsymbol{\omega}), \varepsilon)$.

*Proof.* First, we show that if $\hat{\mathbf{w}} \in \arg\min_{\mathbf{w} \in \mathbb{R}^p} \mathcal{L}_{\mathbf{w},\lambda}(\mathbf{w})$, then $\exists \hat{\boldsymbol{\omega}} \in \arg\min_{\boldsymbol{\omega} \in \mathbb{R}^{Dp}} \mathcal{L}_{\boldsymbol{\omega},\lambda}(\boldsymbol{\omega})$ such that $\mathcal{K}(\hat{\boldsymbol{\omega}}) = \hat{\mathbf{w}}$ and $\mathcal{L}_{\mathbf{w},\lambda}(\hat{\mathbf{w}}) = \mathcal{L}_{\boldsymbol{\omega},\lambda}(\hat{\boldsymbol{\omega}})$.
Since $\hat{\mathbf{w}}$ is a local minimizer of $\mathcal{L}_{\mathbf{w},\lambda}(\mathbf{w})$, $\exists \varepsilon_0 > 0 : \forall \mathbf{w}' \in B(\hat{\mathbf{w}}, \varepsilon_0) : \mathcal{L}_{\mathbf{w},\lambda}(\hat{\mathbf{w}}) \leq \mathcal{L}_{\mathbf{w},\lambda}(\mathbf{w}')$. By surjectivity and the multiplicative structure of $\mathcal{K}$, we can pick a balanced factorization $\hat{\boldsymbol{\omega}}$ of $\hat{\mathbf{w}}$ so that $\mathcal{K}(\hat{\boldsymbol{\omega}}) = \hat{\mathbf{w}}$ and $M(\hat{\boldsymbol{\omega}}) = 0$. Balanced factorizations are unique up to sign flip permutations that leave the product sign invariant. Therefore $\mathcal{L}_{\mathbf{w},\lambda}(\hat{\mathbf{w}}) = \mathcal{L}_{\mathbf{w},\lambda}(\mathcal{K}(\hat{\boldsymbol{\omega}})) = \mathcal{L}_{\boldsymbol{\omega},\lambda}(\hat{\boldsymbol{\omega}})$. By continuity of $\mathcal{K}$, $\exists \delta_0 : \mathcal{K}(B(\hat{\boldsymbol{\omega}}, \delta_0)) \subseteq B(\mathcal{K}(\hat{\boldsymbol{\omega}}), \varepsilon_0) = B(\hat{\mathbf{w}}, \varepsilon_0)$, i.e., all $\boldsymbol{\omega}' \in B(\hat{\boldsymbol{\omega}}, \delta_0)$ map to some $\mathbf{w}' \in B(\hat{\mathbf{w}}, \varepsilon_0)$. Then we obtain the following chain of inequalities

$$\forall \boldsymbol{\omega}' \in B(\hat{\boldsymbol{\omega}}, \delta_0) : \mathcal{L}_{\boldsymbol{\omega},\lambda}(\hat{\boldsymbol{\omega}}) = \mathcal{L}_{\mathbf{w},\lambda}(\hat{\mathbf{w}}) \leq \mathcal{L}_{\mathbf{w},\lambda}(\underbrace{\mathcal{K}(\boldsymbol{\omega}')}_{\mathbf{w}'}) \leq \mathcal{L}_{\mathbf{w},\lambda}(\mathcal{K}(\boldsymbol{\omega}')) + \underbrace{M(\boldsymbol{\omega}')}_{\geq 0} = \mathcal{L}_{\boldsymbol{\omega},\lambda}(\boldsymbol{\omega}'),$$

where the first equality holds because $M(\hat{\boldsymbol{\omega}}) = 0$ and the subsequent inequality because $\hat{\mathbf{w}}$ is a local minimizer of $\mathcal{L}_{\mathbf{w},\lambda}(\mathbf{w})$. This shows that $\hat{\boldsymbol{\omega}} \in \arg\min_{\boldsymbol{\omega} \in \mathbb{R}^{Dp}} \mathcal{L}_{\boldsymbol{\omega},\lambda}(\boldsymbol{\omega})$ with $\mathcal{L}_{\mathbf{w},\lambda}(\hat{\mathbf{w}}) = \mathcal{L}_{\boldsymbol{\omega},\lambda}(\hat{\boldsymbol{\omega}})$.

For the other direction, assume that $\hat{\boldsymbol{\omega}} \in \arg\min_{\boldsymbol{\omega} \in \mathbb{R}^{Dp}} \mathcal{L}_{\boldsymbol{\omega},\lambda}(\boldsymbol{\omega})$, i.e., $\exists \varepsilon_0 > 0 : \forall \boldsymbol{\omega}' \in B(\hat{\boldsymbol{\omega}}, \varepsilon_0) :$ $\mathcal{L}_{\boldsymbol{\omega},\lambda}(\hat{\boldsymbol{\omega}}) \leq \mathcal{L}_{\boldsymbol{\omega},\lambda}(\boldsymbol{\omega}')$. By Lemma 1, $M(\hat{\boldsymbol{\omega}}) = 0$ and therefore $\mathcal{L}_{\boldsymbol{\omega},\lambda}(\hat{\boldsymbol{\omega}}) = \mathcal{L}_{\mathbf{w},\lambda}(\mathcal{K}(\hat{\boldsymbol{\omega}})) + M(\hat{\boldsymbol{\omega}}) =$ $\mathcal{L}_{\mathbf{w},\lambda}(\hat{\mathbf{w}})$. We prove that $\hat{\mathbf{w}} = \mathcal{K}(\hat{\boldsymbol{\omega}})$ is a local minimizer of $\mathcal{L}_{\mathbf{w},\lambda}(\mathbf{w})$ by contradiction. Assume $\hat{\mathbf{w}}$ is not a local minimizer of $\mathcal{L}_{\mathbf{w},\lambda}(\mathbf{w})$, then $\forall \delta > 0 : \exists \mathbf{w}' \in B(\hat{\mathbf{w}}, \delta) : \mathcal{L}_{\mathbf{w},\lambda}(\mathbf{w}') < \mathcal{L}_{\mathbf{w},\lambda}(\hat{\mathbf{w}})$. However, if $\hat{\boldsymbol{\omega}}$ is a balanced factorization of $\hat{\mathbf{w}}$, the following auxiliary result shows that for a perturbed $\mathbf{w}'$ around $\hat{\mathbf{w}}$, there must also be a balanced factorization $\boldsymbol{\omega}'$ of $\mathbf{w}'$ close to $\hat{\boldsymbol{\omega}}$:

**Lemma 3.** *Let* $\hat{\boldsymbol{\omega}} \in \mathbb{R}^{Dp}$ *so that* $M(\hat{\boldsymbol{\omega}}) = 0$ *and let* $\mathcal{K}(\hat{\boldsymbol{\omega}}) = \hat{\mathbf{w}}$. *Then* $\forall \varepsilon > 0 \exists \delta > 0 : \mathbf{w}' \in B(\hat{\mathbf{w}}, \delta) \implies \exists \boldsymbol{\omega}' \in B(\hat{\boldsymbol{\omega}}, \varepsilon) : \mathcal{K}(\boldsymbol{\omega}') = \mathbf{w}'$ *and* $M(\boldsymbol{\omega}') = 0$.

*Proof.* We first consider the scalar case of a balanced factorization $\hat{\boldsymbol{\omega}}_j = (\hat{\omega}_{j,1}, \ldots, \hat{\omega}_{j,D})$ mapping to $\hat{\mathbf{w}}_j \in \mathbb{R}$, i.e., $M(\hat{\boldsymbol{\omega}}_j) = 0$ and $\mathcal{K}_j(\hat{\boldsymbol{\omega}}_j) = \hat{\mathbf{w}}_j$. By Lemma 1, the magnitudes of the factors are equal: $|\hat{\omega}_{j,d}| = |\hat{\mathbf{w}}_j|^{1/D}$ for all $d \in [D]$. We now construct factors $\boldsymbol{\omega}'_j$ for any $\mathbf{w}'_j$ close to $\hat{\mathbf{w}}_j$ such that $\mathcal{K}_j(\boldsymbol{\omega}'_j) = \mathbf{w}'_j$, $M(\boldsymbol{\omega}'_j) = 0$, and $\|\boldsymbol{\omega}'_j - \hat{\boldsymbol{\omega}}_j\|_2 < \varepsilon/\sqrt{p}$.

Let $\varepsilon > 0$ be arbitrary. Since $\mathbf{w}_j \mapsto |\mathbf{w}_j|^{1/D}$ is continuous at $\hat{\mathbf{w}}_j$, there exists $\delta_j > 0$ such that $|\mathbf{w}'_j - \hat{\mathbf{w}}_j| < \delta_j$ implies $\left| |\mathbf{w}'_j|^{1/D} - |\hat{\mathbf{w}}_j|^{1/D} \right| < \varepsilon/(\sqrt{D}p)$. The factors $\boldsymbol{\omega}'_j$ are defined as follows:

$$\omega'_{j,d} = \begin{cases} \text{sign}(\hat{\omega}_{j,d}) \cdot |\mathbf{w}'_j|^{1/D}, & \text{if } \hat{\mathbf{w}}_j \neq 0, \ \forall d \in [D], \\ \text{sign}(\mathbf{w}'_j) \cdot |\mathbf{w}'_j|^{1/D}, & \text{if } \hat{\mathbf{w}}_j = 0, \ d = 1, \\ |\mathbf{w}'_j|^{1/D}, & \text{if } \hat{\mathbf{w}}_j = 0, \ d = 2, \ldots, D. \end{cases}$$

This ensures the magnitudes are equal, so $M(\boldsymbol{\omega}'_j) = 0$, and that the product of the factors satisfies $\mathcal{K}_j(\boldsymbol{\omega}'_j) = \mathbf{w}'_j$. For $\hat{\mathbf{w}}_j \neq 0$, we can apply the sign pattern of $\hat{\boldsymbol{\omega}}_j$ to the $\omega'_{j,d}$ by choosing $\delta_j$ small enough. The resulting distance between $\boldsymbol{\omega}'_j$ and $\hat{\boldsymbol{\omega}}_j$ is then:

$$\|\boldsymbol{\omega}'_j - \hat{\boldsymbol{\omega}}_j\|_2 = \sqrt{D} \cdot \left| |\mathbf{w}'_j|^{1/D} - |\hat{\mathbf{w}}_j|^{1/D} \right| < \frac{\varepsilon}{\sqrt{p}},$$

since $\left| |\mathbf{w}'_j|^{1/D} - |\hat{\mathbf{w}}_j|^{1/D} \right| < \varepsilon/(\sqrt{D}p)$. Extending to the vector case, let $\delta = \min_j\{\delta_j\}$. For any $\mathbf{w}' \in B(\hat{\mathbf{w}}, \delta)$, each component $\mathbf{w}'_j$ satisfies $|\mathbf{w}'_j - \hat{\mathbf{w}}_j| < \delta_j$. Applying the scalar construction to each $\mathbf{w}'_j$, we obtain $\boldsymbol{\omega}'$ such that $\mathcal{K}(\boldsymbol{\omega}') = \mathbf{w}'$ and $M(\boldsymbol{\omega}') = 0$. Together, we get

$$\|\boldsymbol{\omega}' - \hat{\boldsymbol{\omega}}\|_2 = \Big( \sum_{j=1}^{p} \|\boldsymbol{\omega}'_j - \hat{\boldsymbol{\omega}}_j\|_2^2 \Big)^{1/2} < \varepsilon.$$

Therefore, for any $\varepsilon > 0$, there exists $\delta > 0$ such that $\mathbf{w}' \in B(\hat{\mathbf{w}}, \delta)$ implies the existence of $\boldsymbol{\omega}' \in B(\hat{\boldsymbol{\omega}}, \varepsilon)$ with $\mathcal{K}(\boldsymbol{\omega}') = \mathbf{w}'$ and $M(\boldsymbol{\omega}') = 0$. $\quad\square$

Choosing $\varepsilon = \varepsilon_0$, then $\exists \delta_0 > 0 : \forall \tilde{\mathbf{w}} \in B(\hat{\mathbf{w}}, \delta_0) \exists \tilde{\boldsymbol{\omega}} \in B(\hat{\boldsymbol{\omega}}, \varepsilon_0) : \mathcal{K}(\tilde{\boldsymbol{\omega}}) = \tilde{\mathbf{w}}$ and $M(\tilde{\boldsymbol{\omega}}) = 0$. By assumption $\hat{\mathbf{w}}$ is not a minimizer of $\mathcal{L}_{\mathbf{w},\lambda}(\mathbf{w})$, hence $\exists \mathbf{w}' \in B(\hat{\mathbf{w}}, \delta_0) : \mathcal{L}_{\mathbf{w},\lambda}(\mathbf{w}') < \mathcal{L}_{\mathbf{w},\lambda}(\hat{\mathbf{w}})$. Let $\boldsymbol{\omega}' \in B(\hat{\boldsymbol{\omega}}, \varepsilon_0)$ be the corresponding balanced factorization of $\mathbf{w}'$ constructed using Lemma 3, with the properties $\mathcal{K}(\boldsymbol{\omega}') = \mathbf{w}'$, $M(\boldsymbol{\omega}') = 0$ and thus $\mathcal{L}_{\boldsymbol{\omega},\lambda}(\boldsymbol{\omega}') = \mathcal{L}_{\mathbf{w},\lambda}(\mathbf{w}')$. But then

$$\exists \boldsymbol{\omega}' \in B(\hat{\boldsymbol{\omega}}, \varepsilon_0) : \mathcal{L}_{\boldsymbol{\omega},\lambda}(\boldsymbol{\omega}') = \mathcal{L}_{\mathbf{w},\lambda}(\mathbf{w}') < \mathcal{L}_{\mathbf{w},\lambda}(\hat{\mathbf{w}}) = \mathcal{L}_{\boldsymbol{\omega},\lambda}(\hat{\boldsymbol{\omega}}), \tag{13}$$

contradicting $\hat{\boldsymbol{\omega}} \in \arg\min_{\boldsymbol{\omega} \in \mathbb{R}^{Dp}} \mathcal{L}_{\boldsymbol{\omega},\lambda}(\boldsymbol{\omega})$. Therefore, if $\hat{\boldsymbol{\omega}}$ is a local minimizer of $\mathcal{L}_{\boldsymbol{\omega},\lambda}(\boldsymbol{\omega})$, then $\hat{\mathbf{w}} = \mathcal{K}(\hat{\boldsymbol{\omega}})$ is a local minimizer of $\mathcal{L}_{\mathbf{w},\lambda}(\mathbf{w})$ with $\mathcal{L}_{\boldsymbol{\omega},\lambda}(\hat{\boldsymbol{\omega}}) = \mathcal{L}_{\mathbf{w},\lambda}(\hat{\mathbf{w}})$. This finishes the proof. $\quad\square$

## C.4 BALANCED FACTORS AND ABSORBING STATES IN SGD OPTIMIZATION (LEMMA 4)

**Lemma 4** (Balanced factors are absorbing states in SGD). *Consider the SGD iterates of a depth-$D$ factorized network with parameters* $\boldsymbol{\omega}^{(t)} = (\boldsymbol{\omega}_1^{(t)}, \ldots, \boldsymbol{\omega}_D^{(t)})$ *at iteration* $t \in \mathbb{N}$, *where the $j$-th entry of the collapsed weight vector* $\boldsymbol{\varpi}^{(t)}$ *is* $\varpi_j^{(t)} = \omega_{j,1}^{(t)} \cdot \ldots \cdot \omega_{j,D}^{(t)}$. *Then, i) if* $\varpi_j^{(t)} = 0$ *and* $M(\boldsymbol{\omega}_j^{(t)}) = 0$, *then* $\omega_{j,d}^{(t')} = 0$ *for all* $d \in [D]$ *and* $t' > t$. *Further, ii)* $M(\boldsymbol{\omega}_j^{(t)}) = 0$ *implies* $M(\boldsymbol{\omega}_j^{(t')}) = 0$ *for all* $t' > t$.

In other words, a balanced factorization at $0$ causes the SGD dynamics to "collapse" and the factors remain zero for all subsequent iterations, effectively reducing the expressiveness of the model.

*Proof.* Consider the SGD updates for the factors $\boldsymbol{\omega}_d \in \mathbb{R}^p$, $d \in [D]$, in a factorized network with $L_2$ regularization. Let $\mathcal{L}_{\boldsymbol{\omega},0}(\boldsymbol{\omega})$ denote the part of the loss function without regularization and assume a batch size of $n$ without loss of generality:

$$\boldsymbol{\omega}_d^{(t+1)} = \boldsymbol{\omega}_d^{(t)} - \eta^{(t)}\big(\nabla_{\boldsymbol{\omega}_d}\mathcal{L}_{\boldsymbol{\omega},0}(\boldsymbol{\omega}^{(t)}) + 2D^{-1}\lambda\boldsymbol{\omega}_d^{(t)}\big) \tag{14}$$

Using the chain rule, the SGD updates are given by:

$$\boldsymbol{\omega}_d^{(t+1)} = \boldsymbol{\omega}_d^{(t)} - \eta^{(t)}\big(\nabla_{\boldsymbol{\varpi}}\mathcal{L}_{\boldsymbol{\omega},0}(\boldsymbol{\omega}^{(t)}) \odot \big(\textstyle\bigodot_{k \neq d}\boldsymbol{\omega}_k^{(t)}\big) + 2D^{-1}\lambda\boldsymbol{\omega}_d^{(t)}\big) \tag{15}$$

To show the collapse in the dynamics for a balanced zero factorization, consider the scalar case $\varpi_j^{(t)} = 0$ with factorization $\boldsymbol{\omega}_j^{(t)} = \{\omega_{j,d}^{(t)}\}_{d=1}^D$ such that $M(\boldsymbol{\omega}_j^{(t)}) = 0$. Then $\omega_{j,d}^{(t)} = 0$ for all $d \in [D]$, and the update becomes:

$$\omega_{j,d}^{(t+1)} = 0 - \eta^{(t)}\big([\nabla_{\boldsymbol{\varpi}}\mathcal{L}_{\boldsymbol{\omega},0}(\boldsymbol{\omega}^{(t)})]_j \cdot 0 + 2D^{-1}\lambda \cdot 0\big) = 0 \tag{16}$$

This holds for all subsequent iterations, proving $\omega_{j,d}^{(t')} = 0$ for all $d \in [D]$ and $t' > t$. Next we show the more general case of SGD dynamics conserving balancedness, i.e., $M(\boldsymbol{\omega}_j^{(t)}) = 0$, or equivalently, $|\omega_{j,1}^{(t)}| = \cdots = |\omega_{j,D}^{(t)}| := m_j^{(t)}$. Let $\omega_{j,d} := s_{j,d}^{(t)} m_j^{(t)}$, where $s_{j,d}^{(t)} = \text{sign}(\omega_{j,d}^{(t)})$ and $s_{\varpi_j}^{(t)} = \text{sign}(\prod_{d=1}^D s_{j,d}^{(t)})$. We investigate the scalar updates:

$$\omega_{j,d}^{(t+1)} = s_{j,d}^{(t)} m_j^{(t)} - \eta^{(t)}\big([\nabla_{\boldsymbol{\varpi}}\mathcal{L}_{\boldsymbol{\omega},0}(\boldsymbol{\omega}^{(t)})]_j \cdot (m_j^{(t)})^{D-1} \cdot \frac{s_{\varpi_j}^{(t)}}{s_{j,d}^{(t)}} + 2D^{-1}\lambda s_{j,d}^{(t)} m_j^{(t)}\big) \tag{17}$$

Because $1/s_{j,d}^{(t)} = s_{j,d}^{(t)}$, we can factor out $s_{j,d}^{(t)}$ from all terms in the update. Hence, the resulting magnitude at iteration $t+1$ is:

$$|\omega_{j,d}^{(t+1)}| = \big|m_j^{(t)} - \eta^{(t)}\big([\nabla_{\boldsymbol{\varpi}}\mathcal{L}_{\boldsymbol{\omega},0}(\boldsymbol{\omega}^{(t)})]_j \cdot (m_j^{(t)})^{D-1} \cdot s_{\varpi_j}^{(t)} + 2D^{-1}\lambda m_j^{(t)}\big)\big| \tag{18}$$

Since the magnitude is constant over $d$, it is shown that $M(\boldsymbol{\omega}_j^{(t')}) = 0$ for all $t' > t$. $\qquad \square$

This "stochastic collapse" (Chen et al., 2024) in the gradient dynamics is a recently investigated phenomenon where the noise in SGD dynamics drives iterates toward simpler *invariant sets* of the weight space that remain unchanged under SGD. The previous result (Lemma 4) about zero misalignment being an absorbing state in DWF with SGD optimization exemplifies this collapse. However, the dynamics that govern the collapse are poorly understood, including how it is determined when and to which simpler structure the model collapses, with unclear implications for generalization in broad settings. The attractivity of these simpler structures is associated with symmetries and high gradient noise levels and closely related to the recently studied Type-II saddle points (Ziyin et al., 2023), potentially helping to explain the benefits of large initial LRs, adaptively regularizing overly expressive networks to constrained substructures via stochastic collapse. While potentially positive effects on generalization were shown, the research community is not yet certain about the broader consequences of this phenomenon.

# D   ALGORITHMS

In the following, we provide the algorithms for the proposed initialization (Section 4.1) of DWF networks in Appendix D.1 and how to train these networks in Appendix D.2.

## D.1   DWF INITIALIZATION

---

**Algorithm 1** DWF Initialization with Variance-Matching and Absolute Value Truncation

---

1: **Input:**
2:    Number $L$ and parameter size $n_l$ of layers, factorization depth $D$, minimum absolute value $\varepsilon$
3:    Standard initializations $\{\mathcal{P}(\mathrm{w}_j^{(l)}) \sim \mathcal{N}(0, \sigma_{\mathrm{w},l}^2)\}_{l=1}^{L}$
4: **for** $l = 1$ to $L$ **do**
5:    $\sigma_l \leftarrow (\sigma_{\mathrm{w},l})^{1/D}$
6:    $\omega_{\min}^{(l)} \leftarrow \varepsilon^{1/D}$
7:    $\omega_{\max}^{(l)} \leftarrow \min\left\{1, (2\sigma_{\mathrm{w},l})^{1/D}\right\}$
8:    **for** each weight $\mathrm{w}_j^{(l)}$ in $n_l$ **do**
9:       **for** $d = 1$ to $D$ **do**
10:          **repeat**
11:             $\omega_{j,d}^{(l)} \sim \mathcal{N}(0, \sigma_l^2)$
12:          **until** $\omega_{\min}^{(l)} < |\omega_{j,d}^{(l)}| < \omega_{\max}^{(l)}$
13:       **end for**
14:    **end for**
15: **end for**
16: **Output:**
17:    Initialized factors $\{\omega_{j,d}^{(l)}\}_{d=1}^{D}$ for all weights $j \in [n_l]$ per layer and all layers $l \in [L]$.

---

## D.2   DWF TRAINING

---

**Algorithm 2** Training Factorized Neural Networks

---

1: **Input:**
2:    Dataset $\mathcal{D} = \{(\boldsymbol{x}_i, y_i)\}_{i=1}^{n}$, network architecture $\mathcal{A}$ with $L$ layers and weights $\mathbf{w} \in \mathbb{R}^p$
3:    Factorization depth $D \geq 2$,
4:    Factor initialization method $\{\texttt{DWF-Init}, \text{base initialization } \mathcal{P}, \varepsilon\}$, (Alg. 1)
5:    Training hyperparameters $\{T, |\mathcal{B}|, \text{LRSchedule } \{\eta^{(t)}\}_{t=1}^{T}, \lambda\}$
6:    $\varepsilon_{\text{tiny}}$   (e.g., float32 machine epsilon $\approx 1.19 \times 10^{-7}$)
7: **Deep Weight Factorization:**
8:    Factorize the weights $\mathbf{w}$ of $\mathcal{A}$ as:
9:    $\mathbf{w} \leftarrow \boldsymbol{\omega}_1 \odot \ldots \odot \boldsymbol{\omega}_D$ and obtain $f_{\boldsymbol{\omega}}(\boldsymbol{\omega})$ from $f_{\mathbf{w}}(\mathrm{w})$
10: **Initialize weights $\boldsymbol{\omega}$ of $f_{\boldsymbol{\omega}}(\boldsymbol{\omega})$:**
11:    $\boldsymbol{\omega} \leftarrow \texttt{DWF-Init}(\mathcal{A}, D, \varepsilon, \text{standard init } \mathcal{P})$
12: **for** each training step $t \in \{0, \ldots, T-1\}$ **do**
13:    Sample mini-batch $\mathcal{B}^{(t)} = \{(\boldsymbol{x}_i, y_i)\}_{i=1}^{|\mathcal{B}|}$ from $\mathcal{D}$ and compute gradient
14:    Update $\boldsymbol{\omega}_d$ using SGD:
15:       $\boldsymbol{\omega}_d^{(t+1)} \leftarrow \boldsymbol{\omega}_d^{(t)} - \frac{\eta^{(t)}}{|\mathcal{B}|} \nabla_{\boldsymbol{\omega}_d}\left(\mathcal{L}_{\boldsymbol{\omega},0}(\boldsymbol{\omega}^{(t)}) + \lambda D^{-1}\sum_{d=1}^{D}\|\boldsymbol{\omega}_d^{(t)}\|_2^2\right) \quad \forall\, d \in [D]$
16:    Update LR:
17:       $\eta^{(t+1)} \leftarrow \text{LRSchedule}(t+1)$
18: **end for**
19: **Post-training factor collapse:**
20:    Collapse factors to obtain weights for $\mathcal{A}$:
21:    $\hat{\boldsymbol{\varpi}} = \boldsymbol{\omega}_1^{(T)} \odot \ldots \odot \boldsymbol{\omega}_D^{(T)}$
22:    Apply numerical mach. epsilon threshold $\varepsilon_{\text{tiny}}$ to remove approx. 0 weights:
23:    $\hat{\varpi}_j \leftarrow 0$ if $|\hat{\varpi}_j| < \varepsilon_{\text{tiny}} \;\forall\, j \in [p]$
24:    Transfer sparse weights $\hat{\boldsymbol{\varpi}}$ back to $\mathcal{A}$
25: **Output:**
26:    Sparse collapsed network parameters $\hat{\boldsymbol{\varpi}} = \hat{\mathbf{w}}$

---

# E DETAILS ON OPTIMIZATION

## E.1 LEARNING RATES IN FACTORIZED NETWORKS

**Ablation study on overall learning rate** Our goal is to determine suitable LR ranges for achieving high sparsity with good generalization in factorized networks. Additionally, we investigate how deeper factorization affects LR requirements. We train factorized LeNet-300-100 with $D \in \{2, 3, 4\}$ and our `DWF` initialization on MNIST, using initial LRs ranging from $10^{-3}$ to 2. All models are trained using SGD with a cosine LR schedule. The results, displayed in Fig. 11, show excessively high LRs lead to unstable results, especially at higher compression ratios. Similarly, too small LRs result in poor or even no sparsification. Notably, models with greater $D$ exhibit more robustness to LR variations, maintaining performance over a wider range of compression ratios compared to shallower factorizations. Across all depths, selecting a large initial LR slightly below the edge where training becomes unstable yields the best overall results, balancing both effective training with high compression ratios. and providing evidence for the importance of a large LR phase in DWF training.

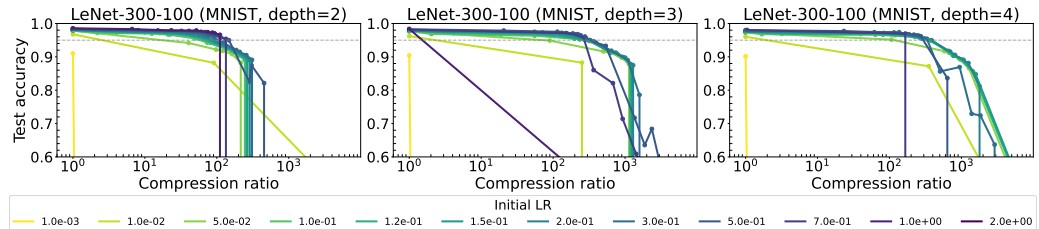

Figure 11: Sparsity-accuracy tradeoffs for a grid of learning rates, demonstrating the importance of appropriately large LRs for DWF. Left to right shows factorization depths $D \in \{2, 3, 4\}$.

**Ablation on the stability of optimal LRs across the sparsity range** In another ablation study, we investigate the impact of different sparsity requirements on the optimal initial LR. To do this, we train a LeNet-300-100 on MNIST for $D \in \{2, 3, 4\}$ on a large number of LR and $\lambda$ combinations. For each $D$, we train all combinations of the learning rate $\eta$ and the regularization $\lambda$, comprising 8 different LRs between $10^{-3}$ and 1, and a grid of 20 $\lambda$ values logarithmically spaced between $10^{-6}$ and $10^{-1}$. We obtain the Pareto frontier for each $D$ by removing all runs that are dominated by other runs in either test accuracy or compression ratio. Fig. 12a shows the corresponding tradeoffs, with the color of the points indicating the optimal learning rate for the corresponding $\lambda$. Confirming the importance of large LRs, the result further demonstrates that the range of optimal LRs remains at a high level across sparsity requirements, except for a slight trend toward distinctly larger LRs for models with little regularization. This can be explained by the intricate relationship between LR and $\lambda$, together forming the *intrinsic* LR. When $\lambda$ is reduced, this is compensated using a larger LR to recover optimal performance (Li et al., 2020).

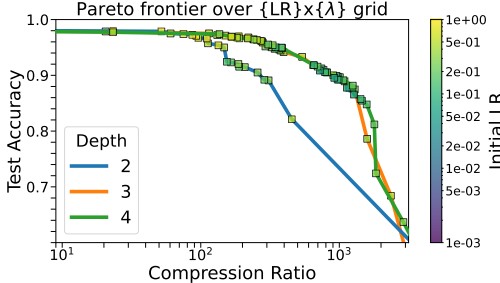

(a) Ablation on optimal LRs at different amounts of sparsity using LeNet-300-100 on MNIST. Note that none of the smaller LRs are selected as optimal.

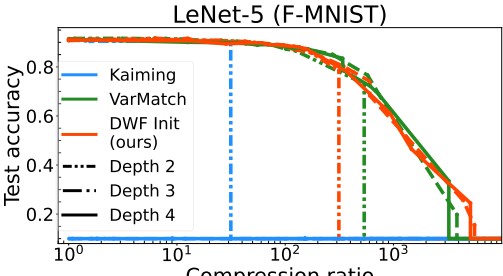

(b) Factor initializations and depths $D$. For $D = 2$, standard initialization performs worse and becomes untrainable for $D > 2$.

Figure 12: Experiments on optimal LRs at different amounts of sparsity and different initialization approaches.

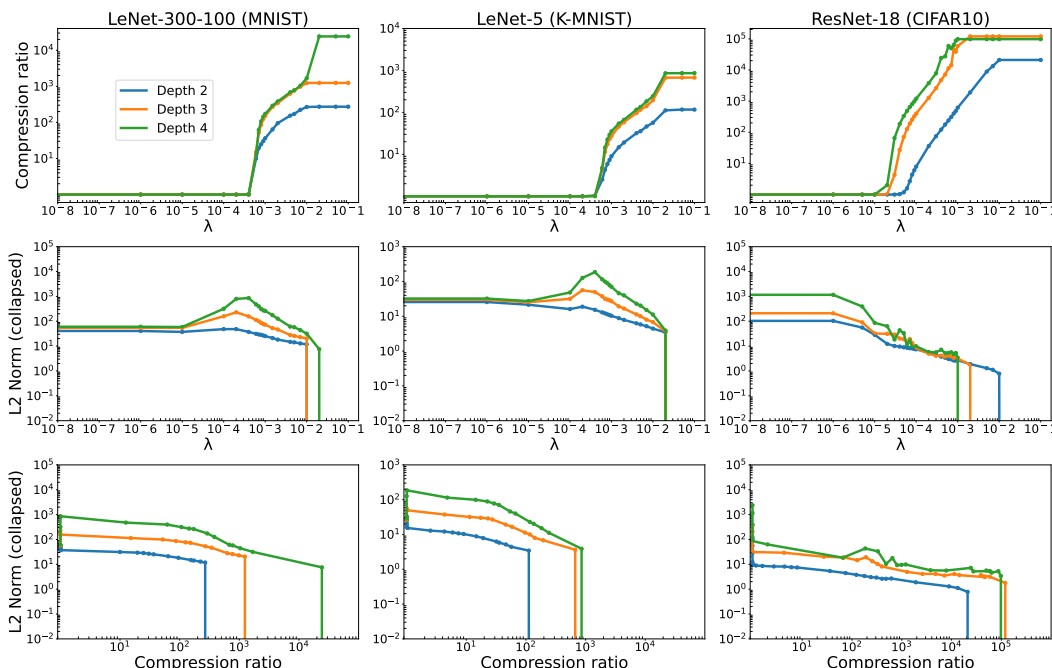

Figure 13: Relationship between different regularization strengths and compression ratio (first row), regularization strength and $L_2$ norm (second row), as well as compression ratio and $L_2$ norm (third row) for different datasets (columns) and different factorization depths $D$ (colors).

## E.2 ABLATION STUDY ON INITIALIZATIONS

In Fig. 12b, we extend the experimental analysis of standard and corrected initialization schemes on performance and sparsity in factorized networks, complementing the experiment on a fully-connected architecture (right plot of Fig. 4) by a convolutional LeNet-5 architecture. Similar to the results in Fig. 4, we observe a failure of standard initialization for $D > 2$. For $D = 2$, contrasting the results for LeNet-300-100 in Fig. 4, standard initialization indeed achieves some sparsity for LeNet-5 on F-MNIST. The attainable tradeoff, however, is vastly outperformed by using the two corrections in the DWF initialization (Algorithm 1).

## E.3 RELATIONSHIP BETWEEN SPARSITY, REGULARIZATION AND WEIGHT NORMS

In Fig. 13, we present results on the relationship between sparsity (measured via the CR), regularization induced by different $\lambda$ values, and the implicit $L_2$ weight norms of the collapsed parameter $\varpi$. From the first row, we see that increases in compression ratio for increasing $\lambda$ values have a similar trend for all depths, starting to induce sparsity at approximately the same regularization strengths. For all datasets and regularization strengths, except for extremely large $\lambda$ values on ResNet-18, the $D = 4$ model always yields a higher compression than $D = 3$, which in turn is sparser than the $D = 2$ model for given $\lambda$. In the second row and for the smaller models, we see a short increase in the $L_2$ norm with increasing $\lambda$, followed by a drop in $L_2$ norm that finally goes to zero at the point where the highest compression is achieved. Remarkably, the collapsed model $L_2$ norm increases with $\lambda$ exactly up to the point where sparsity emerges. A slightly different behavior can be seen for ResNet, where the collapsed norm seems to monotonically decrease for increasing $\lambda$ values (i.e., norms do not increase first and then decrease). Finally, the third row indicates smaller $L_2$ norms the more compressed models become, again with deeper factorizations achieving higher compression ratios at the same $L_2$ norm just before model collapse.

# F    ADDITIONAL RESULTS AND ABLATION STUDIES

## F.1    ABLATION STUDY ON THE FACTORIZATION DEPTH $D$

In our experiments, we considered deeper factorizations up to a depth of $D = 4$. This cut-off is not chosen arbitrarily but follows empirical observations that non-convex $L_q$ regularization achieves an optimal tradeoff between superior sparsity performance and difficulty of numerical optimization roughly at $q = 0.5$ (Hu et al., 2017). In an ablation study, we investigate if this also holds for the DWF approach. Fig. 14 displays the sparsity-accuracy curves attained by factorizations depths up to $D = 8$ and three different LRs in the range that performed well for $D = 4$. We use the same hyperparameter configuration as described in Appendix G. Results show that in all settings, deeper factorizations beyond $D = 4$ offer no improvements in generalization or sparsity, while their training becomes increasingly unstable.

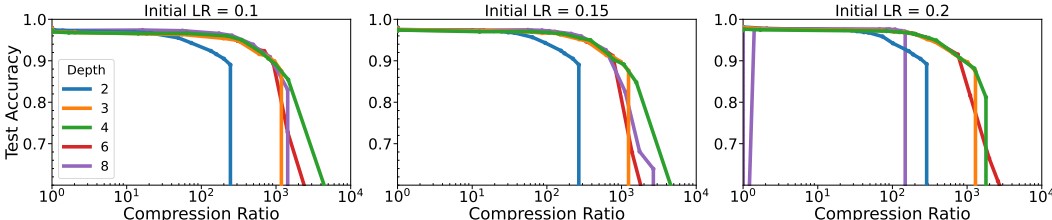

Figure 14: Factorization depths $D > 4$ empirically do not improve performance but become unstable to train. Sparsity-accuracy curves for LeNet-300-100 on MNIST with increasing LRs shown from left to right.

## F.2    COMBINED TRAINING AND VALIDATION ACCURACY

Fig. 16 contains the deferred training and compression trajectories over a range of $\lambda$ values, as shown exemplarily for ResNet-18 on CIFAR10 in the main text (Fig. 7). For improved clarity, we display the running mean of the validation accuracy over three iterations. In addition, Fig. 15 illustrates the learning dynamics for a much finer grid of $\lambda$ values in the top row to provide a clearer picture of how the training trajectories are affected by different $\lambda$ values. Validation accuracies without moving average smoothing are displayed in the bottom row.

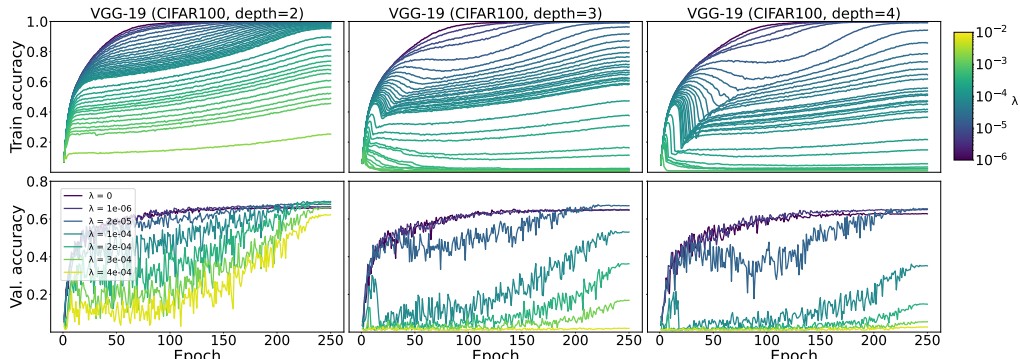

Figure 15: Impact of regularization $\lambda$ on training (**top**) and validation accuracy (**bottom**) for VGG-19 on CIFAR100 and $D \in \{2, 3, 4\}$. The top row shows the training curves for the whole grid of $\lambda$ values. Bottom row shows validation accuracies without running mean for selected $\lambda$.

## F.3    EVOLUTION OF LAYER-WISE COMPRESSION AND WEIGHT NORMS

This section provides a detailed examination of the layer-wise dynamics regarding the evolution of sparsity, complementing the analysis in Section 4.4. Figure 17 illustrates the layer-wise evolution of sparsity (top) and collapsed weight norm (bottom) for different architectures and datasets, using a factorization depth $D = 3$ and increasing regularization strength $\lambda$. The plots reveal broadly consistent patterns across different architectures. For stronger regularization, we observe a more

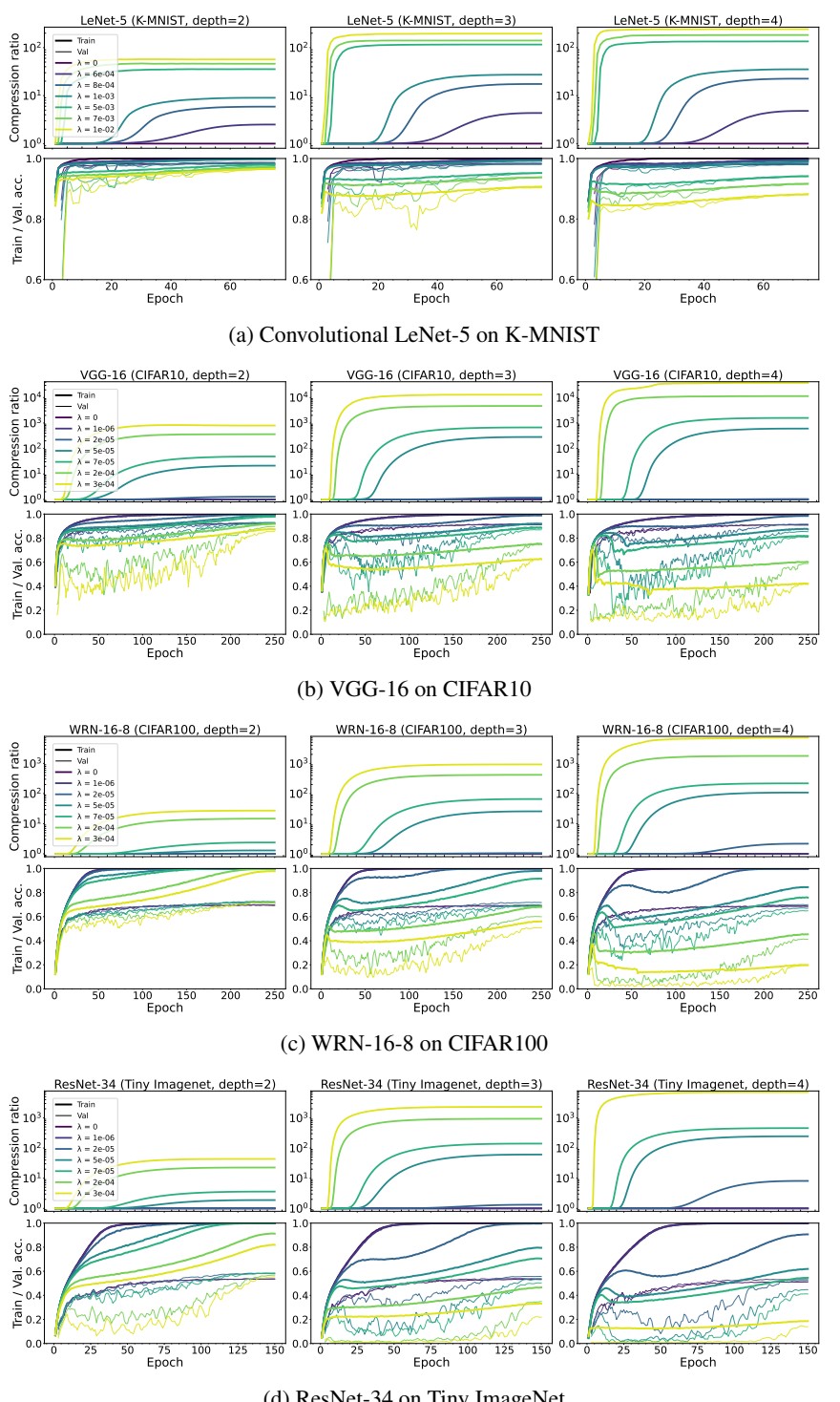

(a) Convolutional LeNet-5 on K-MNIST

(b) VGG-16 on CIFAR10

(c) WRN-16-8 on CIFAR100

(d) ResNet-34 on Tiny ImageNet

Figure 16: Impact of regularization $\lambda$ on compression (**top**), training, and validation accuracy (**bottom**) for various architectures and datasets, using $D \in \{2, 3, 4\}$.

rapid and pronounced onset of sparsity across all layers. Different layers exhibit varying rates of sparsification, with deeper layers generally achieving higher compression ratios more quickly than earlier layers. The layer-wise norm trajectories show a characteristic pattern of initial increase, for the first layer, followed by a peak and gradual decrease. The deeper levels exhibit a simpler dynamic, showing an initial short decline followed by a low plateau. Stronger regularization leads to earlier

peaking and faster decay of weight norms, corresponding to faster sparsification. Notably, the first layer exhibits distinct behavior (cf. Fig. 18), often showing the lowest compression ratio and the highest peak in weight norm. These more complex dynamics indicate stronger feature learning in earlier layers closer to the input. Combined, this analysis provides insights into how DWF affects different parts of the network during training and how this process is mediated by regularization.

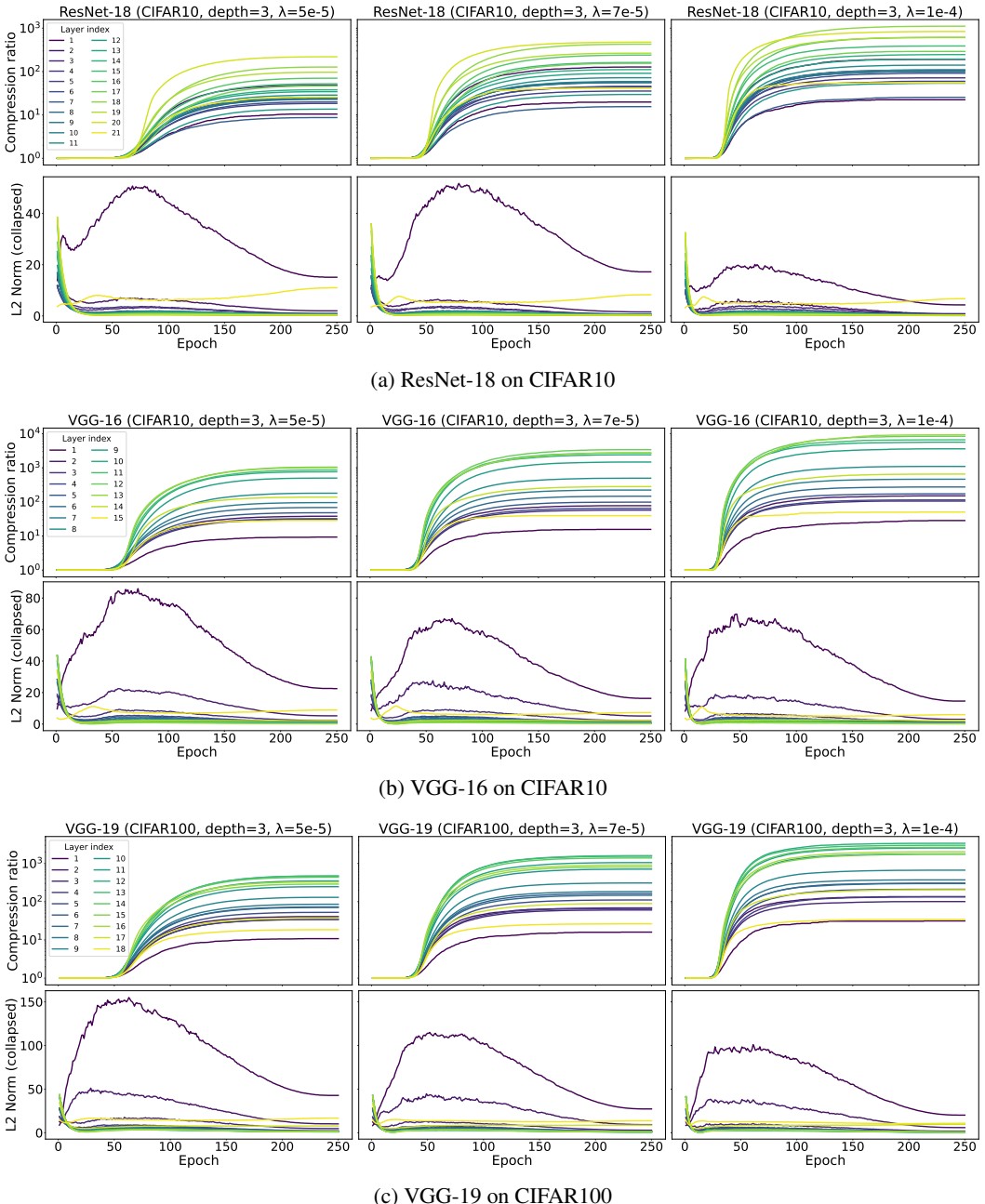

(a) ResNet-18 on CIFAR10

(b) VGG-16 on CIFAR10

(c) VGG-19 on CIFAR100

Figure 17: Layer-wise evolution of sparsity (**top**) and collapsed weight norm (**bottom**) using $D = 3$ and increasing regularization $\lambda$ (left to right) for different architectures and datasets.

## F.4 Evolution of misalignment and onset of sparsity

We investigate the empirical dynamics of the factor misalignment $M(\boldsymbol{\omega})$ and demonstrate that DWF ensures balanced factorizations for sufficiently large $\lambda$. Our analysis reveals an interesting connection between the reduction of misalignment and the onset of sparsity in the learning dynamics, both at the layer-wise and overall model levels.

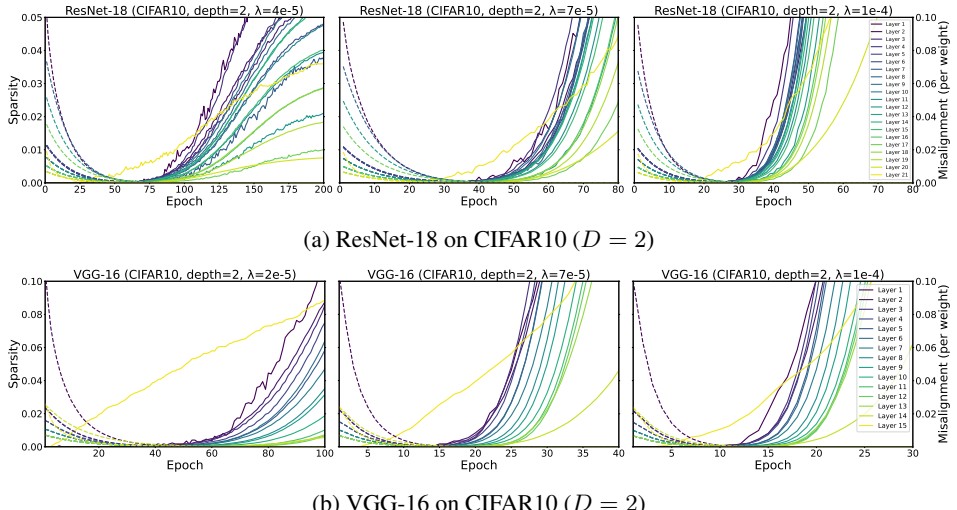

(a) ResNet-18 on CIFAR10 ($D = 2$)

(b) VGG-16 on CIFAR10 ($D = 2$)

Figure 18: Evolution of the average layer-wise factor misalignment (dashed) together with layer-wise sparsity (solid) for ResNet-18 and VGG-16 on CIFAR10 and $D = 2$. Increasing values of $\lambda$ shown from left to right.

Figures 18a and 18b illustrate the layer-wise evolution of sparsity and the average misalignment per layer for depth-2 factorized ResNet-18 and VGG-16 trained on CIFAR10. The factor misalignment $M(\boldsymbol{\omega})$ is calculated at the layer level and normalized by the number of weights in each layer, providing a granular view of misalignment evolution across the network.

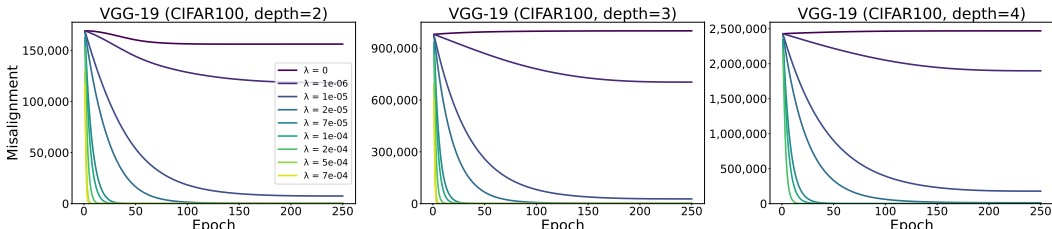

Figure 19: Evolution of factor misalignment $M(\boldsymbol{\omega})$ for VGG-19 on CIFAR100 with increasing $\lambda$ and factorization depths $D \in \{2, 3, 4\}$ (left to right).

The results reveal a clear relationship between the elimination of misalignment and sparsity emergence. The onset of sparsity coincides almost exactly with the elimination of average misalignment per layer, providing empirical evidence for the theoretical connection discussed in Section 3. Larger values of $\lambda$ lead to faster reduction in misalignment and earlier onset of sparsity, demonstrating stronger regularization favors more balanced factorizations.

Two important observations emerge from these results. First, earlier layers broadly exhibit higher initial layer-wise misalignment but decrease at a higher rate than later layers. Surprisingly, a larger initial misalignment coincides with the most rapid and pronounced onset of sparsity as the average misalignment approaches zero. Second, the final layer (yellow) displays distinctly decoupled dynamics, with sparsity emerging within the first few epochs, as opposed to the approximately simultaneous onset for the remaining layers.

We also explore if the onset of sparsity relates to the dynamics of different components of the regularized loss. Figure 20 shows the overall training loss $\mathcal{L}_{\boldsymbol{\omega},\lambda}(\boldsymbol{\omega}^{(t)})$, the data fit part $\mathcal{L}_{\boldsymbol{\omega},0}(\boldsymbol{\omega}^{(t)})$, and the (non-collapsed) factor $L_2$ penalty $D^{-1}\lambda\|\boldsymbol{\omega}^{(t)}\|_2^2$. The $L_2$ penalty is further decomposed into its minimal penalty and the excess penalty or misalignment $\lambda \cdot M(\boldsymbol{\omega}^{(t)})$, as described in Lemma 1. Early in training, the $L_2$ component strongly exceeds the data fit component. Since the data fit levels out much earlier than the $L_2$ penalty, they intersect at some point during training that both LR and $\lambda$ influence. Notably, this point where the loss components are balanced coincides precisely with the onset of sparsity and the overall misalignment approaching zero.

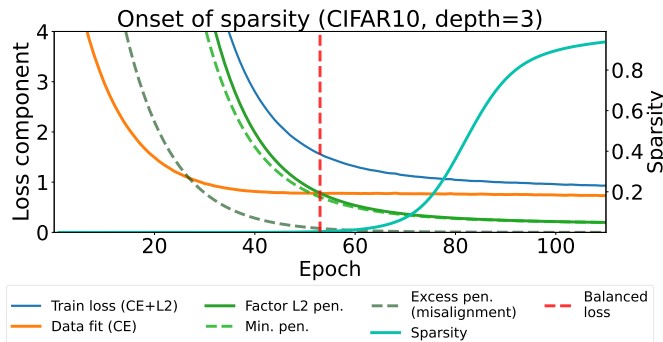

Figure 20: Evolution of loss components and sparsity for ResNet-18 with depth $D = 3$ and $\lambda = 9 \times 10^{-5}$.

### F.5    POST-HOC PRUNING AND FINE-TUNING

Since DWF operates distinctly from most sparsification methods, this offers potential for integration with other pruning techniques. To demonstrate this, we combined DWF with post-hoc pruning on a ResNet-18 with $D = 2$ factorization trained on CIFAR10. The setup used an initial learning rate of $0.27$ and a batch size of $256$. Each model was trained across a range of $\lambda$ values to obtain a raw sparsity-accuracy tradeoff curve. These models were then further pruned along a sequence of compression ratios and fine-tuned for $50$ epochs using SGD with an LR of $0.11$. Fig. 21 presents the results of this experiment. Combining DWF with post-hoc pruning led to increased sparsity at certain accuracy levels up to three times while maintaining comparable accuracy. This demonstrates the potential for integrating DWF with existing pruning techniques.

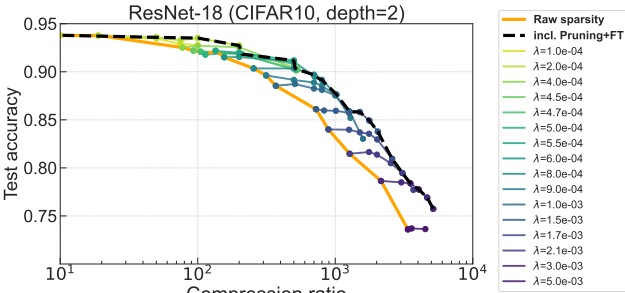

Figure 21: Additional post-hoc pruning and fine-tuning. ResNet-18 ist first trained with DWF and $D = 2$. The models are post-hoc magnitude pruned and re-trained for another 50 epochs.

### F.6    ADDITIONAL SPARSITY-ACCURACY TRADEOFFS

Figure 22 presents sparsity-accuracy tradeoffs for WRN-16-8, ResNet-18, and ResNet-34 on CI-FAR100 and Tiny ImageNet datasets, using factorization depths $D \in \{2, 3, 4\}$. Contrasting our training protocol for section Section 5.3, we do not tune the LRs here and set them to fixed values across datasets and architectures. The results show that DWF consistently produces a range of sparsity-accuracy tradeoffs across different architectures and datasets without incurring model collapse. Deeper factorizations generally achieve higher accuracies at extreme sparsity levels.

### F.7    ADDITIONAL BENCHMARK RESULTS

The following Table 2 shows test accuracies for different compression ratios on different LeNet model specifications and different MNIST datasets. While GMP or SNIP sometimes perform best for 90% or 95% sparsity, DWF models show the highest sparsity in all medium- and high-sparsity cases. In total, Synflow and SNIP each work best in 1 case, GMP in 6 cases, $D = 2$ yields the highest sparsity in 5 cases, $D = 3$ in 14 cases, and $D = 4$ in 21 cases.

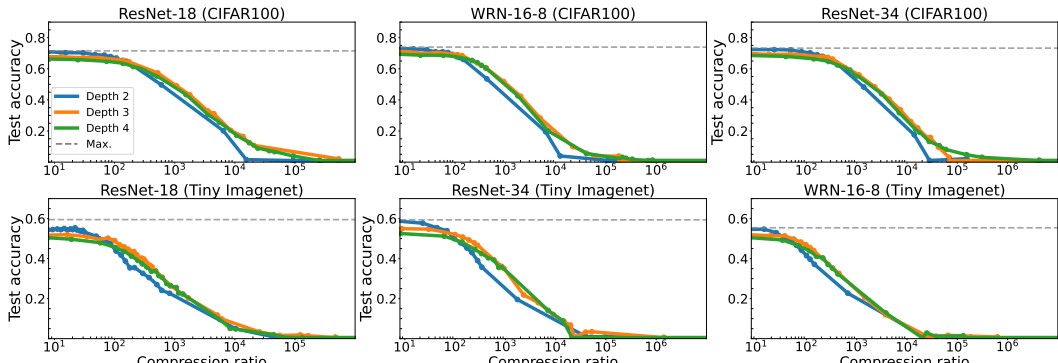

Figure 22: Additional experiments applying DWF to WRN-16-8 and ResNet-18. For these experiments, the LRs were not tuned for each setting but set to $\{0.2, 0.5, 0.7\}$ for $D \in \{2, 3, 4\}$ across models and datasets.

Table 2: Test accuracy (%) for different compression ratios (columns), models (rows), and datasets (table sections).

| Sparsity | 90% | 95% | 98% | 99% | 99.5% | 99.75% | 99.875% | 99.9% |
|---|---|---|---|---|---|---|---|---|
| **LeNet-5** | | | | | | | | |
| **MNIST** | | | | | | | | |
| Dense | 99.26 ± 0.03 | | | | | | | |
| Depth 2 | **99.26 ± 0.04** | **99.27 ± 0.06** | **99.02 ± 0.06** | 98.23 ± 0.09 | 66.88 ± 40.23 | 10.00 ± 0.00 | 10.00 ± 0.00 | 10.00 ± 0.00 |
| Depth 3 | 99.10 ± 0.06 | 99.09 ± 0.05 | **99.02 ± 0.01** | **98.79 ± 0.08** | **97.80 ± 0.10** | 81.22 ± 3.14 | 46.63 ± 26.19 | 40.52 ± 21.58 |
| Depth 4 | 98.95 ± 0.04 | 98.94 ± 0.05 | 98.88 ± 0.02 | 98.66 ± 0.10 | 97.76 ± 0.12 | **85.57 ± 6.45** | **61.31 ± 4.48** | 25.35 ± 21.71 |
| GMP | 99.00 ± 0.07 | 98.75 ± 0.14 | 97.97 ± 0.10 | 83.69 ± 10.07 | 11.35 ± 0.00 | 11.35 ± 0.00 | 11.35 ± 0.00 | 11.35 ± 0.00 |
| SNIP | 98.92 ± 0.21 | 98.63 ± 0.24 | 97.29 ± 0.38 | 64.85 ± 8.81 | 21.73 ± 11.19 | 14.34 ± 5.18 | 11.35 ± 0.00 | 11.35 ± 0.00 |
| Synflow | 99.00 ± 0.04 | 98.68 ± 0.09 | 98.18 ± 0.23 | 96.71 ± 0.63 | 91.97 ± 2.82 | 74.37 ± 7.80 | 56.60 ± 2.68 | **43.68 ± 2.34** |
| Random | 98.28 ± 0.13 | 97.29 ± 0.27 | 59.70 ± 7.65 | 22.61 ± 4.38 | 11.35 ± 0.00 | 11.35 ± 0.00 | 11.35 ± 0.00 | 11.35 ± 0.00 |
| **F-MNIST** | | | | | | | | |
| Dense | 90.41 ± 0.20 | | | | | | | |
| Depth 2 | **91.30 ± 0.13** | **90.78 ± 0.26** | 89.78 ± 0.20 | 88.06 ± 0.18 | 34.68 ± 34.91 | 10.00 ± 0.00 | 10.00 ± 0.00 | 10.00 ± 0.00 |
| Depth 3 | 90.82 ± 0.22 | 90.67 ± 0.16 | **90.10 ± 0.18** | 88.75 ± 0.26 | **85.84 ± 0.72** | **79.15 ± 0.74** | **64.87 ± 2.78** | **60.53 ± 3.17** |
| Depth 4 | 90.65 ± 0.03 | 90.28 ± 0.25 | 90.00 ± 0.19 | **88.77 ± 0.11** | 85.21 ± 0.36 | 77.76 ± 1.41 | 62.90 ± 0.94 | 56.29 ± 2.23 |
| GMP | 90.24 ± 0.14 | 89.71 ± 0.16 | 84.61 ± 0.83 | 25.57 ± 7.01 | 10.00 ± 0.00 | 10.00 ± 0.00 | 10.00 ± 0.00 | 10.00 ± 0.00 |
| SNIP | 90.21 ± 0.59 | 87.21 ± 3.00 | 68.27 ± 15.13 | 48.92 ± 16.00 | 10.00 ± 0.00 | 10.00 ± 0.00 | 10.00 ± 0.00 | 10.00 ± 0.00 |
| Synflow | 89.77 ± 0.13 | 89.16 ± 0.31 | 87.21 ± 0.12 | 84.86 ± 0.37 | 78.68 ± 2.30 | 66.01 ± 9.94 | 45.45 ± 1.90 | 38.97 ± 1.97 |
| Random | 89.28 ± 0.26 | 86.29 ± 0.11 | 46.32 ± 9.87 | 15.60 ± 6.67 | 10.00 ± 0.00 | 10.00 ± 0.00 | 10.00 ± 0.00 | 10.00 ± 0.00 |
| **K-MNIST** | | | | | | | | |
| Dense | 95.58 ± 0.33 | | | | | | | |
| Depth 2 | **95.45 ± 0.24** | 94.56 ± 0.25 | 90.52 ± 0.30 | 81.88 ± 0.15 | 10.00 ± 0.00 | 10.00 ± 0.00 | 10.00 ± 0.00 | 10.00 ± 0.00 |
| Depth 3 | 95.17 ± 0.19 | **94.91 ± 0.15** | **93.08 ± 0.10** | **88.68 ± 0.59** | 78.02 ± 0.58 | 61.69 ± 0.29 | 21.95 ± 16.90 | 10.00 ± 0.00 |
| Depth 4 | 94.72 ± 0.19 | 94.41 ± 0.22 | 92.91 ± 0.21 | 87.91 ± 0.40 | **79.37 ± 0.19** | **61.75 ± 0.90** | **43.09 ± 1.87** | **27.10 ± 12.12** |
| GMP | 93.18 ± 0.42 | 90.92 ± 0.40 | 79.28 ± 0.71 | 50.75 ± 11.04 | 20.12 ± 9.81 | 10.95 ± 1.64 | 10.00 ± 0.00 | 10.00 ± 0.00 |
| SNIP | 91.86 ± 0.73 | 89.00 ± 0.18 | 71.32 ± 1.48 | 26.25 ± 0.37 | 12.64 ± 4.57 | 10.00 ± 0.00 | 10.00 ± 0.00 | 10.00 ± 0.00 |
| Synflow | 92.21 ± 0.31 | 90.59 ± 0.17 | 82.77 ± 1.28 | 72.95 ± 0.46 | 58.95 ± 1.42 | 44.67 ± 3.48 | 27.69 ± 6.51 | 26.96 ± 4.16 |
| Random | 89.52 ± 0.60 | 82.18 ± 0.79 | 32.65 ± 7.40 | 11.20 ± 2.92 | 9.47 ± 0.92 | 10.00 ± 0.00 | 10.00 ± 0.00 | 10.00 ± 0.00 |
| **LeNet-300-100** | | | | | | | | |
| **MNIST** | | | | | | | | |
| Dense | 98.29 ± 0.05 | | | | | | | |
| Depth 2 | 97.49 ± 0.10 | 97.30 ± 0.12 | 96.33 ± 0.02 | 94.54 ± 0.27 | 91.15 ± 0.13 | 10.00 ± 0.00 | 10.00 ± 0.00 | 10.00 ± 0.00 |
| Depth 3 | 97.30 ± 0.13 | 97.25 ± 0.15 | 97.11 ± 0.21 | 96.79 ± 0.19 | 95.83 ± 0.26 | 93.58 ± 0.34 | 90.29 ± 0.24 | **88.55 ± 0.63** |
| Depth 4 | 97.31 ± 0.10 | 97.22 ± 0.12 | 97.08 ± 0.15 | **96.81 ± 0.15** | **96.05 ± 0.25** | **94.27 ± 0.26** | **90.62 ± 0.40** | 88.49 ± 1.00 |
| GMP | **98.34 ± 0.16** | **98.14 ± 0.32** | **97.39 ± 0.56** | 96.59 ± 0.03 | 93.30 ± 1.75 | 75.38 ± 15.13 | 25.97 ± 9.12 | 19.16 ± 6.95 |
| SNIP | 98.09 ± 0.13 | 97.67 ± 0.19 | 96.25 ± 0.66 | 94.51 ± 0.21 | 86.45 ± 4.37 | 59.67 ± 6.41 | 40.38 ± 7.88 | 14.52 ± 2.79 |
| Synflow | 97.83 ± 0.17 | 97.40 ± 0.30 | 96.26 ± 0.59 | 94.03 ± 0.19 | 88.86 ± 0.46 | 75.61 ± 2.02 | 48.75 ± 11.40 | 49.49 ± 5.19 |
| Random | 97.35 ± 0.09 | 95.83 ± 0.35 | 79.32 ± 7.61 | 38.01 ± 10.45 | 14.64 ± 2.90 | 11.35 ± 0.00 | 11.35 ± 0.00 | 11.35 ± 0.00 |
| **F-MNIST** | | | | | | | | |
| Dense | 89.12 ± 0.40 | | | | | | | |
| Depth 2 | 87.95 ± 0.03 | 87.42 ± 0.12 | 86.16 ± 0.12 | 85.05 ± 0.09 | 82.80 ± 0.06 | 53.87 ± 31.02 | 41.90 ± 22.68 | 38.05 ± 20.10 |
| Depth 3 | 87.84 ± 0.12 | 87.66 ± 0.10 | 87.34 ± 0.12 | 86.97 ± 0.11 | 86.02 ± 0.20 | 84.84 ± 0.35 | **82.35 ± 0.11** | **81.35 ± 0.16** |
| Depth 4 | 87.68 ± 0.15 | 87.53 ± 0.21 | **87.35 ± 0.25** | **87.15 ± 0.09** | **86.52 ± 0.16** | **84.96 ± 0.26** | 82.32 ± 0.31 | 81.30 ± 0.40 |
| GMP | 88.22 ± 1.00 | **87.95 ± 1.07** | 87.10 ± 1.30 | 85.22 ± 2.15 | 79.77 ± 5.47 | 55.70 ± 25.45 | 26.55 ± 14.94 | 17.29 ± 6.42 |
| SNIP | **88.61 ± 1.08** | 87.68 ± 0.84 | 82.35 ± 5.34 | 83.76 ± 1.10 | 75.64 ± 5.02 | 21.69 ± 17.05 | 13.23 ± 5.59 | 10.00 ± 0.00 |
| Synflow | 88.16 ± 1.06 | 87.54 ± 0.74 | 86.57 ± 0.75 | 85.23 ± 0.41 | 82.04 ± 0.59 | 76.75 ± 0.33 | 68.29 ± 1.46 | 51.62 ± 13.97 |
| Random | 87.79 ± 0.12 | 87.14 ± 0.57 | 73.35 ± 8.92 | 29.92 ± 9.75 | 16.21 ± 4.80 | 10.16 ± 0.27 | 10.00 ± 0.00 | 10.00 ± 0.00 |
| **K-MNIST** | | | | | | | | |
| Dense | 91.44 ± 0.24 | | | | | | | |
| Depth 2 | 88.26 ± 0.09 | 86.61 ± 0.15 | 80.74 ± 0.24 | 73.23 ± 0.36 | 63.19 ± 0.63 | 10.00 ± 0.00 | 10.00 ± 0.00 | 10.00 ± 0.00 |
| Depth 3 | 87.83 ± 0.20 | 87.49 ± 0.22 | 86.60 ± 0.06 | 83.92 ± 0.19 | 77.75 ± 0.22 | 66.56 ± 3.07 | 52.17 ± 1.94 | 48.71 ± 1.67 |
| Depth 4 | 87.96 ± 0.16 | 87.63 ± 0.17 | **86.99 ± 0.24** | **84.49 ± 0.21** | **78.85 ± 0.66** | **70.16 ± 0.51** | **55.77 ± 1.78** | **51.13 ± 2.19** |
| GMP | **90.43 ± 0.37** | **88.09 ± 0.43** | 83.18 ± 0.34 | 75.34 ± 0.61 | 51.39 ± 0.57 | 21.50 ± 4.05 | 9.41 ± 1.02 | 10.00 ± 0.00 |
| SNIP | 88.51 ± 0.49 | 85.15 ± 0.20 | 79.96 ± 1.03 | 68.30 ± 0.79 | 47.25 ± 2.25 | 25.02 ± 3.44 | 10.00 ± 0.00 | 10.85 ± 1.47 |
| Synflow | 88.52 ± 0.32 | 86.05 ± 0.34 | 82.30 ± 0.45 | 77.29 ± 0.39 | 65.65 ± 1.26 | 52.01 ± 1.36 | 36.22 ± 1.61 | 37.92 ± 2.96 |
| Random | 87.02 ± 0.31 | 82.10 ± 0.15 | 57.30 ± 2.39 | 22.22 ± 3.61 | 11.96 ± 3.39 | 11.65 ± 2.35 | 10.00 ± 0.00 | 10.00 ± 0.00 |

# G EXPERIMENTAL DETAILS

## G.1 DESCRIPTION OF COMPARISON METHODS

In the following, we briefly describe the comparison methods used in our study, covering different approaches of network sparsification before or post-training.

**SNIP** (Single-shot Network Pruning): This method introduces the concept of connection sensitivity to quantify the impact of individual weights on the network's loss function, given by $\mathbf{z}^{(l)} = \left| \mathbf{g}^{(l)} \odot \mathbf{w}^{(l)} \right|$ for layer $l \in [L]$, where $\mathbf{g}^{(l)}$ is the loss gradient with respect to $\mathbf{w}^{(l)}$. By computing this score for each weight at initialization, SNIP identifies and preserves the most crucial connections, enabling effective one-shot pruning before training. This approach has shown remarkable efficacy in maintaining network performance even at high sparsity levels (Lee et al., 2019).

**SynFlow**: As a data-independent pruning approach, SynFlow addresses the important issue of layer collapse in neural network pruning. It utilizes a layerwise conservation principle to ensure conservation of synaptic flow across the network, thereby maintaining high model capacity even under extreme compression ratios. SynFlow has demonstrated state-of-the-art performance at very high sparsity levels, outperforming many data-driven approaches in scenarios where over 99% of parameters are pruned (Tanaka et al., 2020).

**Global Magnitude Pruning** (GMP): This method is based on the assumption that the weight magnitudes are a good proxy for their importance in the network. Despite its heuristic nature, GMP has proven remarkably effective, especially at low sparsity levels. Its success has led to numerous refinements and adaptations of pruning schedules and criteria, with its lasting popularity in both research and practice highlighting its robustness and efficacy (Han et al., 2015; Blalock et al., 2020; Frankle et al., 2020).

**Random Pruning**: Serving as a baseline method, random pruning uniformly removes weights or structures without considering their importance, thereby helping to evaluate the effectiveness of more sophisticated pruning strategies.

## G.2 DETAILS ON ARCHITECTURES, DATASETS, AND TRAINING HYPERPARAMETERS

**Neural network architectures**  In the following, we briefly describe the neural architectures used in our experiments.

- *LeNet-300-100*: This fully-connected network, designed for MNIST classification, consists of an input layer (784 units), two hidden layers (300 and 100 units respectively), and an output layer (10 units). All layers utilize ReLU activation functions. The architecture closely follows the original version proposed by (LeCun et al., 1989), adapted to incorporate modern activations for improved performance.

- *LeNet-5* (LeCun et al., 1998) is a small but effective convolutional network with two convolutional layers (6 and 16 filters, both 5x5), and three fully connected layers (120, 84, and 10 units). We use ReLU activations and add batch normalization (Ioffe & Szegedy, 2015) and average pooling after each convolutional layer.

- *VGG-16* for CIFAR10/100 consists of 13 convolutional layers and 3 fully connected layers (Simonyan & Zisserman, 2014). The convolutional part is described by 2x(64 filters), 2x(128 filters), 3x(256 filters), 3x(512 filters), 3x(512 filters), with max pooling inserted after each group. All filter sizes are 3x3. Batch normalization is applied before each ReLU activation as described by (Lee et al., 2019). *VGG-19* extends VGG-16 by adding one more convolutional layer to each of the last three convolutional blocks, resulting in 19 layers in total. Following (Zagoruyko, 2015), the two fully-connected layers before the output are reduced to a single layer layer with 512 units compared to the ImageNet version.

- *ResNet-18* is a popular residual network with 18 layers (He et al., 2016). In our implementation, the architecture is adapted following common practice for smaller image datasets (Tanaka et al., 2020). We modify the first convolutional layer to use 3x3 filters and remove the initial max pooling layer. The network consists of an initial convolutional layer, followed by 4 stages of basic blocks (2 blocks each), with filter sizes [64, 128, 256, 512]. Global average pooling is used before the fully connected output layer. Likewise, our *ResNet-34* implementation is also adapted for smaller datasets. The architecture follows a

Table 3: Summary of datasets used in experiments.

| Dataset | Training Samples | Test Samples | Classes | Input Features |
|---|---|---|---|---|
| MNIST | 60,000 | 10,000 | 10 | 784 ($28\times28\times1$) |
| F-MNIST | 60,000 | 10,000 | 10 | 784 ($28\times28\times1$) |
| K-MNIST | 60,000 | 10,000 | 10 | 784 ($28\times28\times1$) |
| CIFAR-10 | 50,000 | 10,000 | 10 | 3,072 ($32\times32\times3$) |
| CIFAR-100 | 50,000 | 10,000 | 100 | 3,072 ($32\times32\times3$) |
| Tiny ImageNet | 100,000 | 10,000 | 200 | 12,288 ($64\times64\times3$) |

similar pattern to ResNet-18 with more layers in each stage. As with ResNet-18, we use 3x3 filters in the first layer and omit the initial max pooling, appropriate for the image size of our experiments.

- *WideResNet* is a ResNet variant whose increased width compared to plain ResNets allows for better feature representations. In our experiments, we choose WRN-16-8, which is specifically suited for CIFAR-like tasks (Zagoruyko & Komodakis, 2016).

**Datasets** In our experimental evaluation, we use several standard image classification datasets of varying size and complexity, summarized in Table 3.

MNIST, Fashion-MNIST (F-MNIST), and Kuzushiji-MNIST (K-MNIST) are grayscale image datasets, each containing 10 classes with images of 28x28 pixels. The original MNIST comprises handwritten digits, while F-MNIST contains images of clothing items, and K-MNIST has handwritten Japanese characters. These datasets combine a range of classification tasks with similar input dimensions but varying levels of difficulty.

CIFAR10 and CIFAR100 contain 32x32x3 (color) images with 10 and 100 classes respectively. These datasets present more challenging classification tasks due to their higher resolution, color information, and larger number of classes for CIFAR100.

Finally, Tiny ImageNet is a subset of the ImageNet dataset featuring 200 classes with 64x64x3 color images. This dataset is markedly more challenging and computationally intensive due to the relatively complex task with more and higher resolution images, as well as a larger number of classes. All datasets are split into training (50,000 or 60,000 samples) and test (10,000 samples) sets. We further apply standard data pre-processing and augmentation techniques: For the three MNIST variants, we use pixel rescaling to $[0, 1]$. The CIFAR and Tiny ImageNet images are normalized. For larger networks, we additionally employ data augmentation, including horizontal flips, width and height shifts (up to 12.5%), and rotations (up to $15°$). Table 4 contains the combinations of architectures and datasets we conducted experiments on.

**Training hyperparameters** In our experiments, we use training hyperparameter configurations following broadly established standard settings (Simonyan & Zisserman, 2014; He et al., 2015; Zagoruyko & Komodakis, 2016), as displayed in Table 4. For both LeNet-300-100 and LeNet-5, we set the initial LR to $0.15$ and found it to perform well across datasets, with the exception of LeNet-300-100 on K-MNIST. Because established LRs were found to be suboptimal for DWF, we additionally select the best-performing LR (using small $\lambda = 10^{-6}$) from a discrete grid between $0.05$ and 1 for each factorization depth, architecture, and dataset. For DWF, the sparsity level is controlled using a logarithmically spaced sequence of $\lambda$ parameters between $10^{-6}$ and $10^{-1}$ on which we train each model to obtain the sparsity-accuracy tradeoff curves. For the comparison methods in Section 5.3, we follow the implementation details provided in Frankle et al. (2020); Lee et al. (2019) if available. To make for a fair comparison, we also train the two LeNet architectures using the same LR of $0.15$ and cosine decay. For the larger networks, we only adjust the LR schedule from step to cosine decay but use the prescribed initial LR. To obtain tradeoff curves for the respective pruning methods, we train each method on a sequence of 15 compression ratios between $10^1$ and $10^5$.

**Further details for DWF** Although our method requires no post-hoc pruning, it is sensible to apply a sufficiently small threshold to the final collapsed weights to account for numerical inaccuracies which have no impact on performance. We set this threshold to `float32.mach.eps` $\approx 1.19 \times 10^{-7}$. Additionally, the `DWF` initialization (Algorithm 1) requires specification of the lower truncation threshold for the factor initializations, which we set to $\varpi_{\min} = 3 \times 10^{-3}$ for all our experiments (cf. left plot of Fig. 4).

Table 4: Training hyperparameters for different architectures and datasets. The LRs for the larger models correspond to factorization depths $D = 2, 3, 4$. The comparison methods use standard Kaiming initialization. LRs for the supplementary results on WRN-16-8 and ResNet-34 were not tuned.

| Architecture | Dataset | Epochs | Batch size | Optim. | Mom. | Init. | LR | Schedule |
|---|---|---|---|---|---|---|---|---|
| **LeNet-300-100** | MNIST | 75 | 256 | SGD | 0.9 | DWF-Init | 0.15 | Cosine |
| | F-MNIST | 75 | 256 | SGD | 0.9 | DWF-Init | 0.15 | Cosine |
| | K-MNIST | 75 | 256 | SGD | 0.9 | DWF-Init | 0.4 | Cosine |
| **LeNet-5** | MNIST | 75 | 256 | SGD | 0.9 | DWF-Init | 0.15 | Cosine |
| | F-MNIST | 75 | 256 | SGD | 0.9 | DWF-Init | 0.15 | Cosine |
| | K-MNIST | 75 | 256 | SGD | 0.9 | DWF-Init | 0.15 | Cosine |
| **VGG-16** | CIFAR-10 | 250 | 128 | SGD | 0.9 | DWF-Init | {0.5,0.6,0.6} | Cosine |
| **VGG-19** | CIFAR-100 | 250 | 128 | SGD | 0.9 | DWF-Init | {0.3,0.6,0.6} | Cosine |
| **ResNet-18** | CIFAR-10 | 250 | 128 | SGD | 0.9 | DWF-Init | {0.2,0.5,0.7} | Cosine |
| | CIFAR-100 | 250 | 128 | SGD | 0.9 | DWF-Init | {0.2,0.5,0.7} | Cosine |
| | Tiny ImageNet | 250 | 128 | SGD | 0.9 | DWF-Init | {0.5,0.8,1.1} | Cosine |
| **WRN-16-8** | CIFAR-10 | 250 | 128 | SGD | 0.9 | DWF-Init | {0.2,0.5,0.7} | Cosine |
| | CIFAR-100 | 250 | 128 | SGD | 0.9 | DWF-Init | {0.2,0.5,0.7} | Cosine |
| | Tiny ImageNet | 250 | 128 | SGD | 0.9 | DWF-Init | {0.2,0.5,0.7} | Cosine |
| **ResNet-34** | CIFAR-100 | 250 | 128 | SGD | 0.9 | DWF-Init | {0.2,0.5,0.7} | Cosine |
| | Tiny ImageNet | 150 | 128 | SGD | 0.9 | DWF-Init | {0.2,0.5,0.7} | Cosine |

# H   OTHER APPROACHES TO FACTOR INITIALIZATION

## H.1   ROOT INITIALIZATION AND RESULTS

An alternative option to obtain an initialization of factors $\boldsymbol{\omega}_d$ that recovers the distribution of the original weight $\mathbf{w}$ is given in the following.

**Definition 4** (Root initialization). A root initialization of a depth-$D$ factorized weight $\mathbf{w} = \boldsymbol{\omega}_1 \odot \ldots \odot \boldsymbol{\omega}_D$ is given by first drawing a single standard weight initialization (Definition 3) for each entry of $\mathbf{w}$ and assigning $\boldsymbol{\omega}_1 \leftarrow \text{sign}(\mathbf{w}) \cdot |\mathbf{w}|^{1/D}$ and $\boldsymbol{\omega}_2, \ldots, \boldsymbol{\omega}_D \leftarrow |\mathbf{w}|^{1/D}$ element-wise.

Fig. 23 compares the root initialization against the vanilla initialization and the proposed DWF initialization with and without truncation. While the root initialization yields satisfactory results improving upon vanilla initialization for $D = 2$, we observe that it behaves similarly to the VarMatch initialization for $D = 3$, both outperformed compared to our DWF initialization and yields the worst results for $D = 4$.

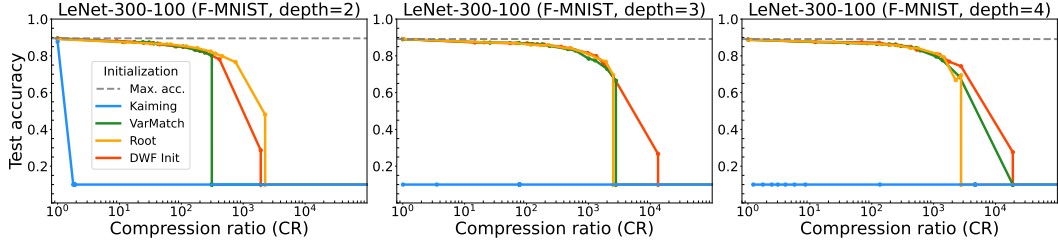

Figure 23: Sparsity-accuracy tradeoffs for different depths $D$ (columns) and initializations (colors).

In Fig. 24, we further analyze the learning dynamics of a DWF model with root initialization. The results demonstrate qualitatively similar learning dynamics to our proposed DWF initialization, suggesting them to be a general feature of DWF and SGD optimization.

## H.2   EXACT GAUSSIAN FACTOR REPRESENTATION

Although the density of random variables whose product is Gaussian is non-trivial, an exact expression in terms of compositions of transformations of Gamma-distributed random variables can be derived. The expression is found by writing the (symmetric) standard Gaussian as $\mathbf{w} = R \cdot e^U$, where $R$ is a Rademacher variable taking values $\pm 1$ with equal probability and $U := \ln |\mathbf{w}|$. Pinelis (2018) establish the infinite divisibility of the exp-normally distributed $U$ by inspection of its characteristic function. This result can be readily exploited to obtain factor distributions such that their $D$-times product is a zero-mean Gaussian with arbitrary variance $\sigma_{\mathbf{w}}^2 > 0$:

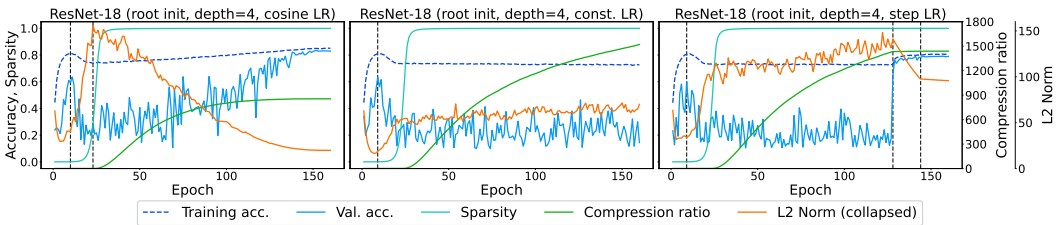

Figure 24: Learning dynamics for the root initialization for different learning rate schedules (columns).

**Lemma 5** (Gaussian Product Factor (GPF) Distribution). *Let $D \in \mathbb{N}^+$. Consider independent and identically (i.i.d.) distributed random variables $\{\omega_d\}_{d=1}^D$ constructed as:*

$$\omega_d \overset{\mathcal{D}}{\sim} R_d \cdot \exp\left\{ \frac{\ln(2 \cdot \sigma_{\mathrm{w}}^2)}{2D} - \mathcal{G}_{0,d} - \sum_{k=1}^{\infty} \left[ \frac{\mathcal{G}_{k,d}}{2k+1} - \frac{1}{2D} \ln\left(1 + \frac{1}{k}\right) \right] \right\}, \tag{19}$$

*where $R_d$ are i.i.d. Rademacher random variables (taking values $\pm 1$ with probability $\frac{1}{2}$) and $\mathcal{G}_{k,d}$ are i.i.d. Gamma$(\frac{1}{D}, 1)$ variables for all $k \geq 0$ and $d \in \{1, \dots, D\}$. Then their product follows a zero-mean normal distribution:*

$$\prod_{d=1}^{D} \omega_d \overset{\mathcal{D}}{\sim} \mathcal{N}(0, \sigma_{\mathrm{w}}^2) \tag{20}$$

*Proof.* We first establish the result for $\sigma_{\mathrm{w}}^2 = 1$ and then extend to arbitrary positive variances. Let $\mathrm{w} \sim \mathcal{N}(0, 1)$. Given the symmetry of the normal distribution, we can represent w as $\mathrm{w} = R \cdot e^U$, where $R$ is a Rademacher random variable and $U = \ln|\mathrm{w}|$ is the logarithm of its absolute value. The characteristic function of $U$ is given by $\phi_U(t) = \mathbb{E}[e^{itU}] = \mathbb{E}[e^{it \ln |w|}]$. Since w has an even density, we can simplify the expectation as:

$$\phi_U(t) = 2 \int_0^\infty e^{it \ln x} \cdot f_{\mathrm{w}}(x) dx \tag{21}$$

Substituting the standard normal density $f_{\mathrm{w}}(x) = \frac{1}{\sqrt{2\pi}} e^{-x^2/2}$:

$$\phi_U(t) = \frac{2}{\sqrt{2\pi}} \int_0^\infty x^{it} e^{-x^2/2} dx \tag{22}$$

Substituting $u = x^2/2$ implies $x = \sqrt{2u}$ and $dx = \frac{du}{\sqrt{2u}}$ and therefore

$$\phi_U(t) = \frac{2}{\sqrt{2\pi}} \int_0^\infty (\sqrt{2u})^{it} e^{-u} \frac{du}{\sqrt{2u}}, \tag{23}$$

The term $(\sqrt{2u})^{it}$ can be written as $2^{it/2} u^{it/2}$, giving using further simplification:

$$\phi_U(t) = \frac{2^{it/2}}{\sqrt{\pi}} \int_0^\infty u^{(it-1)/2} e^{-u} du \tag{24}$$

The integral is the Gamma function with argument $s = \frac{1+it}{2}$:

$$\phi_U(t) = \frac{2^{it/2}}{\sqrt{\pi}} \cdot \Gamma\left(\frac{1+it}{2}\right) = 2^{it/2} \cdot \frac{\Gamma(\frac{1+it}{2})}{\Gamma(\frac{1}{2})} \tag{25}$$

using the identity $\Gamma(\frac{1}{2}) = \sqrt{\pi}$. We now apply Euler's product formula to both gamma functions:

$$\Gamma(z) = \frac{1}{z} \prod_{k=1}^{\infty} \left( \left(1 + \frac{1}{k}\right)^z \cdot \frac{1}{1 + \frac{z}{k}} \right) \tag{26}$$

Let $z = \frac{1+it}{2}$ and $z_0 = \frac{1}{2}$. Then:

$$\Gamma\left(\frac{1+it}{2}\right) = \frac{2}{1+it} \prod_{k=1}^{\infty} \left( \left(1 + \frac{1}{k}\right)^{\frac{1+it}{2}} \cdot \frac{1}{1 + \frac{1+it}{2k}} \right) \text{ and } \Gamma\left(\frac{1}{2}\right) = 2 \prod_{k=1}^{\infty} \left( \left(1 + \frac{1}{k}\right)^{\frac{1}{2}} \cdot \frac{1}{1 + \frac{1}{2k}} \right) \tag{27}$$

Taking their ratio and substituting into our expression for $\phi_U(t)$:

$$\phi_U(t) = 2^{it/2} \cdot \frac{2}{1+it} \cdot \frac{1}{2} \prod_{k=1}^{\infty} \left( \left(1 + \frac{1}{k}\right)^{\frac{1+it}{2} - \frac{1}{2}} \cdot \frac{1 + \frac{1}{2k}}{1 + \frac{1+it}{2k}} \right) \tag{28}$$

The exponent simplifies as $\frac{1+it}{2} - \frac{1}{2} = \frac{it}{2}$, giving:

$$\phi_U(t) = 2^{it/2} \cdot \frac{1}{1+it} \prod_{k=1}^{\infty} \left( \left(1 + \frac{1}{k}\right)^{it/2} \cdot \frac{1 + \frac{1}{2k}}{1 + \frac{1+it}{2k}} \right) \tag{29}$$

For the fraction in the product, multiply numerator and denominator by $2k$:

$$\frac{1 + \frac{1}{2k}}{1 + \frac{1+it}{2k}} = \frac{2k+1}{2k+1+it} = \frac{1}{1 + \frac{it}{2k+1}} \tag{30}$$

Writing the expression in exponential form:

$$\phi_U(t) = e^{\frac{it}{2} \ln 2} \cdot \frac{1}{1+it} \prod_{k=1}^{\infty} \left( e^{\frac{it}{2} \ln(1+\frac{1}{k})} \cdot \frac{1}{1 + \frac{it}{2k+1}} \right) \tag{31}$$

This characteristic function reveals that $U$ can be expressed as:

$$U \stackrel{\mathcal{D}}{\sim} \frac{\ln(2)}{2} - \mathcal{E}_0 - \sum_{k=1}^{\infty} \left[ \frac{\mathcal{E}_k}{2k+1} - \frac{1}{2} \ln(1 + \frac{1}{k}) \right] \tag{32}$$

where $\mathcal{E}_k \stackrel{iid}{\sim} \text{Exp}(1)$ for all $k \geq 0$, since $e^{\frac{it}{2} \ln 2}$ is the characteristic function of the constant $\frac{\ln 2}{2}$, $\frac{1}{1+it}$ is the characteristic function of $-\mathcal{E}_0$, and for each $k$, $e^{\frac{it}{2} \ln(1+\frac{1}{k})}$ is the characteristic function of $\frac{1}{2} \ln(1 + \frac{1}{k})$. Lastly, $\frac{1}{1 + \frac{it}{2k+1}}$ is the characteristic function of $-\frac{\mathcal{E}_k}{2k+1}$. For the factors $\omega_d$, we distribute the components of $U$ across $D$ dimensions using the property that the sum of $D$ independent Gamma$(\frac{1}{D}, 1)$ random variables follows $\text{Exp}(1)$:

$$\ln |\omega_d| \stackrel{\mathcal{D}}{\sim} \frac{\ln(2)}{2D} - \mathcal{G}_{0,d} - \sum_{k=1}^{\infty} \left[ \frac{\mathcal{G}_{k,d}}{2k+1} - \frac{1}{2D} \ln(1 + \frac{1}{k}) \right] \tag{33}$$

where $\mathcal{G}_{k,d} \sim \text{Gamma}(\frac{1}{D}, 1)$. Taking the product:

$$\prod_{d=1}^{D} \omega_d = \left( \prod_{d=1}^{D} R_d \right) \cdot \exp \left( \sum_{d=1}^{D} \ln |\omega_d| \right) \tag{34}$$

Since $\sum_{d=1}^{D} \mathcal{G}_{k,d} \sim \text{Gamma}(1, 1) = \mathcal{E}_k$, the sum of logarithms recovers the distribution of $U$:

$$\sum_{d=1}^{D} \ln |\omega_d| \stackrel{\mathcal{D}}{\sim} U \tag{35}$$

Therefore, $\prod_{d=1}^{D} \omega_d \stackrel{\mathcal{D}}{\sim} \mathcal{N}(0, 1)$. For arbitrary $\sigma_{\text{w}}^2 > 0$, we modify the constant term in the factor distribution to $\frac{\ln(\sigma_{\text{w}}^2)}{2D}$ to produce the desired product variance, yielding the result:

$$\omega_d \stackrel{iid}{\sim} R_d \cdot \exp \left\{ \frac{\ln(2 \cdot \sigma_{\text{w}}^2)}{2D} - \mathcal{G}_{0,d} - \sum_{k=1}^{\infty} \left[ \frac{\mathcal{G}_{k,d}}{2k+1} - \frac{1}{2D} \ln \left(1 + \frac{1}{k}\right) \right] \right\} \Rightarrow \prod_{d=1}^{D} \omega_d \stackrel{\mathcal{D}}{\sim} \mathcal{N}(0, \sigma_{\text{w}}^2). \tag{36}$$

$\square$

.

**Practical challenges** Unfortunately, it is hard to efficiently implement sampling from the factor distribution due to the infinite sum and the series of additional operations required to compute $\omega_d$. Since the summands tend to 0 as $k \to \infty$, it seems practical to truncate the infinite sum at a reasonable value to approximate the true factor distribution. Fig. 25 shows the kernel density estimates of the factor distribution for $D = \{2, 3, 4, 10, 20\}$ as well as the densities of the resulting product

alongside an overlay of the ground truth Gaussian. To obtain the estimates, $n = 1000$ products were sampled using the target standard deviation $\sigma_{\mathrm{w}} = 0.1$. The plots in Fig. 25 show the approximation power computing only the first $k \in \{1, 5\}$ terms of the infinite sum. The results show that even the coarsest approximation using only a single summand $k = 1$ yields product distributions that are statistically indistinguishable from a Gaussian using a Kolmogorov-Smirnoff test for any factorization depth $D$. Expectedly, the approximation improves with the number of summands $k$.

A significant drawback to the exact Gaussian factorization approach using Lemma 5 over the proposed `DWF` initialization is its impractical initialization complexity, rendering the approach unfeasible in practice. This is owed to the accumulating number of additional operations required for each scalar weight, such as drawing multiple random variables per single factor $\omega_d$, computing several constants, summing or multiplying them, and exponentiation. For example, initializing a factorized ResNet-18 with $D = 2$ using a vanilla initialization takes $\approx 1s$, for the proposed `DWF` initialization $\approx 3s$, but for the coarsely approximated exact Gaussian factor distribution with $k = 1$ around 180 minutes on an A-4000 GPU with 48GB RAM, despite yielding identical performance to the `DWF` init. The same network could not be initialized within a time budget of six hours for depths $D > 2$.

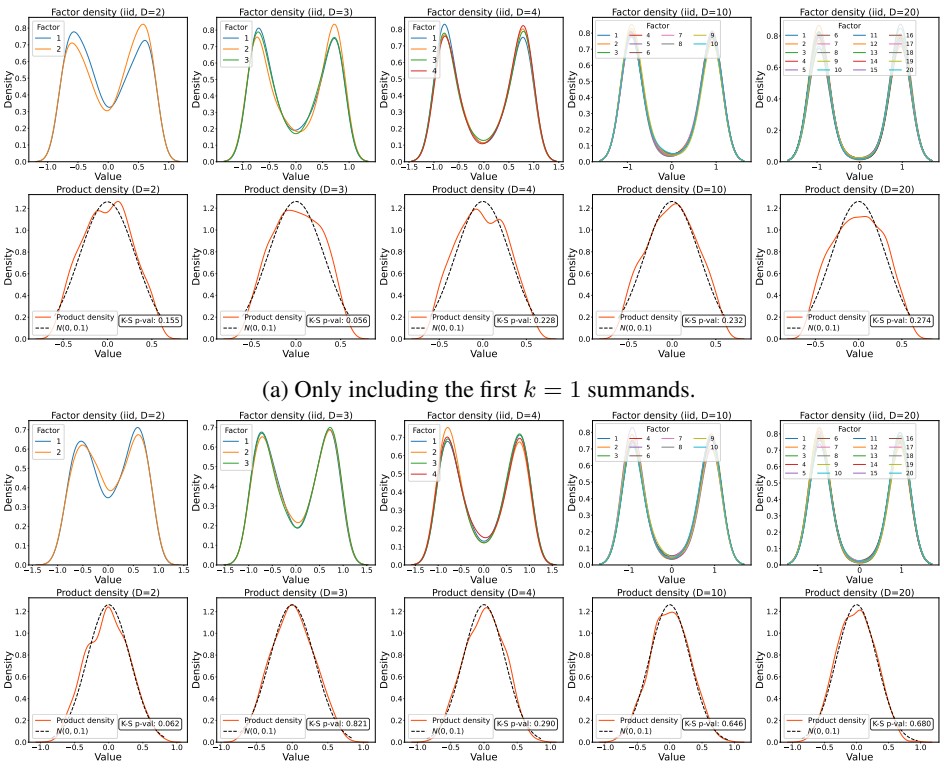

(a) Only including the first $k = 1$ summands.

(b) Only including the first $k = 5$ summands.

Figure 25: Approximation of a Gaussian product for different truncation values of the infinite sum in Lemma 5. K-S corresponds to the Kolmogorov-Smirnoff test for normality.

# I COMPUTATIONAL ENVIRONMENT AND RUNTIME ANALYSIS

## I.1 COMPUTATIONAL ENVIRONMENT

Large experiments on ResNet-18 and VGG-19 on datasets CIFAR10, CIFAR100, and Tiny ImageNet were run on an A-100 GPU server with 32GB RAM and 16 CPU cores. Smaller experiments were conducted on a single A-4000 GPU with 48GB RAM or CPU workstations.

## I.2 RUNTIME ANALYSIS

Here, we investigate the computational overhead induced by DWF compared to vanilla training. We conducted experiments with WRN-16-8 on CIFAR10 and VGG-19 on CIFAR100 across various batch sizes. Each model was trained for 1000 iterations using SGD. We measured the average wall-clock time per sample and peak GPU memory utilization during training. All experiments were performed on a single A-4000 GPU with 48GB RAM, repeated five times to report means and standard deviations. Our results, displayed in Fig. 26 and Fig. 27, show that the factorization depth $D$ in the DWF model only has a minor impact on computational costs during training. For batch sizes of 256 or higher, both networks exhibit an indistinguishable time per sample comparable to vanilla training across all levels of $D$. At smaller batch sizes, we observe a slight monotonic increase in runtime with greater D. For example, WRN-16-8 with a batch size of 128 and $D = 2$ runs approximately 10% longer than vanilla training, while VGG-19 with a batch size of 64 and $D = 4$ shows the largest increase of about 80%. These findings demonstrate that DWF training under typical settings incurs only a small additional cost compared to standard training. This contrasts with many sparsification techniques like Iterative Magnitude Pruning (Frankle & Carbin, 2019), which can lead to several-fold increases in training time due to multiple cycles of pruning and re-training.

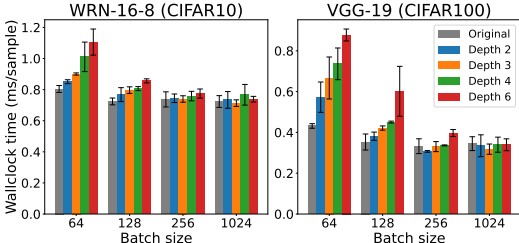

Figure 26: Comparison of wall-clock time per sample for WRN-16-8 (**left**) and VGG-19 (**right**) on CIFAR10/100 across factorization depths $D$. Results indicate insignificant runtime overhead for DWF compared to vanilla training, particularly for larger batch sizes where runtime is identical.

GPU memory utilization is primarily dependent on batch size, with $D$ having rather small effects in total. In conclusion, besides factorizing the weights into $D$ factors, DWF incurs only a minor additional runtime and memory cost on commonly used convolutional architectures. The minimal increase, especially for typical batch sizes, suggests DWF can be readily integrated into existing training protocols without major changes in computational overhead.

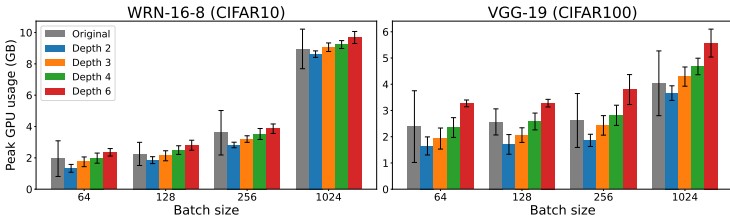

Figure 27: Peak GPU memory utilization for WRN-16-8 (**left**) and VGG-19 (**right**) across depths $D$. The results show that batch training is the dominant factor for memory usage, with only a minimal impact of $D$.

