# OpenReview forum: "Deep Weight Factorization: Sparse Learning Through the Lens of Artificial Symmetries"
_ICLR.cc/2025/Conference — ICLR 2025 Poster_

### Official Review · Reviewer_ogCH · 2024-10-31

**Soundness:** 3
**Presentation:** 4
**Contribution:** 3
**Rating:** 8
**Confidence:** 3

**Summary:**

Building upon the fact that regularizing neural networks with an L1 penalty is not adequate due to its non-smoothness and its limited compatibility with SGD, the authors investigate the application of the weight factorization framework to induce sparse weights upon training of deep learning models.

They extend the ideas of [Hoffn, 2017] and [Ziyin & Wang, 2023] to arbitrary factorization depths of neural network weights, and provide theoretically grounded equivalence of training deep factorized neural nets and sparse optimization problems (Theorem 1).

The authors then address the difficulties in optimizing their model with standard optimization practices and suggest more suited strategies. Furthermore, they analyze the training dynamics of these networks (namely the accuracy-sparsity interactions during training) and provide an empirical evaluation of the performances of their approach against other sparsification frameworks.

**Strengths:**

The presentation is very clear, with a nice balance of theoretical investigations coupled with empirical evaluations.
I enjoyed reading this paper.

I find Figure 2 to be especially clear and informative, offering a nice intuitive overview of the approach.

I would also like to highlight the thoroughness of the appendix which provides appreciated details and references to related literature.

Overall, factorizing the weights of a network to arbitrary depths to ensure the sparsity of the trained model upon collapsing the weights is an interesting idea that I find worth exploring.

**Weaknesses:**

I will provide here some suggestions on the presentation as well as technical questions.

* Figure 1 displays accuracy as function of compression ratio (CR), but we don't know what that refers to at that point (it is introduced later in section 2.1). I would suggest defining CR in the caption of the figure or at least point to its definition.

* I am not sure to understand what Figure 3 aims to show. Are the curves' colors referring to the value of $c$ in Definition 1 ?

* I do understand the inevitability of a sparsity-accuracy tradeoff, and acknowledge the link made by the authors regarding the use of large LRs and generalization performance. However, I still fail to wrap my head around why the LR has such a significant impact on the sparsity of the collapsed weights, given that the authors previously show the equivalence between DWF and sparse regularization. At convergence, shouldn't sparsity be present no matter what ? I would greatly appreciate if you could perhaps provide me some more intuition on this.

* I acknowledge the memory and computational advantages in using sparser neural networks, but I was wondering how relevant these advantages would be if one could instead just directly train a dense but smaller (with less parameters) network to begin with. If a large but sparse network can achieve competitive performance with its dense counterpart, wouldn't it be fair to assume that a small but dense network could also be competitive ? If so, what would be the point in sparsifying the large network ?

* This is more of a research question, but there seems to be intricate interactions between accuracy and sparsity in DWF models. Do you plan on investigating these from a theoretical standpoint ? Perhaps it could be interesting to study the drops in accuracy observed in Figure 1 after a certain compression ratio is reached ?

**Questions:**

Please refer to the Weaknesses section.

**Details Of Ethics Concerns:**

Not applicable.

---

> ### Author Response · Authors · 2024-11-15
> **Response to questions/weaknesses**
>
> Dear reviewer ogCH,
>
> We would like to thank you for your comments and interesting questions and are happy that you enjoyed reading the paper. We will address your questions in the following:
>
> - (Q1) Thank you for pointing this out. The revised version of our submission will make the meaning of the compression ratio (CR) in Fig. 1 clear directly where it appears.
> - (Q2) Fig. 3 shows the level sets of possible factorizations of a scalar $w$ using two factors $w=\omega_1 \cdot \omega_2$, all of which result in the same product weight $w$ and therefore the same induced network and loss. The different colors indicate different values of $w$ (0 and 0.25), showing how the set of possible factorization looks like for zero and non-zero products. Different values of $c$ in Def. 1 would correspond to different points on the same curve. The level set of $w=0$ is simply the coordinate axes, and $L_2$ regularization enforces the selection of the min-norm factorization $(0,0)$. For $w=0.25$ the level set forms a rectangular hyperbola, with two points (its vertices) having minimal distance to the origin and therefore minimal $L_2$ penalty. The figure also illustrates the balancedness property of minimum-norm factorizations, as the vertices of a rectangular hyperbola always lie either on the diagonal $\omega_2=\omega_1$ or $\omega_2=-\omega_1$. A similar reasoning applies to deeper factorizations, which are, however, hard to visualize. We will include a dedicated section for the intuition underlying DWF using Fig. 3 in the Appendix of our revised submission.
> - (Q3) Thank you for this interesting comment. While it is true that all minima of the factorized objective are identical with the sparse solutions using $L_{2/D}$ regularization, it is a vastly different question whether these minima can be reached in reasonable time using standard computational budgets (e.g., epochs) for the respective applications. Using too small of an LR considerably slows down the reduction in misalignment, and large misalignment values at the end of training (for $\lambda>0$) show that we can not be close to a minimum by Lemma 1. For the illustration of the failure of small LRs in Fig. 5 (b), the model with LR=0.05 has a misalignment of $1060.7$ at the end of training, while the model with LR=0.15 has a misalignment of $0.38$, supporting this explanation.
> - (Q4) We agree with the conjecture that training a dense but small network can yield the same performance as a sparsified large network if the network topology is not significantly altered (see Lottery Ticket Hypothesis by Frankle and Carbin, 2019). While the value of network pruning is often debated (e.g. Liu et al., 2019; Wang et al., 2023), in practice, fixed architectures like specific ResNets are used without tuning the network topology to the task, necessitating sparsification for efficient parameter use. Secondly, DWF is an integrated sparse learning method (instead of post-hoc pruning) that directly learns sparse representations, delivering an additional benefit over pruning methods where optimization and sparsification are decoupled.
> - (Q5) We attribute the sudden drop in accuracy in Fig. 1 to the discrete grid of 20 $\lambda$ values we evaluate. However, we plan on studying the interesting drop in training acc. observed **during training** (Fig. 7) from a theoretical standpoint.
>
> Does this answer your questions and can we clarify anything else for you?
>
> **References**:
>
> Li, Zhiyuan, Kaifeng Lyu, and Sanjeev Arora. "Reconciling modern deep learning with traditional optimization analyses: The intrinsic learning rate." NeurIPS 33 (2020): 14544-14555.
>
> Frankle, Jonathan, and Michael Carbin. "The Lottery Ticket Hypothesis: Finding Sparse, Trainable Neural Networks." ICLR. 2019.
>
> Liu, Zhuang, et al. "Rethinking the Value of Network Pruning." ICLR. 2019.
>
> Wang, Huan, et al. "Why is the state of neural network pruning so confusing? on the fairness, comparison setup, and trainability in network pruning." arXiv preprint arXiv:2301.05219 (2023).

---

> ### Author Response · Authors · 2024-11-20
> **Revised submission**
>
> Dear reviewer ogCH,
>
> We would like to thank you again for your helpful comments and have uploaded a revised submission. The revised submission includes the following changes related to your comments:
>
> >Figure 1 displays accuracy as function of compression ratio (CR), but we don't know what that refers to at that point (it is introduced later in section 2.1). I would suggest defining CR in the caption of the figure or at least point to its definition.
>
> We have now included the definition of the compression ratio directly in the axis label of Fig. 1.
>
> >I am not sure to understand what Figure 3 aims to show. Are the curves' colors referring to the value of c in Definition 1 ?
>
> We have included a dedicated section (Appendix B in the revised submission) containing an in-depth discussion of the geometric intuition underlying the sparsity-promoting effects of $L_2$ regularized (minimum-norm) weight factorization, as illustrated in Fig. 3.
>
> We hope these additions are able to adequately address your comments and suggestions and thank you again for your valuable feedback.

---

> ### Comment · Reviewer_ogCH · 2024-11-21
> **Response to authors**
>
> I would like to thank the authors for the clarifications and the thoroughness of their responses. I will retain my original rating of 8 (accept).

---

### Official Review · Reviewer_EGQD · 2024-11-01

**Soundness:** 3
**Presentation:** 3
**Contribution:** 3
**Rating:** 6
**Confidence:** 4

**Summary:**

This paper extends earlier results on weight factorization and proposes a differentiable (though non-convex) regularized training objective for neural networks whose global and local optima coincide with those of a problem with the same training loss and with $\frac{2}{D}$-regularization on the network weights. For $D \geq 2$, the naive regularizer is sparsity-inducing but non-differentiable, and solutions to the latter problem fail to achieve competitive results at high compression ratios. The proposed solution is to factorize the network weights into a product of $D$ matrices and apply $L_2$ regularization to each factor independently; the authors call their factorization “deep weight factorization” (DWF). The paper’s key result, Theorem 1, conclusively supports the correctness of their method in theory – though there is no guarantee that one reaches equally desirable local minima in practice while solving each problem.

The authors then note that standard neural network weight initializations yield poor results for DWF and propose an alternative initialization that fixes the variance of the weights while avoiding weights that are too close or far from 0. They also study the impact of learning rate schedules on the trained networks’ sparsity-accuracy tradeoffs. They finally present empirical results to show that DWF outperforms shallow weight factorizations (albeit marginally for datasets such as CIFAR10/100 and Tiny Imagenet) and post-training pruning methods.

**Strengths:**

- The paper develops a natural extension of existing shallow weight factorization methods to construct differentiable regularizers whose corresponding optima are equivalent to those obtained by non-convex and non-differentiable $L^{(2/D)}$ regularization. This is a valuable – and to my knowledge original – theoretical contribution to the literature.
- The authors present extensive empirical results to study the impact of choices such as initializations and learning rates on their method’s performance. Doing this legwork to resolve critical pitfalls in the implementation of their method enables other researchers to use their work in practice, and substantially increases their method’s potential for impact.
- The paper is generally well-written, and I was able to follow its exposition without too much trouble.

**Weaknesses:**

My primary critique of this paper is that DWF’s performance improvements over shallow weight factorization and Synflow (in terms of compression ratio achieved at acceptable cost to accuracy) seem marginal for CIFAR10/100 and Tiny Imagenet. DWF does improve over competing methods by large margins on toy problems such as MNIST and its variants – but the gap between its relative performance on MNIST and on somewhat more complex datasets such as CIFAR and Tiny Imagenet makes me wonder whether the claimed improvements would become even narrower once one leaves the realm of standard ML benchmarks and tackles real-world problems. In light of this and my first question below, the authors may consider tempering their claims regarding the empirical benefits of their method over shallow weight factorization and pruning.

**Questions:**

- On lines 475-6, the authors claim that “For VGG-19 on CIFAR100 at 10% tolerance, DWF (D = 4) achieves a compression ratio of 1014, surpassing the runner-up (SynFlow at 218) by a factor of almost 5.” Perhaps I’m misreading the bottom-left plot in Figure 9, but don’t $D=2$ and D=3 both outperform $D=4$ at the 10% tolerance threshold (i.e. ~60% accuracy)? Furthermore, $D=2$ may be the best-performing method in this regime (though it is very close to $D=3$) – and this corresponds to shallow weight factorization, which is not an original contribution of this paper.
- Why use LeNet architectures for the MNIST experiments? To my knowledge, these are essentially never used in practice. Is there a particular scientific reason for using a very simple architecture in these experiments?
- The plots in Figures 6 and 7 are quite cluttered. Would the authors consider depicting curves for fewer values of $\lambda$ in Figure 7? They could also be more selective with the information they present in Figure 6.

---

> ### Author Response · Authors · 2024-11-15
> **Response to questions**
>
> Dear reviewer EGQD,
>
> We thank you for your time and interesting questions and respond to the questions below. A response to the weaknesses is included in a separate comment.
>
> ### Question 1
> Thank you for pointing this out. The relevant lines 475-476 incorrectly referenced $D=4$ instead of $D=3$ and are replaced by “**For VGG-19 on CIFAR100 at $10\%$ tolerance, DWF ($D=3$) achieves a compression ratio of $1014$, surpassing the runner-up pruning method (SynFlow at $218$) by a factor of almost $5$.**” in a revised submission.
>
> ### Question 2
> >Why use LeNet architectures for the MNIST experiments? To my knowledge, these are essentially never used in practice. Is there a particular scientific reason for using a very simple architecture in these experiments?
>
> Our motivation for using these architectures and datasets is three-fold. First, as mentioned in your question, they are **well-established ML benchmarks** whose results researchers can interpret intuitively and are for this reason often also included in other recent sparse learning works (e.g., Yang et al., 2019; Schwarz et al., 2021; Vysogorets and Kempe, 2023; Dhahri et al., 2024). Secondly, the small size of the architectures allows for **extensive ablation studies** to be conducted without much computational overhead, as done in, e.g., Fig. 1; Fig. 4 (right); Fig. 5 (right); Appendices D.1-D.3, E.1-E.3. Third, a common counterargument against simple architectures or tasks is that all methods perform equally well and they become indistinguishable (Liu and Wang, 2023), which does not hold in our experiments (where there is a **clear distinction in the accuracy-compression trade-offs**). Besides these points, our implementation of LeNet-300-100 is a fully-connected network with two hidden layers and ReLU activation. While small architectures are superseded by larger models for modern computer vision tasks, they are **still commonly applied**, e.g., for tabular data applications or in more applied fields, also in a sparse learning context (Lemhadri et al., 2021; Balderas et al., 2024).
>
> ### Question 3
> >The plots in Figures 6 and 7 are quite cluttered. Would the authors consider depicting curves for fewer values of $\lambda$ in Fig. 7? They could also be more selective with the information they present in Fig. 6.
>
> We **agree** with the observation and will remove two $\lambda$ values from the plot in Fig. 7 in a revised submission to declutter it. The analysis of training dynamics in Fig. 6 depicts the interplay of different quantities throughout training. It requires a simultaneous depiction of a) training fit, b) generalization, c) effective model norm/size, and d) sparsity/compression ratio to identify the different phases. The only partially redundant information is plotting both sparsity and compression ratio (CR). This was done because the CR does not show the onset of sparsity well, whereas plotting only the sparsity does not visualize the continued sparsification above a sparsity of 99% (which is reflected well in the CR). For example, the final sparsity in the second and third plots in the bottom row of Fig. 6 looks identical, although their final CR differs by a factor of $\approx 20$. However, since the emphasis of Fig. 6 is on the phased training dynamics, we will omit the CR in a revised submission and only plot sparsity alongside the other metrics to improve the readability of the plots.
>
> Does this answer your questions and is there anything else we can clarify?
>
> **References**:
>
> Yang, Huanrui, Wei Wen, and Hai Li. "Deephoyer: Learning sparser neural network with differentiable scale-invariant sparsity measures." arXiv preprint arXiv:1908.09979 (2019).
>
> Schwarz, Jonathan, et al. "Powerpropagation: A sparsity inducing weight reparameterisation." NeurIPS 34 (2021): 28889-28903.
>
> Lemhadri, Ismael, et al. "Lassonet: A neural network with feature sparsity." Journal of Machine Learning Research 22.127 (2021): 1-29.
>
> Liu, Shiwei, and Zhangyang Wang. "Ten lessons we have learned in the new" sparseland": A short handbook for sparse neural network researchers." arXiv preprint arXiv:2302.02596 (2023).
>
> Vysogorets, Artem, and Julia Kempe. "Connectivity matters: Neural network pruning through the lens of effective sparsity." Journal of Machine Learning Research 24.99 (2023): 1-23.
>
> Dhahri, Rayen, et al. "Shaving Weights with Occam's Razor: Bayesian Sparsification for Neural Networks using the Marginal Likelihood." Sixth Symposium on Advances in Approximate Bayesian Inference-Non Archival Track. 2024.
>
> Balderas, Luis, Miguel Lastra, and José M. Benítez. "Optimizing dense feed-forward neural networks." Neural Networks 171 (2024): 229-241.

---

> > ### Comment · Reviewer_EGQD · 2024-11-15
> > **Response to rebuttal**
> >
> > Thank you for answering my questions; I'm eager to see the revision when it's released. I have one remaining question re: lines 475-476. While it seems plausible to me that the $D=3$ case achieves a compression ratio of 1014, it still seems like $D=2$ (the blue line) performs very comparably. Shouldn't this be referenced as the runner-up rather than SynFlow, especially since the $D=2$ case is not an original contribution of this paper?

---

> > > ### Author Response · Authors · 2024-11-18
> > >
> > > Dear reviewer EGQD,
> > >
> > > We agree with your remaining question and will incorporate it into a first revised submission, which we plan to upload around the midpoint of the discussion period.

---

> > ### Author Response · Authors · 2024-11-20
> > **Revised submission**
> >
> > Dear reviewer EGQD,
> >
> > We would like to thank you again for your helpful comments and have uploaded a revised submission. The revised submission includes the following changes related to your comments:
> >
> > We have revised lines 475-476 related to the description of results in (Q1) to be more nuanced and differentiated. The respective part of our original submission was:
> >
> > >The magnitude of improvement over established pruning methods can be substantial: the best-ranked DWF method achieves 2 to 5 times higher compression than the best-performing comparison method. For VGG-19 on CIFAR100 at 10% tolerance, DWF (D = 4) [sic!] achieves a compression ratio of 1014, surpassing the runner-up (SynFlow at 218) by a factor of almost 5.
> >
> > In the revised submission, we replaced the above by:
> >
> > >The best-ranked DWF model achieves 2 to 5 times the compression ratio of the best pruning method, and surpasses shallow factorization in all but one setting, albeit with smaller improvements ranging from 8% to 298%. For example, DWF with D = 3 reaches a compression ratio of 1014 (1456) for VGG-19 on CIFAR100 (ResNet-18 on CIFAR10) at 10% tolerance, improving by 8% (25%) over D = 2 and by 381% (102%) over the best pruning method SynFlow.
> >
> > Related to your suggestion on decluttering Figures 6 and 7 in (Q3), we would like to thank you again for pointing this out and have removed the compression ratio from Fig. 6 and two of the six non-zero $\lambda$ values from the plots in Fig. 7.
> >
> > We hope these changes are able to adequately address your comments and suggestions. Thank you again for your time and feedback.

---

> > > ### Comment · Reviewer_EGQD · 2024-11-25
> > > **Response to revised submission**
> > >
> > > Thank you for revising your manuscript and for explaining the changes you've made. I appreciate your engagement with my concerns and believe that lines 475-476 now more accurately reflect the results you report in Figure 9 and Table 1.

---

> > > > ### Author Response · Authors · 2024-11-26
> > > > **Response to response to revised submission**
> > > >
> > > > Dear reviewer EGQD,
> > > >
> > > > Thank you for your thoughtful review and for taking the time to assess our (revised) manuscript. We are glad to hear that the revised lines now better align with the results presented in Fig. 9 and Table 1. We believe your comments and suggestions improved the consistency and quality of our submission, and we greatly appreciate your contributions to this process.

---

> ### Author Response · Authors · 2024-11-15
> **Response to weaknesses**
>
> We would like to address the raised critique (small improvements for larger architectures and tasks) in the following:
>
> 1. Besides the mentioned improvements in the accuracy-compression trade-off for $D>2$, deeper factorizations also have the advantage of delaying or avoiding the model collapse observed for $D=2$ in Figures 8,9, and 23, permitting more extreme overall compression ratios.
> 2. The pattern of improvement for $D>2$ over $D=2$ also consistently appears in the additional experiments in Appendix E.2 for ResNet-18, WRN-16-8, and ResNet-34 on TinyImagenet and CIFAR100 (most pronounced for higher compression ratios), corroborating the conclusion that the improvement of DWF over shallow factorizations is not only an artifact of a specific architecture-dataset combination.
> 3. As argued in our response to (Q1), real-world applications do not always imply very large models (outside of computer vision and language modeling), making the study of fully-connected or smaller architectures also relevant for modern machine learning. A clear improvement of DWF in these settings can therefore translate into a significant improvement for such real-world applications.
> 4. As DWF is an integrated sparse learning method and not a pruning-after-training procedure decoupled from optimization, the applicability of DWF extends beyond merely reducing the size of large models. Since DWF directly learns sparse representations during training, it can be also used to enhance the interpretability of the learned models (similar to how the lasso is used in linear models) or, e.g., to overcome catastrophic forgetting (Schwarz et al., 2021).
>
> We hope that these points adequately address the points raised in your comment.
>
> **References**:
>
> Schwarz, Jonathan, et al. "Powerpropagation: A sparsity inducing weight reparameterisation." NeurIPS 34 (2021): 28889-28903.

---

### Official Review · Reviewer_pmzc · 2024-11-03

**Soundness:** 3
**Presentation:** 3
**Contribution:** 3
**Rating:** 8
**Confidence:** 4

**Summary:**

This paper is concerned with sparsity-inducing training of neural networks. The fundamental (pre-existing) technical tool is the transformation of a (non-differentiable) L1-penalized problem, into an “equivalent” (differentiable) L2-penalized problem via a Hadamard-product reparametrization of the weights. This paper introduces Deep Weight Factorization (DFW), an extension of the previous idea to reparametrizations including D \ge 3 Hadamard factors, and shows an equivalence with non-convex sparse regularization. The authors present an analysis of the interplay between DFW and common optimization aspects such as the weight initialization and the choice of step size. Experimentally, the paper includes several comparisons against existing sparsity methods on vision benchmarks, favorably to DWF.

**Strengths:**

S1. The paper is well structured, and the presentation of the ideas is easy to follow.

S2. While the idea of generalizing “weight factorization” to multiple factors is quite natural, the authors do a good job at motivating it, and providing theoretical and empirical insights on their proposed DWF technique.

S3. The authors transparently explore the empirical uncertainties stemming from the change in parametrization. In particular, I appreciated the section on weight initialization: it identifies challenges with traditional initializations, discusses a potential explanation (based on the kurtosis of the distributions) and designs/prescribes a new initialization scheme that is more suitable for use in conjunction with DWF.

S4. The problem of neural network sparsity is very relevant in the current large model context. The simple core principles of the DWF approach, along with the theoretical insights presented by the authors, make it an appealing technique for the deep learning community.

**Weaknesses:**

W1. The paper exclusively focuses on vision-related tasks, of relatively small scale.
  * The submission does not provide results on ImageNet-scale vision tasks.
  * The submission does not include any experiments on language or image generation.

W2. Theorem 1 presents an equivalence between the DWF and “original” non-convex penalized problem, claiming that they “have the same global and local minima”. However, the current discussion does not explore whether the relative quality of a local DWF minimum is or not comparable to the quality of its related minimum for the original problem. In other words, let $\hat{\omega}_1 \odot \ldots \odot \hat{\omega}_D = \hat{w}$ be a local minimum of the DWF problem. How good of a local minimum is $\hat{w}$ for the original problem?

W3. I found the visual presentation of experimental results –in particular, Fig 6, and Fig 7 bottom– too crowded, making it difficult for the reader to extract the relevant information.

**Questions:**

The questions marked with * are most important.

*Q1. I think the contributions of the work are sufficient for publication, but I would like to know the authors’ position on W1.

*Q2. In Fig 7 (and related plots in App E), what is the step-size selection protocol? Is it tuned individually for each depth?

Q3. For Sec 5.1, why is the magnitude pruning without fine-tuning protocol justified? Especially at high compression levels.

*Q4. The paper mentions explicitly (L151) “in principle [...] the factorization can also be selectively applied to arbitrary subsets of the parameters w” in the context of unstructured sparsity (in implicit contrast with structured sparsity). Orthogonal to the structured vs unstructured discussion, what is the relationship between DWF and the type of layers involved? Do certain layers benefit or suffer disparately with different factorization depths?

Q5A. I was pleasantly surprised to read the conclusions of the runtime analysis in Sec 5.2 and App I.2. Given (1) the claimed low computational overhead induced by DWF and (2) the change in optimization dynamics caused by DWF, what do the authors think could be the effects of this factorization idea on the training of non-regularized objectives?

*Q5B. As I stated in W1, the paper focuses on relatively small-scale experiments. How do the authors think the experimental conclusions of their runtime analysis would transform at extremely high parameter counts (>1B)?

Q5C. Along the same line of high parameter counts as in Q2B, do we need to “persistently hold” a (multi-factor) reparametrization during training? During the forward phase, it makes no difference to use the collapsed or not-collapsed parametrization, the only difference is for the gradient application. Would it be possible to hold an “ephemeral reparametrization”, sampled on demand during the parameter update? The idea would be to locally simulate the reparameterized GD dynamics, without requiring the persistent footprint of multiple model copies.

Q6. I noticed the absence of optimization “momentum” (as in Nesterov/Polyak) in the experimental details (as presented in App G.2). I understand that this could be a consequence of the specific architectures and tasks considered in the submission. However, I would like the authors to expand on any potential consequences/challenges of using DWF with momentum-based optimizers.

*Q7. The paper begins almost exclusively concerned with the case of the Lasso regularizer (q=1). Eventually, the theoretical connection presented in Thm 1 is established with a fractional regularized (q=2/D). Is this departure from the Lasso case “necessary”, or merely an artifact of the analysis/proof? In other words, if the problem I truly care about is the Lasso, can I still take advantage of the DWF-induced dynamics?

Minor comments/suggestions (No response expected)
* In Fig 7 (and more generally App E), the case “Depth=1” would be a useful baseline to understand how much of the improvement is afforded by DWF.
* There are several instances in the paper where the authors mention that “Appendix X contains information on the relationship between Y and Z”, but do not highlight any core insight on the main paper. I believe the current appendices are quite valuable, and the submission could benefit from (very!) briefly summarizing said key discoveries in the main body.
* Given the many appendices present, the authors may consider including a table of contents for the appendix to aid navigation.

---

> ### Author Response · Authors · 2024-11-15
> **Reseponse to weaknesses**
>
> Dear reviewer pmzc,
>
> We would like to thank you for your comments and interesting questions and respond to the weaknesses and questions in separate comments.
>
> - (W1) We agree that studying DWF in very large models would be an interesting avenue worth exploring, but the focus of our work lies in the analysis of the sparsification and training dynamics on a range of different architectures. Moreover, the scope of our experimental evaluation is in line with many other works on sparse learning (e.g., Lee et al., 2019; Frankle et al., 2021; Lu et al., 2022; Dhahri et al., 2024).
> - (W2) Thank you for this question, which we might be able to clarify: the “quality” of the collapsed parameter $\hat{{w}}$ in the original problem will be the same as the quality of ($\hat{\omega_1},...,\hat{\omega_{D}}$) in the factorized formulation, since both induce functionally the same network structure, i.e., identical predictions, losses, and generalization. A more interesting question is whether the DWF formulation yields better solutions than direct optimization of the original (regularized) objective using SGD, which we can answer in the affirmative using the experiment in Section 5.1 (Fig. 1). The DWF approach can be understood as a differentiable optimization tool for $L_{2/D}$ regularization, so the question becomes which optimizer yields better solutions, not if equivalent solutions instantiating the same effective network differ in their quality (which they don’t).
> - (W3) We thank you for your suggestion and agree with your concern, also raised by another reviewer. We will remove the compression ratio (CR) from Fig. 6 and reduce the number of displayed $\lambda$ values in Fig. 7 to declutter both plots in a revised version of our submission.
>
> **References**:
>
> Lee, N., T. Ajanthan, and P. Torr. "SNIP: single-shot network pruning based on connection sensitivity." International Conference on Learning Representations. 2019.
>
> Frankle, Jonathan, et al. "Pruning Neural Networks at Initialization: Why Are We Missing the Mark?." International Conference on Learning Representations. 2021.
>
> Lu, Miao, et al. "Learning pruning-friendly networks via frank-wolfe: One-shot, any-sparsity, and no retraining." International Conference on Learning Representations. 2022.
>
> Dhahri, Rayen, et al. "Shaving Weights with Occam's Razor: Bayesian Sparsification for Neural Networks using the Marginal Likelihood." Sixth Symposium on Advances in Approximate Bayesian Inference-Non Archival Track. 2024.

---

> ### Author Response · Authors · 2024-11-15
> **Response to questions**
>
> - (Q1) We thank you for your evaluation and refer to our reply to (W1).
> - (Q2) The step-size selection protocol is described in lines 1791 to 1799. For the fully-connected and convolutional LeNet architectures we set the LR to 0.15, while for the ResNet and VGG applications, we select the best-performing LR from a grid of values for each depth, architecture, and dataset using a small fixed $\lambda$. Departing from that, for the additional experimental results in Fig. 23 of the Appendix, we did not tune the LR for each architecture and dataset to reduce computation overhead, and only varied it by depth $D$.
> - (Q3) We specifically exclude iterative pruning and re-training approaches to make for a fairer comparison including only approaches without pruning and additional re-training (as argued in Liu et al., 2023). Secondly, some works argue that the advantage of sparsification techniques should not be attributed to the fine-tuning phase (e.g., Wang et al., 2023), as this complicates the comparison significantly, needing to account for potentially different optimal finetuning schedules for each method. Third, the DWF approach corresponds to a differentiable formulation of sparse $L_{2/D}$ regularization, and can therefore be arbitrarily combined with post-hoc pruning approaches and re-training. In Appendix E.6, we briefly explore the combination of DWF training with post-hoc magnitude pruning and re-training, indicating a superior accuracy-sparsity trade-off when combining methods.
> - (Q4) In our experiments we did not find that $D$ impacts layer types differently (see Fig. 10), but that their relative location (near input/output or in the middle) is more important.
> - (Q5a) Thank you for this Interesting question. The local and global minima using unregularized weight factorization can be shown to be the same as for the original unregularized objective (Lemmas 3 and 4 in proof of Thm. 1 with $\lambda=0$). However, small initialization scales or large LRs could induce an implicit bias toward sparse minima in overparametrized problems, as studied, e.g., in Woodworth et al. (2020) for factorized linear models. However, the implicit bias derived in their work only yields implicit $L_1$ regularization for any depth $D \geq 2$ instead of the beneficial non-convex $L_{2/D}$ regularizer obtained with $L_2$ regularized DWF.
> - (Q5b) We expect the runtime conclusions to carry over to larger models *ceteris paribus*. However, models with >1b parameters are usually transformer architectures instead of fully-connected or convolutional networks, and different behavior could arise.
> - (Q5c) Thank you for this interesting suggestion. In principle, we don’t need to persistently hold the full factorized parametrization throughout training, which could be collapsed into fewer factors or even $D=1$ for the forward pass or at different training stages. However, as shown in App. I.2, the computational overhead using DWF is surprisingly low and nowhere near an expected $D$-times increase.
> - (Q6) Table 3 in App. G.2 lists "Mom." for the momentum parameter, which is used consistently for all models included in our work with a value of $0.9$.
> - (Q7) The departure from $L_1$ to non-convex $L_{2/D}$ regularization is indeed an explicitly desired result, as the fractional quasi-norm regularizer is a closer approximation of the intractable $L_0$ penalty, promising higher sparsities at the same performance (cf. Xu et al., 2010; Wen et al., 2018). Typically, $L_1$ regularization is motivated as the tightest convex relaxation of the $L_0$ penalty (which we "actually care about” in sparse learning), trading off the better performance of non-convex penalties against easier convex optimization.
>
> Does this answer your questions and is there anything else we can clarify?
>
> **References**:
>
> Xu, Zongben, et al. "L 1/2 regularization." Science China Information Sciences 53 (2010): 1159-1169.
>
> Wen, Fei, et al. "A survey on nonconvex regularization-based sparse and low-rank recovery in signal processing, statistics, and machine learning." IEEE Access 6 (2018): 69883-69906.
>
> Woodworth, Blake, et al. "Kernel and rich regimes in overparametrized models." CoLT. PMLR, 2020.
>
> Liu, Shiwei, and Zhangyang Wang. "Ten lessons we have learned in the new" sparseland": A short handbook for sparse neural network researchers." arXiv preprint arXiv:2302.02596 (2023).
>
> Wang, Huan, et al. "Why is the state of neural network pruning so confusing? on the fairness, comparison setup, and trainability in network pruning." arXiv preprint arXiv:2301.05219 (2023).

---

> ### Comment · Reviewer_pmzc · 2024-11-18
> **Response to rebuttal**
>
> I thank the authors for their thorough responses. I appreciate them taking the time to answer all my questions, even the ones not marked as "important". I retain my original score for this submission.

---

> > ### Author Response · Authors · 2024-11-20
> > **Revised submission**
> >
> > Dear reviewer pmzc,
> >
> > We are glad to hear our response was appreciated and your score retained. For completeness, we would like to thank you for suggesting including a table of contents for the appendix, which we have now incorporated in our uploaded revised submission.
> >
> > Thank you again for the insightful comments and valuable feedback.

---

### Official Review · Reviewer_h7VL · 2024-11-04

**Soundness:** 3
**Presentation:** 3
**Contribution:** 3
**Rating:** 8
**Confidence:** 4

**Summary:**

This paper extends existing (shallow) weight factorization techniques to deep weight factorization(DWF), and demonstrates the equivalence of DWF  with the $L_\frac{2}{D}$ (where $D$ denotes the depth of the factorization) regularization, thus potentially achieving higher sparsity compared to its shallow counterpart. The authors also provided enhanced strategies for  initialization and  learning rate to ensure the performance of DWF. Additionally, they analyzed the impact of DWF on training dynamics and optimization by empirical experiments, which show that DWF outperforms the shallow weight factorization and several existing pruning approaches.

**Strengths:**

1. The author presented a generalization of the shallow weight factorization  to overcome the non-differentiability of $L_1$ regularization and proved its equivalence to the $L_\frac{2}{D}$ regularization.
2. The strategies like initialization and large learning rate ensure the application of DWF.
3. The experiments show that DWF outperforms the original weight factorization and several pruning methods.1. The motivation of the paper is kind of vague. Why extending the shallow weight factorization in [1] to the deep one could potentially result in performance gain?  One  suggestion is: building on the equivalence of shallow weight factorization with the $L_1$ regularization, and taking into account that $L_p (p<1)$ regularization would potentially yield higher sparsity thanks to it is closer to the $L_0$ regularization compared to $L_1$ regularization, it is kind of natural to conjecture that deep weight factorization is equivalent to the $L_\frac{2}{D}$ (where $D$ denotes the depth of the factorization) regularization.

**Weaknesses:**

1. The motivation of the paper is kind of vague. Why extending the shallow weight factorization in [1] to the deep one could potentially result in performance gain?  One  suggestion is: building on the equivalence of shallow weight factorization with the $L_1$ regularization, and taking into account that $L_p (p<1)$ regularization would potentially yield higher sparsity thanks to it is closer to the $L_0$ regularization compared to $L_1$ regularization, it is kind of natural to conjecture that deep weight factorization is equivalent to the $L_\frac{2}{D}$ (where $D$ denotes the depth of the factorization) regularization.

2. In Figure 1, does the different compression ratios of vanilla $L_1$ regularization relevant to different values of $\lambda$? To my knowledge, achieving the highest test accuracy at various compression ratios often requires selecting different $\lambda$ values. If the value of $\lambda$  is fixed across different compression ratios, it would be beneficial to search $\lambda$  at different compression ratios.

3. Considering DWF is a pruning-before-training method, it is not much fair to only conduct comparison with those pruning-before-training methods, such as SNIP and SynFlow. It is suggested to add comparison with some high-performance pruning-after-training methods, such as [2] and [3] (pruning with $L_1$ regularization).

[1] Liu Ziyin and Zihao Wang. spred: Solving l1 penalty with sgd. In International Conference on Machine Learning, pp. 43407–43422. PMLR, 2023.

[2]Renda, A., Frankle, J., and Carbin, M. Comparing rewinding and fine-tuning in neural network pruning. arXiv preprint arXiv:2003.02389, 2020. 3, 4, 6, 9.

[3]Zhang, Q., Zhang, R., Sun, J., & Liu, Y. (2023). How Sparse Can We Prune A Deep Network: A Fundamental Limit Viewpoint. *arXiv preprint arXiv:2306.05857*.

**Questions:**

1. The equivalence of DWF to the $L_p$ regularization largely relies on the fact that balanced factorization (or zero factor misalignment) is optimal. Would you provide some intuitive explanation that regularization by the balanced DWF would imply (or encourage) the sparsity of weights?

2. Whether the training for DWF could achieves a global minima, given that the weight factorization approach is  non-convex?

---

> ### Author Response · Authors · 2024-11-15
> **Response to questions**
>
> Dear reviewer h7VL,
>
> We thank you for your insightful comments and interesting questions, which we will respond to below. A response to the weaknesses is included in a separate comment.
> ### Question 1
> >The equivalence of DWF to the $L_p$ regularization largely relies on the fact that balanced factorization (or zero factor misalignment) is optimal. Would you provide some intuitive explanation that regularization by the balanced DWF would imply (or encourage) the sparsity of weights?
>
> Overparameterization using the DWF (without additional $L_2$ regularization) results in infinitely many factorizations of each original weight. Fig. 3 shows the possible factorizations for weight values 0 and 0.25 using two factors, but the same reasoning applies to deeper factorizations. All possible factorizations induce the same collapsed network and therefore leave the loss function unchanged. However, as Fig. 3 illustrates, the possible factorizations differ in the distance to the origin and thus in their (squared) Euclidean norm. Hence, if we use additional $L_2$ regularization, only the minimum-norm factorizations can be optimal, as otherwise, we could always decrease the $L_2$ penalty by picking a more balanced factorization while leaving the unregularized loss unchanged. For non-zero weights, the possible factorizations form a hyperbola (Fig. 3), implying there are two minimum-norm factorizations located at the vertices of the hyperbola. Points on the vertices of a rectangular hyperbola are always balanced, implying that the minimum-norm factorizations are given by $(\sqrt{w},\sqrt{w})$ and $(-\sqrt{w},-\sqrt{w})$ for positive weights $w$ and $(\sqrt{|w|},-\sqrt{|w|})$ and $(-\sqrt{|w|},\sqrt{|w|})$ for negative weights. If we evaluate the $L_2$ penalty at the balanced factorizations, it turns into $2|w|$ which is the sparsity-inducing $L_1$ penalty multiplied by $2$. For more than two factors, the $L_2$ penalty at balanced factorizations reduces to a sparsity-inducing $L_{2/D}$ penalty on the collapsed weight. This serves as a lower bound of the $L_2$ penalty. Once a balanced factorization is achieved, the factor $L_2$ penalty reduces to an $L_{2/D}$ penalty on $w$, and the optimization encourages sparse solutions. Furthermore, because balancedness is an “absorbing state” of SGD optimization (Lemma 6), this state becomes “locked” and the $L_2$ regularizer remains in its sparsity-inducing reduced form for all future iterations.
> We hope this provides some intuition as to why balanced factorizations with $L_2$ regularization encourage sparsity in the collapsed weights and will **include a dedicated section for this intuition in the Appendix** of our revised submission.
> ### Question 2
> >Whether the training for DWF could achieves a global minima, given that the weight factorization approach is non-convex?
>
> Thank you for this interesting question. Particularly in deep learning applications, the loss landscape is highly non-convex (for vanilla networks), and all we can usually hope for is finding good local minima, which are provably identical for vanilla and factorized networks (Theorem 1). Moreover, the $L_{2/D}$ regularizer itself is non-convex for $D>2$, and finding global minima is known to be NP-hard (e.g., Ge et al., 2011; Chen et al., 2017). Still, empirically, non-convex regularization often offers better sparsity-performance trade-offs than convex $L_1$ regularization (Xu et al., 2010; Wen et al., 2018). In our experiments, these findings are reinforced for deep learning showing that deep weight factorization often outperforms shallow weight factorization.
>
> Does this answer your questions and is there anything else we can clarify?
>
> **References**:
>
> Chen, Yichen, et al. "Strong NP-hardness for sparse optimization with concave penalty functions." International Conference on Machine Learning. PMLR, 2017.
>
> Ge, Dongdong, Xiaoye Jiang, and Yinyu Ye. "A note on the complexity of L p minimization." Mathematical programming 129 (2011): 285-299.
>
> Xu, Zongben, et al. "L 1/2 regularization." Science China Information Sciences 53 (2010): 1159-1169.
>
> Wen, Fei, et al. "A survey on nonconvex regularization-based sparse and low-rank recovery in signal processing, statistics, and machine learning." IEEE Access 6 (2018): 69883-69906.

---

> ### Author Response · Authors · 2024-11-15
> **Response to weaknesses**
>
> - (W1) We thank you for this suggestion and fully agree with this interpretation. We will integrate this into our main text in a revised submission to provide a clearer and more intuitive motivation.
> - (W2) We agree with your intuition that achieving the highest test accuracy at different compression ratios requires selecting different $\lambda$ values. The left plot of Fig. 1 shows the inherent sparsity obtained from a range of $\lambda$ values. In the right plot, each of these models is further pruned to fixed increasing compression ratios (using weight magnitude as the pruning criterion) until the model degrades to random chance performance. Finally, among all pruned models for each compression ratio value, the model with that $\lambda$ which yields the highest test accuracy is chosen. This means the right plot with additional post-hoc pruning already considers the optimal $\lambda$ for each compression ratio. We will make this set-up more clear in a revised version of our submission.
> - (W3) We thank you for mentioning both references. Reference [2] includes a re-winding and re-training step after pruning, which complicates the comparison set-up tremendously. As argued, e.g., in Wang et al. (2023), the performance of a sparsification method should not be based on the performance after re-training, as they could significantly alter the ranking of pruning methods simply by adjusting the re-training/finetuning learning rate schedule. As the main focus of our submission is not a detailed benchmark study of pruning methods, we exclude iterative sparsification schemes including re-training. However, we did conduct an experiment studying whether DWF can be combined with pruning and re-training in Appendix E.6, observing a performance boost from re-training. Reference [3], on the other hand, studies $L_1$ regularized one-shot magnitude pruning, which does not entail the same comparison problem of iterative pruning approaches. Although our main experimental evaluation includes magnitude pruning without direct $L_1$ regularization, the method studied in [3] corresponds to the blue line in the right plot of Fig. 1 (SGD+$L_1$ with additional magnitude pruning), showing that the differentiable formulation of $L_1$ regularization leads to better sparsity-accuracy trade-offs after magnitude pruning than direct $L_1$ regularization (and differentiable $L_{2/3}$ regularization leads to better trade-offs than differentiable $L_1$). As DWF constitutes a differentiable approach to optimize $L_{2/D}$ regularized objectives, it can be used as an easier-to-optimize replacement for direct $L_1$ regularization and combined with any pruning-after-training method, including re-training or not.
>
> Does this clarify our experimental procedure and resolve potential misunderstandings of the setup in Fig. 1? If not we will gladly respond to further questions and suggestions.
>
> **References**:
>
> Wang, Huan, et al. "Why is the state of neural network pruning so confusing? on the fairness, comparison setup, and trainability in network pruning." arXiv preprint arXiv:2301.05219 (2023).

---

> > ### Comment · Reviewer_h7VL · 2024-11-22
> > **Response to the Rebuttal**
> >
> > I appreciate the authors' detailed explanations and corresponding revisions. My new comments are as follows:
> >
> > Regarding the performance comparison between WF and the $L_1$ regularizer in Fig. 1, I doubt the curve of "SGD + $L_1$" was not the optimal one, because in [1] it has been demonstrated that with careful selection of the regularization parameter $\lambda$, $L_1$ regularization can actually achieve the $\textit{fundamental limit}$ of one-shot post-training network pruning.
> >
> > In view of the above, I think the major role of WF (both shallow and deep) needs to reconsidered and restated, since in both this paper and [2] the motivation is to address the non-differentiability of $L_1$ norm, however, as can be seen from the left plot of Fig.1, there are actually no weights that are exactly equal to zero, which is the only non-differentiable point  for $L_1$ norm. In this sense, the non-differentiability of $L_1$ norm in the origin seems actually not a big problem. Moreover, combining this observation and the above-mentioned conclusion in [1],  it seems the main advantage of WF  lies in that its less sensitivity to the  regularization parameter $\lambda$ as compared with  the $L_1$ regularization.
> >
> >
> >
> > [1] Zhang, Q., Zhang, R., Sun, J., & Liu, Y, How Sparse Can We Prune A Deep Network: A Fundamental Limit Viewpoint, NeurIPS 2024  https://openreview.net/pdf?id=IAAPhOLhcX
> > [2] Liu Ziyin and Zihao Wang. spred: Solving l1 penalty with sgd. In International Conference on Machine Learning, pp. 43407–43422. PMLR, 2023.

---

> ### Author Response · Authors · 2024-11-20
> **Revised submission**
>
> Dear reviewer h7VL,
>
> We would like to thank you again for your helpful comments and have uploaded a revised submission. The revised submission includes the following changes related to your comments:
>
> We have included the following motivation in lines 198-200 of our revised submission to reflect the intuition provided in (W1):
>
> >The $L_{2/D}$ penalty in Eq. (4) becomes non-convex and increasingly closer to the $L_0$ penalty for $D>2$, permitting a more aggressive penalization of small weights than $L_1$ regularization (Frank & Friedman, 1993).
>
> Regarding the discussion about Fig. 1 (W2), we have made the experimental set-up for Fig. 1 more clear in Section 5.1. Specifically, our revised submission now includes
>
>
> >The left plot in Fig. 1 (page 1) shows the tradeoff between performance and inherent sparsity (before pruning) across a range of $\lambda$ values, confirming prior findings on the limitations of vanilla $L_1$ optimization. [...]. In the right plot of Fig. 1, we subsequently apply post-hoc magnitude pruning to each of the models at increasing compression ratios (without fine-tuning) until reaching random chance performance and use the best-performing pruned model at each fixed compression ratio to obtain the pruning curves.
>
> Addressing the intuitive explanation for the sparsity-inducing effect of $L_2$ regularization and weight factorization requested in (Q1), we have included a dedicated section (Appendix B in the revised submission) containing an in-depth explanation of the geometric intuition underlying the sparsity-promoting effects of $L_2$ regularized (minimum-norm) weight factorization, as illustrated in Fig. 3.
>
> We hope these changes are able to adequately address your comments and suggestions and thank you again for your valuable feedback.

---

> ### Author Response · Authors · 2024-11-26
> **Reseponse to response to the rebuttal**
>
> Dear reviewer h7VL,
>
> Thank you very much for responding to our rebuttal.
>
> > Regarding the performance comparison between WF and the $L_1$ regularizer in Fig. 1, I doubt the curve of "SGD + $L_1$" was not the optimal one
>
> We thank you for sharing your concern. To ensure that our results were not just due to a suboptimal choice of $\lambda$s, we reran the experiment using **100 (instead of 20)** logarithmically spaced $\lambda$ values between 1e-6 (where all methods achieve full performance) and 5e-2 (where all methods degrade to random guessing) in the left plot of Fig. 1. For each method, we then magnitude-prune the 100 resulting models to increasing CRs of [50, 70, 80, 90, 100, 200, 300, 400, 500, 600, 700, 800, 900, 1000, 2000, 5000, 10000]. For each CR, we record the highest test accuracy among the 100 models to obtain the pruning curves in the right plot of Fig. 1. The extended results can be found in Fig. 1 of our newly revised submission, with qualitatively identical conclusions to our initial experiment.
>
> > because in [1] it has been demonstrated that with careful selection of the regularization parameter $\lambda$, $L_1$ regularization can actually achieve the fundamental limit of one-shot post-training network pruning.
>
> We thank the reviewer for sharing reference [1] again and have added this reference to our literature section in the appendix as follows:
>
> “Recently, Zhang et al. (2024) establish theoretical bounds on network prunability using convex geometry, showing that the fundamental one-shot pruning limit without sacrificing performance is determined by weight magnitudes and the sharpness of the loss landscape, providing a unified framework to help explain the effectiveness of magnitude-based pruning. They empirically show that $L_1$ regularized post-hoc magnitude pruning approximately matches the derived pruning limit before performance degrades significantly.”
>
> We further highlight that all three methods shown in the right plot of Fig. 1 are nearly indistinguishable up to a compression ratio of ~200 (or 99.5% sparsity) before their performances diverge. The fundamental limit result in [1] identifies the point of degradation, however, methods might still differ in their full sparsity-accuracy tradeoffs beyond that limit point.
>
> > as can be seen from the left plot of Fig.1, there are actually no weights that are exactly equal to zero, which is the only non-differentiable point for $L_1$ norm. In this sense, the non-differentiability of $L_1$ norm in the origin seems actually not a big problem.
>
> We thank the reviewer for this comment and would like to note that there might be a misunderstanding. The weights not being zero does not mean the non-differentiability is benign. On the contrary, due to the properties of the $L_1$ penalty we know that theoretically the solutions should be sparse, so if SGD does not find sparse solutions, this points to an optimization problem. As a result of the non-differentiability, the optimization starts to oscillate and will randomly stop at some value, which can be far from 0. Below a simple 2D example showing the first and last 5 iterations of a 1000-step GD optimization of an $L_1$ regularized linear model (without WF) where one solution ($\beta_2$) is 0:
>
> | Iteration | $\beta_1$   | $\beta_2$   |
> |-----------|----------|----------|
> | 0         | 0.1000   | 0.1000   |
> | 1         | 0.0975   | -0.0825  |
> | 2         | 0.0951   | 0.0685   |
> | 3         | 0.0929   | -0.0573  |
> | 4         | 0.0907   | 0.0483   |
> | ...       | ...      | ...      |
> | 996       | 0.0500   | 0.0125   |
> | 997       | 0.0500   | -0.0125  |
> | 998       | 0.0500   | 0.0125   |
> | 999       | 0.0500   | -0.0125  |
> | 1000      | 0.0500   | 0.0125   |
>
>
> As can be seen, the GD routine starts oscillating for $\beta_2$ and does not converge to a value significantly closer to 0 than $\pm$ 0.0125. Therefore, despite none of the final weights being 0, GD does not produce reliable results.
>
> > it seems the main advantage of WF lies in that its less sensitivity to the regularization parameter $\lambda$ as compared with the $L_1$ regularization.
>
> Thank you for this additional comment. Due to the equivalence of the two optimization problems, we can, a priori, expect the sensitivity to $\lambda$ to be the same for both the vanilla and differentiable $L_1$ formulations. This is confirmed in, e.g., Ziyin and Wang (2023), who show in their Fig. 3 that $L_2$ regularized WF of a linear model yields exactly the lasso solution for the corresponding $\lambda$ (i.e., WF has the same sensitivity to $\lambda$ as direct $L_1$ regularization).
>
> We hope that this better clarifies the motivation of our work and adequately addresses the additional comments.
>
> **References**:
>
> Liu Ziyin and Zihao Wang. “spred: Solving l1 penalty with sgd”. ICML. 2023.

---

> > ### Comment · Reviewer_h7VL · 2024-11-28
> > **Response to the Clarifications**
> >
> > I greatly appreciate the additional experiments the authors made to verify the optimality of the $L_1$ + SGD curve in Fig. 1, which shows that when the compression ratio is very high (for example, above $10^3$), the WF method  does outperform the vanilla $L_1$ regularization. Nevetheless, it is worth noting that in the regime of not that high compression ratio (say, below $10^2$) ,  which is of more practical importance since the pruning there incurs little or very insignificant performance degradation, the sparsity-accuracy curves of vanilla $L_1$ and WF are almost indistinguishable. In this sense, adding a zoom-in plot in the regime  of compression ratio 1~100 for the right plot in Fig.1 would be helpful to provide the readers a better understanding towards the comparison among the algorithms.
> >
> > Besides, I totally agree with the authors that the non-differentiability of $L_1$ norm might cause the oscillation problem in SGD.  My real point  is that in the compression ratio regime of practical importance (foe example, below 100) , that problem seems not quite severe, as evidenced by the fact that by careful selection of $\lambda$, $L_1$ regularizer can perform almost identically with (if not better than) the WF methods. In light of this, the main advantage of WF over vanilla $L_1$ method seems lying in the less sensitivity (thus less searching complexity) of $lambda$ of the former.
> >
> > Considering the thorough explanations and revisions made by the authors,  I'd like to raise the score to 7.

---

> ### Author Response · Authors · 2024-11-28
> **Reseponse to response to clarifications**
>
> Dear reviewer h7VL,
>
> Thank you for your ongoing engagement and appreciation of our additional experiments. We agree with your intuition that the problem of non-differentiability is less severe at lower compression ratios and becomes increasingly important at higher compression ratios (evidenced by the increasing divergence between SGD + vanilla $L_1$ and shallow WF with D=2 at higher CRs). We also fully agree that adding a zoom-in plot of the low sparsity regime would be helpful in providing an improved comparison among the algorithms. To this end we have included such a zoom-in plot in Fig. 1 of a newly revised submission, zooming in on the lowest compression ratios we post-hoc pruned to in our experiment (50,70,80,90,100). The inset shows several interesting observations:
> - at the lowest tested CRs of 50 and 70, all three methods are indistinguishable and perform within one standard deviation. Additionally, since all three methods achieve the same baseline performance before post-hoc pruning, this relationship can be extrapolated to non-tested lower CRs.
> - the pruning curves for SGD+$L_1$ and shallow WF (D=2) are virtually identical in that regime, which extends up to a CR of about 200 looking at the overall plot.
> - starting at a CR of 80, deep WF (D=3) has a slight but visible advantage over the other two methods that increases with higher compression ratios.
>
> Further, we would like to add that the described relationship between shallow WF (D=2) and SGD+$L_1$ has also been confirmed in previous literature (Ziyin and Wang, 2023), where both methods (and some other comparison methods) show closely matching pruning curves in the low-sparsity regime before diverging at higher compression ratios.
>
> Finally, our submission highlights at several places that the main advantage/focus of DWF is at higher compression ratios, e.g.,
>
> >We observe that the D = 2 factorization excels [...]  below a compression ratio of 100, while D>2 shows enhanced resilience to performance degradation and delayed model collapse at more extreme sparsity levels.
>
> >These tolerance levels are suitable for testing the medium to high sparsity regimes we focus on.
>
> We thank you again for your helpful suggestion and hope that our response and the corresponding revision might change your inclination to lean more towards an 8 than retaining a 6.
>
> **References**
>
> Liu Ziyin and Zihao Wang. “spred: Solving l1 penalty with sgd”. ICML. 2023.

---

### Official Review · Reviewer_dKGU · 2024-11-04

**Soundness:** 3
**Presentation:** 2
**Contribution:** 1
**Rating:** 5
**Confidence:** 4

**Summary:**

This paper introduces deep weight factorization, an extension of shallow weight factorization for neural network sparsification. The key idea is to decompose network weights into D factors (D≥2), allowing the use of standard $L_2$ regularization to achieve a similar optimization problem to that of the regularized $L_p$ loss, while avoiding smoothness problems. The authors study the performance of this algorithm as well as analyze the dynamics and representations learned during sparsification.

**Strengths:**

The paper has some strengths:

1) The topic of sparsification during training is of great interest with growing model sizes.
2) The paper is well-written and presented.
3) The equivalence between the different regularization schemes is clearly explained and shown.
4) Section 4 is particularly interesting to me, as I am not aware of other studies which attempt to analyze the sparsification dynamics, or attempt to interpret unstructured sparsification algorithms.
5) The ideas presented in the paper are very good, but unfortunately suffer from a lack of novelty as I discuss in the weaknesses section.

**Weaknesses:**

The main weakness of this paper is that, unfortunately, the method proposed has already been implemented before with minimal difference, both in its current form based on Hadamard products [1] for linear networks and in a simpler completely equivalent form [2] incorporated directly to weight decay for any $p$-norm, including $2/d$.

Therefore the paper offers no real novelty besides the analysis of the onset of sparsity, and since that is not the main focus of the paper it does not justify acceptance. I would suggest the authors reframe the work as an exploratory study of the dynamics of sparsification in deep networks, and submit to another venue.

Minor weaknesses:

1) The datasets and tasks are not studied at large scale, which is the most interesting setting in which sparsification should be performed. In particular, focusing either on small models or too simple data results in untrustworthy results when scaling up. Since this is not the first time this method is proposed, a potential avenue would be studying these types of $L_p$ methods in very large models, or for fine tuning.
2) The baseline of random pruning is not a useful baseline, basic magnitude pruning during training should at least be explored.

**References:**

[1] *Representation Costs of Linear Neural Networks: Analysis and Design*, Zhen Dai, Mina Karzand, Nathan Srebro, NeurIPS 2021 (https://openreview.net/forum?id=3oQyjABdbC8)

[2] *Decoupled Weight Decay for Any p Norm*, N. Outmezguine, N. Levi (https://doi.org/10.48550/arXiv.2404.10824)

**Questions:**

As stated in the weaknesses section,  I would suggest the authors reframe the work as an exploratory study of the dynamics of sparsification in deep networks and learned representations, and submit to another venue, after conducting a more thorough literature survey.

---

> ### Author Response · Authors · 2024-11-13
> **Response to major weakness**
>
> Dear reviewer dKGU,
>
> We would like to thank you for your comments and for bringing references [1] and [2] to our attention. We appreciate the opportunity to clarify the differences between our work and both references. We would like to mention that the ICLR reviewer guidelines explicitly state that in the case of unpublished papers, including arXiv papers (such as [2]), the authors may be excused for not discussing these references. As detailed below, we politely disagree with the review’s conclusion and maintain that both references differ significantly from our work in important aspects. However, as related studies, we will include them in the further related literature section. A response to the minor weaknesses is included in a separate comment.
>
> Differences to [1]:
>
> - Analysis of the representation cost (squared $L_2$ norm of the weights) of a **linear predictor** under **specific linear neural network parametrizations** is a related but significantly different topic of study. Our approach, instead, aims to induce differentiable sparsity regularization in arbitrary non-linear networks.
> - Representation cost analysis is based on the equivalence of global minima. Our results, however, extend to the equivalence of **all (local) minima**, which is an important optimization property ensuring no spurious solutions are created.
> - [1] conducts a **purely theoretical analysis** for a restricted problem class of overparametrized linear models. In contrast, our work focuses not only on the theoretical equivalence of minima under Deep Weight Factorization (DWF) of arbitrary learning models but, importantly, also **investigates how to successfully train factorized networks** by providing a novel initialization scheme and a detailed analysis of the training dynamics. The relevant result in [1] states that the representation cost of a linear predictor using a diagonal linear network (essentially a factorized linear model) is equal to the $L_{2/D}$ quasi-norm of the predictor and is obtained as a special case of our Theorem 1 applied to a linear model. Apart from that result, our work is fundamentally different from [1] in both scope and analysis. In particular, [1] does not discuss how to use the representation cost analysis for differentiable sparse learning.
> - Our focus on details such as initialization and learning rates in addition to our more general theoretical results ensures that they translate into a practical method that is readily usable instead of the sole theoretical focus of [1].
>
> Differences to [2]:
> - The pWD method proposed in [2] uses a different variational expression of non-convex $L_{2/D}$ regularizers based on the so-called Eta trick (Bach, 2012). While this approach is closely related to the variational expression we use in our work (cf. Poon and Peyré, 2021), it causes divergent gradients of the auxiliary variables for vanishing weights and requires proximal methods for stable optimization.
> - [2] proposes optimization of their overparametrized problem using Alternating Convex Search, whereas our goal is a method that is amenable to straightforward stochastic gradient descent (SGD). In particular, our motivation is to provide a fully differentiable formulation of $L_{2/D}$ regularized objectives that facilitates seamless integration with other methods and usability by the research community **without deviating from standard SGD optimization**.
> - In contrast to [2], we analyze 1) the phased training dynamics, 2) the evolution of model weight norms, 3) layer-wise sparsity, and 4) the onset of sparsity, which indeed is one major focus of our submission.
> The alternative variational expression of the $L_{2/D}$ regularizer in pWD and its proximal optimization induce different gradients and therefore learning dynamics, and thus constitutes a distinct approach besides its similarities to DWF at first sight.
>
> **Summary**:
> While both [1] and [2] are indeed related to our work, we believe there are major differences and disagree with the claim the main contributions of our work are already encapsulated in the two references. In particular, the scope and focus of [1] are only loosely connected to our study, differing in theoretical versus practical emphasis and the restriction of [1] to linear models and global minima. On the other hand, [2] employs proximal alternating optimization, whereas we explicitly focus on differentiability and use straightforward SGD.
>
> We hope this clarifies the distinctions and highlights the novelty of our work. We would like to thank you again for pointing out these references. We will meticulously incorporate them into the related literature and highlight their importance while also pointing out differences.
>
> References:
>
> Bach, Francis, et al. "Optimization with sparsity-inducing penalties." Foundations and Trends® in Machine Learning (2012): 1-106.
>
> Poon, Clarice, and Gabriel Peyré. "Smooth bilevel programming for sparse regularization." NeurIPS 34 (2021): 1543-1555.

---

> ### Author Response · Authors · 2024-11-13
> **Response to minor weaknesses**
>
> 1. We agree that studying DWF in very large models would be an interesting avenue worth exploring, but our work instead focuses on the analysis of the sparsification and training dynamics on a range of different architectures. Moreover, the scope of our experimental evaluation is in line with many other works on sparse learning (e.g., Lee et al., 2019; Frankle et al., 2021; Lu et al., 2022; Dhahri et al., 2024).
> 2. We argue that random pruning constitutes a meaningful baseline, providing a measure of the intrinsic prunability of a specific network architecture conditional on a learning task (Liu et al., 2022). Regarding iterative pruning methods, these approaches have been found to be impractical and computationally exceedingly expensive and were thus excluded from our comparison. For example, Glandorf et al. (2023) state that iterative magnitude pruning for, e.g., a VGG-19 on CIFAR10/100 requires 860 epochs in their experiments, whereas DWF training is feasible without repeatedly re-training the model.
>
> **References**:
>
> Lee, N., T. Ajanthan, and P. Torr. "SNIP: single-shot network pruning based on connection sensitivity." International Conference on Learning Representations. 2019.
>
> Tanaka, Hidenori, et al. "Pruning neural networks without any data by iteratively conserving synaptic flow." Advances in neural information processing systems 33 (2020): 6377-6389.
>
> Frankle, Jonathan, et al. "Pruning Neural Networks at Initialization: Why Are We Missing the Mark?." International Conference on Learning Representations. 2021.
>
> Lu, Miao, et al. "Learning pruning-friendly networks via frank-wolfe: One-shot, any-sparsity, and no retraining." International Conference on Learning Representations. 2022.
>
> Liu, Shiwei, et al. "The Unreasonable Effectiveness of Random Pruning: Return of the Most Naive Baseline for Sparse Training." International Conference on Learning Representations. 2022.
>
> Glandorf, Patrick, Timo Kaiser, and Bodo Rosenhahn. "HyperSparse Neural Networks: Shifting Exploration to Exploitation through Adaptive Regularization." Proceedings of the IEEE/CVF International Conference on Computer Vision. 2023.
>
> Dhahri, Rayen, et al. "Shaving Weights with Occam's Razor: Bayesian Sparsification for Neural Networks using the Marginal Likelihood." Sixth Symposium on Advances in Approximate Bayesian Inference-Non Archival Track. 2024.

---

> > ### Comment · Reviewer_dKGU · 2024-11-18
> > **Response to rebuttal**
> >
> > I greatly appreciate the authors' in-depth response and explanations regarding the relations to previous works. I'll first respond to each reference separately:
> >
> > [1] - I disagree that the idea is fundamentally different from the one studied in [1], in spite of the different scope and context. I accept that extending this approach to deep neural networks is a natural and sufficiently interesting reason to justify this work, given your in depth analyses of requirements for achieving sparsity.
> >
> > [2] - Your comments regarding [2] are not accurate, in particular, the motivation is given by the eta trick and proximal alternating approaches, but eventually is summed up by step 9 in Algorithm 1, which generalizes to any update based optimization method, similar to weight-decay for L_2 regularization. This implies no overhead (unlike the DWF), and immediate integration to any update method, and since the two methods implement the same effective regularization, it is unclear how the two are fundamentally different.
> >
> > Nevertheless, while I do not fully agree that your method is sufficiently different from both references, I still believe you were not aware of these references, and I do agree that given the scope and interesting analyses you provide both for the onset of sparsity and initializations justifies the work.
> >
> > **regarding the response to the minor weaknesses:**
> >
> > Iterative MP should not be that expensive, especially when performed not epoch-wise (i.e, once per a fixed number of epochs) and should obtain far superior results to random pruning. I don't think quoting statements from other papers' estimation of training times without reproducing those results is in general a correct practice. I agree with the authors that random MP can be a minimal baseline as stated in Liu et al.
> >
> > Given my comments and your response, I feel comfortable raising my score to 5, but not beyond.

---

> ### Author Response · Authors · 2024-11-18
> **Response to [2] of "response to rebuttal"**
>
> Dear reviewer dKGU,
>
> Thank you for your willingness to discuss, as well as for acknowledging the contributions of our submission and the subsequently raised score.
>
> ### Optimization method in [2]
>
> We would greatly appreciate your help in identifying any inaccuracies in our comments. To our best understanding, the approach described in [2] appears to employ the Eta-trick for non-convex penalties, as outlined, for instance, in the section “Extension to non-convex penalties” by Bach (2019), and step 9 in Algorithm 1 incorporates the closed-form solution for the update of the auxiliary variable $s=|w|^{2-p}$ into the training procedure. Reference [2] further states that they derive a “**proximal gradient update step**” in Equation (7), calling it “the main result of this work, and we refer to this method as p-norm Weight Decay (pWD)”. In section 4.3, the authors of [2] state “To limit the scope of this work, we will focus only on the case where s is updated at every w step”, describing an alternating optimization scheme themselves. Regarding step 9 of Algorithm 1 in [2], the authors write “This weight decay step is the same as the one in Eq. (7), assuming the auxiliary parameters s are set before every w step.”, where Eq. (7) references the aforementioned **proximal update step**. These statements and the underlying equations suggest that [2] indeed proposes a proximal optimization scheme (where step 9 is the proximal step), despite steps 6 and 8 of Alg. 1 permitting arbitrary optimization schemes for the construction of the unprojected iterates $\tilde{w}_t$. We would greatly appreciate getting your opinion on this.
>
> ### Additional sparsity-promoting properties of DWF absent in [2]
>
> Aside from the discussion of whether [2] constitutes a proximal optimization scheme or not, we would like to highlight that the induced (or effective) regularizer is not the only relevant sparsity-inducing property in DWF (which is identical between direct $L_{2/D}$ regularization,  $L_{2/D}$ regularization using DWF, and $L_{2/D}$ regularization using pWD).
> Different from DWF, the method in **[2] does not factorize the parameters** of the original objective but rather augments the regularization term using auxiliary variables to obtain a variational formulation of the sparsity-inducing regularizer. Although this formulation yields the same effective regularizer, the optimization behavior using DWF is significantly different in the following two aspects:
>
> 1. Weight factorization introduces artificial rescaling symmetries absent in [2], which, additionally to the theoretical equivalence of DWF with $L_{2/D}$ regularization, promote learning simple and sparse structures due to stochastic collapse (Chen et al., 2024). Ziyin (2024) shows that factorization-induced rescaling symmetries result in the emergence of sparsity even in the absence of explicit $L_2$ regularization. This essentially constitutes a **second sparsifying mechanism in DWF** orthogonal to the sparsity of solutions established in our Theorem 1.
>
> 2. DWF introduces a sparsity-promoting **“rich get richer”** optimization dynamic absent in [2] (cf. Schwarz et al., 2021). Consider factorizing an objective $\mathcal{L}(\boldsymbol{w})$ using two factors as  $\mathcal{L}(\boldsymbol{\omega_1} \odot  \boldsymbol{\omega_2})$. Then the gradients become multiplicatively dependent on the current iterates, i.e., $\nabla_{\boldsymbol{\omega_1}} \mathcal{L}(\boldsymbol{\omega_1} \odot  \boldsymbol{\omega_2}) = \nabla_{\boldsymbol{w}} \mathcal{L}(\boldsymbol{w}) \odot \boldsymbol{\omega_2}$ and likewise for $\boldsymbol{\omega_2}$. This results in small (collapsed) weights receiving small gradient updates while already large (collapsed) weights receive larger updates, leading to small (collapsed) weights remaining largely unaffected by training. The rich get richer effect also does not rely on explicit $L_2$ regularization or the structure of minima and therefore constitutes a third sparsity-promoting mechanism in DWF.
>
> We hope that this response successfully highlighted key optimization differences between DWF and [2] by pointing to sparsity-promoting properties of DWF beyond the equivalence to $L_{2/D}$ regularization.
>
> **References:**
>
> Bach, Francis. “The “η-trick” or the effectiveness of reweighted least-squares.” 2019. URL: https://francisbach.com/the-%CE%B7-trick-or-the-effectiveness-of-reweighted-least-squares/
>
> Schwarz, Jonathan, et al. "Powerpropagation: A sparsity inducing weight reparameterisation." NeurIPS 34 (2021): 28889-28903.
>
> Ziyin, Liu. "Symmetry Induces Structure and Constraint of Learning." ICML. 2024.
>
> Chen, Feng, et al. "Stochastic collapse: How gradient noise attracts sgd dynamics towards simpler subnetworks." NeurIPS 36 (2024).

---

> ### Author Response · Authors · 2024-11-20
> **Revised submission**
>
> Dear reviewer dKGU,
>
> We would like to thank you again for your helpful comments and have uploaded a revised submission. As stated in our previous comment, we have included both references [1] and [2] in the related literature section of our appendix as follows (893-902 in the revised submission):
>
> >A closely related approach to incorporate the induced non-convex $L_q$ regularization ($q<1$) into DNN training was recently proposed by Outmezguine & Levi (2024), who base their method on the $\boldsymbol{\eta}$-trick (Bach, 2012) instead of the $L_2$ regularized weight factorization we study. This method re-parametrizes the regularization function instead of factorizing the network parameters, utilizing a different variational formulation of the $L_q$ quasi-norm. Despite having the same effective $L_q$ regularization, DWF incorporates additional symmetry-induced sparsity-promoting effects through weight factorization and the stochastic collapse phenomenon (Ziyin, 2023; Chen et al., 2024). Another branch of literature studies the representation cost of DNN architectures $f_{w}$ for a specific function $f$, defined as $R(f)=\min_{w: f_{w} = f } \Vert w \Vert_2^2$, showing that the $L_2$ cost of representing a linear function using a diagonal linear network yields the $L_{2/D}$ quasi-norm (Dai et al., 2021; Jacot, 2023).
>
> We hope that this addition adequately acknowledges and incorporates both related works into our submission and thank you again for your feedback.

---

### Meta-Review · Area_Chair_hGMf · 2024-12-05

**Metareview:**

The paper presents an extension of the shallow weight factorization to a deep weight factorization framework, proposing an innovative method for sparsity-inducing training in non-linear neural networks. Discussions between the authors and reviewers were constructive, addressing critical aspects such as performance gains and comparisons with existing approaches, albeit the latter to varying extents. The reviewers ultimately recognized the work as addressing a topic of wide relevance, with its theoretical contributions and experimental findings representing a well-executed advancement in sparse learning. The authors are advised to integrate the valuable insights provided by the knowledgeable reviewers.

**Additional Comments On Reviewer Discussion:**

As above:

Discussions between the authors and reviewers were constructive, addressing critical aspects such as performance gains and comparisons with existing approaches, albeit the latter to varying extents. The reviewers ultimately recognized the work as addressing a topic of wide relevance, with its theoretical contributions and experimental findings representing a well-executed advancement in sparse learning.

---

### Decision · Program_Chairs · 2025-01-22

Accept (Poster)